# The AICL-KLRF1 axis supports CD4-CD8 T cell communication and cytokine competence in pre-exhausted CD8+ T cells

Matthias Barone [1,8], Stefan Peidli [2,3,8], Anika Neuschulz [1], Karla Riesterer[1], Christina Iwert[1], Laia Junquera [1], Somesh Sai [4], Olufemi Bolaji[1], Diana Bakoueva[1], Christine Appelt [1], Benedikt Obermayer[5], Bertram Klinger [2,3], Alexandra Trinks[6], Anja Sieber [2], Nils Blüthgen [2,3] & Birgit Sawitzki [1,7]✉

## Abstract

**Memory-like or precursor exhausted (Tpex) CD8+ T cells are a critical reservoir in chronic infections and cancer, yet the signals sustaining their cytokine production remain unclear. Here, we identify KLRF1 as part of a CD4–CD8 communication axis that supports cytokine production in late-differentiated human CD8+ T cells. KLRF1 is upregulated in late-differentiated CD8+ T cells, and neutralizing KLRF1 reduces TNF and IFN-γ production. Differentiated CD4+ T cells express the KLRF1 ligand AICL, and in co-culture only AICL+ - not AICL− - CD4+ T cells enhance cytokine output in CD8+ T cells. Using spatial proteomics of lung adenocarcinoma and adjacent tissue, we found that CD4+ AICL+ and CD8+ KLRF1+ T cells are enriched and spatially interacting in non-tumor regions, whereas both populations are reduced within tumor tissue. Single-cell RNA-seq of tissue samples and scRNA/ATAC analyses of circulating immune cells further showed that CD8+KLRF1+ T cells display a Tpex-like transcriptional and chromatin-accessibility profile. Together, these data identify the AICL–KLRF1 axis as a CD4+–CD8+ communication pathway that supports cytokine competence in late-differentiated CD8+ T cells.**

**Keywords** Cytotoxic T Cell; KLRF1; T Cell Exhaustion; Tumor Immunology
**Subject Categories** Cancer; Immunology; Signal Transduction

## Introduction

The cytokines interferon gamma (IFN-γ) and tumor necrosis factor (TNF) are considered important positive regulators of tumor control and antitumor immune responses. Direct activity of both cytokines on tumor cells can mediate cellular senescence (Braumüller et al, 2013), apoptosis (Castro et al, 2018; Montfort et al, 2019), and ferroptosis (Wang et al, 2019). Besides direct effects, both cytokines can critically influence tumor control through action on stromal cells (Kammertoens et al, 2017).

In addition, IFN-γ and TNF regulate the activation and maturation of macrophages and dendritic cells, initiating a sustained cancer-immunity cycle (Chen and Mellman, 2013; Paul et al, 2019; Wang et al, 2019). Indeed, IFN-γ cross-regulates the expression of inflammatory mediators and immune effectors such as IL-12 (Hu and Ivashkiv, 2009; Garris et al, 2022), and the expression scores of IFN-γ-induced genes correlate with the responsiveness to anti-PD-1 therapy in metastatic melanoma, head and neck squamous cell carcinoma, gastric cancer (Ayers et al, 2017), urothelial and non-small cell lung cancer (NSCLC) (Higgs et al, 2018; Karachaliou et al, 2018).

T cells and especially CD8+ cytotoxic T lymphocytes (CTLs) are a major source of IFN-γ and TNF in the tumor microenvironment (TME). Indeed, CTLs have gained substantial attention in antitumor therapy and there is compelling evidence for a role of the CD8+ T cells in tumor control. Consequently, high densities of CD8+ memory and cytotoxic T cells are associated with a longer disease-free survival and/or overall survival (Fridman et al, 2012), especially tissue-resident CD8+ CD103+ tumor-infiltrating lymphocytes (Molodtsov and Turk, 2018; Park et al, 2019).

However, despite the presence of tumor-infiltrating CD8+ T cells (TILs) recognizing tumor-associated antigens, cancer often progresses uncontrollably. The constant antigen exposure drives T cells into a hyporesponsive state that enables them to persist under conditions of chronic stimulation by impairing the response to TCR engagement and increasing the expression of inhibitory receptors such as PD-1, TIM-3 and CTLA-4. This cell state is characterized by a loss of proliferative potential, reduced effector function and cytokine production, and the cells have therefore been

---

[1]Berlin Institute of Health (BIH) at Charité, Charité Universitätsmedizin Berlin, Berlin, Germany. [2]Institute of Pathology, Charité, Charité Universitätsmedizin Berlin, Berlin, Germany. [3]Institute of Biology, Humbolt-Universität zu Berlin, Berlin, Germany. [4]Max Delbrück Center for Molecular Medicine in the Helmholtz Association, Berlin, Germany. [5]Core Unit Bioinformatics (CUBI), Berlin Institute of Health (BIH) at Charité, Charité Universitätsmedizin Berlin, Berlin, Germany. [6]Bioportal Single Cells, Berlin Institute of Health at Charité-Universitätsmedizin Berlin, Corporate Member of Freie Universität Berlin and Humboldt-Universität zu Berlin, 10117 Berlin, Germany. [7]Der Simulierte Mensch, a science framework of Technische Universität Berlin and Charité - Universitätsmedizin Berlin, Berlin, Germany. [8]These authors contributed equally: Matthias Barone, Stefan Peidli. ✉E-mail: birgit.sawitzki@bih-charite.de

defined as terminally differentiated and exhausted, respectively (Kallies et al, 2020; Philip and Schietinger, 2022).

T cell differentiation trajectories have traditionally been identified based on the surface markers CD45RA and CCR7, categorizing them into naïve ($T_N$), central memory ($T_{CM}$), effector memory ($T_{EM}$), and terminally differentiated effector memory ($T_{EMRA}$) T cell subsets. Functional studies, including our own however, have unveiled heterogeneities within the $T_{EMRA}$ population, highlighting the limitations of using CD45RA and CCR7 alone in accurately characterizing T cell differentiation and changes in functionality (Pachnio et al, 2016; Tian et al, 2017; Patil et al, 2018; Hashimoto et al, 2019; Truong et al, 2019). Indeed, some $T_{EMRA}$ cells retain cytokine production potential, and intra-tumor CD8[+] $T_{EMRA}$ numbers correlate with treatment responsiveness in advanced hepatocellular carcinoma (Cappuyns et al, 2023).

In line with these findings, recent studies revealed that dysfunctional TILs consist of at least two hyporesponsive populations, one of which shows memory-like features and was therefore defined as precursor exhausted T cells (Tpex) (Kallies et al, 2020). Tpex are characterized by TCF-1 and PD-1 expression, while lacking TIM-3 and granzyme B. These cells exhibit plasticity, regaining effector function when removed from the tumor. Importantly, Tpex can self-renew and drive the proliferative burst after PD-1 blockade (Im et al, 2016; Sade-Feldman et al, 2018; Miller et al, 2019; Siddiqui et al, 2019). They are believed to sustain T cell responses to chronic antigens, not only in cancer but also in chronic viral infections such as HIV and HCV (He et al, 2016; Utzschneider et al, 2016; Wieland et al, 2017).

In contrast to Tpex, the second subset consists of terminally exhausted T cells (Tex), which lack TCF-1 expression and cannot be rescued by immune checkpoint blockade (ICB). Tex are driven into a fixed, dysfunctional state by TOX, which induces irreversible epigenetic changes through histone acetylation (Philip et al, 2017). TOX expression correlates with high inhibitory receptor levels, and Tex typically expresses exhaustion markers like TIM-3. Recent data suggest that TCF-1[+] Tpex act as precursors, continuously replenishing the Tex pool. Tpex are primarily located in the tumor stroma, while Tex are found within the tumor (Miller et al, 2019; Philip and Schietinger, 2022).

Since "memory"-like/precursor exhausted T cells serve as an important functional reservoir (Rahim et al, 2023), there is great interest in understanding the signals that regulate their cytokine production and release. Previously, we showed that progressive expression of killer-like receptors (KLRs) KLRB1, KLRG1, KLRF1, and the G protein-coupled receptor 56 (GPR56) defines the cytokine production potential of human CD4[+] T cells (Truong et al, 2019).

In this study, we investigated whether this also applies to CD8[+] T cells, which express the corresponding ligands and serve as signal providers, and how this is linked to their spatial organization and molecular phenotype within the lung adenocarcinoma.

Our data confirmed a similar progressive acquisition of KLRs and GPR56 during human CD8[+] T cell differentiation. Interestingly, KLRF1 is primarily expressed by $T_{EMRA}$ cells. Blocking experiments using both plate-bound and soluble anti-KLRF1 antibodies showed that interference with KLRF1 signaling reduces TNF and IFN-γ production.

Activated CD4[+] T helper cells upregulate the KLRF1 ligand AICL (Hamann et al, 1997), and sorting-based co-culture assays revealed that CD8[+] T cells co-cultured with AICL[+] - but not AICL[−] - CD4[+] T cells exhibit increased cytokine production, consistent with a role for AICL–KLRF1 interaction–based support of cytokine competence. Spatial proteomics of lung adenocarcinoma and adjacent tissue revealed close interactions between AICL-expressing CD4[+] T cells and CD8[+] KLRF1[+] T cells in non-tumor areas, while both cell populations were reduced within tumor tissue. Single-cell RNA sequencing (scRNA-seq) of those tissue samples and multimodal scRNA/ATAC analysis of circulating immune cells further showed that CD8[+] KLRF1[+] T cells exhibit a Tpex-like, memory-associated profile.

Together, these findings identify the AICL–KLRF1 axis as an additional CD4[+]–CD8[+] communication pathway that supports cytokine competence in late-differentiated CD8[+] T cells and may offer new avenues to enhance TIL functionality in cancer.

## Results

### KLRF1 expression marks terminally differentiated CD8[+] T cells that retain the capacity to produce IFN-γ and TNF

We first examined whether the differentiation stage-specific expression of KLRB1, KLRG1, GPR56, and KLRF1 previously described in CD4[+] T cells (Truong et al, 2019) also applies to CD8[+] T cells. Since high KLRB1 expression is a characteristic feature of mucosal-associated invariant T (MAIT) cells, we used co-staining of the invariant TCRα chain Vα7.2 to distinguish and exclude KLRB1[high]-expressing MAIT cells from KLRB1 intermediate (KLRB1[int]) conventional CD8[+] T cells (Fig. EV1A). In thereby identified conventional CD8[+] T cells we used CD45RA and CCR7 to classify $T_N$, $T_{CM}$, $T_{EM}$, and $T_{EMRA}$ cell subsets and mapped KLRB1, KLRG1, KLRF1, and GPR56 expression onto these differentiation stages. The progressive acquisition of KLRs and GPR56 expression at different differentiation stages closely mirrored that we previously described for CD4[+] T cells (Fig. 1A). However, CD8[+] T cells appeared to upregulate KLRG1 first before acquiring KLRB1.

We then confirmed that progressive expression of KLRs and GPR56 correlates with TNF and IFN-γ cytokine expression (Fig. 1B). In line with CD4[+] T cells (Truong et al, 2019), TNF and IFN-γ co-production potential of CD8[+] T cells increased with the expression of the two early markers–KLRG1 and KLRB1, while upregulation of GPR56 and KLRF1 was associated with a decline of cytokine production in late-differentiated $T_{EMRA}$ populations (Fig. 1C). Interestingly, while CD8[+] KLRF1[+] T cells retain robust TNF and IFN-γ production, the additional expression of GPR56 is associated with a reduction in this capacity. The frequency of TNF- and IFN-γ–producing cells is notably higher among CD8[+] KLRG1[+] KLRB1[int] GPR56[−] KLRF1[+] T cells compared to their GPR56[+] counterparts (Figs. 1D and EV1B), suggesting that GPR56 may negatively modulate cytokine output even within KLRF1[+] populations. This observation is in line with previous reports which assigned GPR56 expressing CD8[+] T cells a terminal differentiation and exhausted state (Kared et al, 2024).

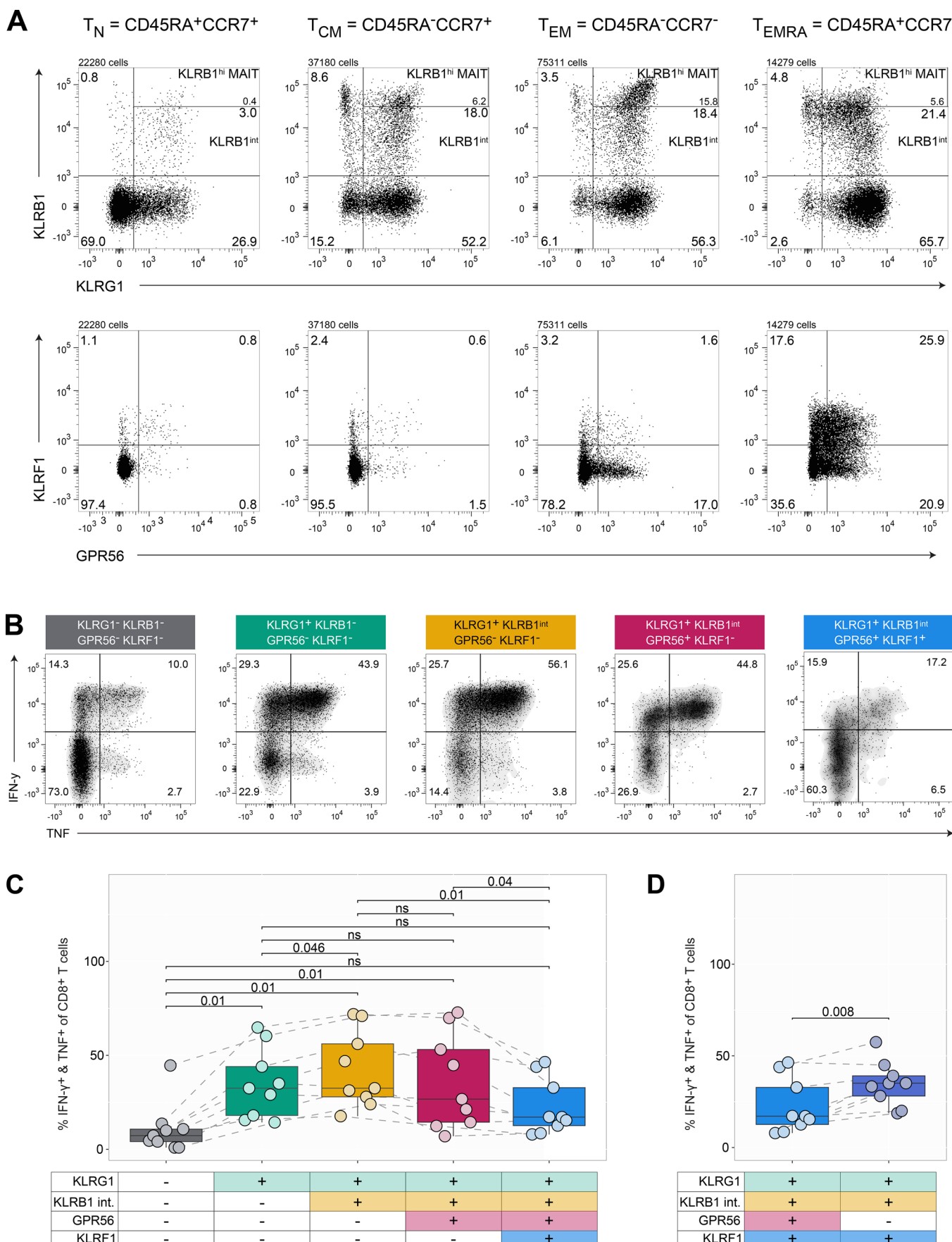

◀

**Figure 1. Progressive expression of KLRs and GPR56 correlates with CD8⁺ T cell differentiation and cytokine production.**

(A) Exemplary dot plots of pre-gated CD8⁺ T cells from unstimulated CD3⁺-enriched PBMCs showing KLRB1 versus KLRG1 and KLRF1 versus GPR56, respectively, of pre-gated naïve ($T_N$), central memory ($T_{CM}$), effector memory ($T_{EM}$), and terminally differentiated effector memory ($T_{EMRA}$) CD8⁺ T cells from unstimulated CD3⁺-enriched PBMC. (B) Exemplary dot plots of T cells from CD3⁺-enriched PBMCs stimulated for 6 h depicting TNF versus IFN-γ of pre-gated subpopulations based on their progressive expression of KLRG1, KLRB1, GPR56, and KLRF1. An additional shading was added to the dot plots to highlight the populations. (C) TNF and IFN-γ production of the stimulated subpopulations as defined in panel (B). Friedman test was conducted to assess differences across multiple related groups, followed by post hoc pairwise comparisons using paired Wilcoxon signed-rank test with Benjamini & Hochberg adjustment. Exact $p$ values are noted in the plot; ns $p > 0.05$ ($n = 9$, biological replicates). Box plots show the median (center line) and interquartile range (box limits 25th to 75th percentile, 1x IQR), whiskers extend to the largest value no more than 1.5x IQR from the box. (D) Same samples as in panel C showing the influence of GPR56 of $T_{EMRA}$ cells on TNF and IFN-γ production ($n = 9$, biological replicates). Paired Wilcoxon signed-rank test. Exact $p$ value is noted in the plot. Box plots show the median (center line) and interquartile range (box limits 25th to 75th percentile, 1x IQR), whiskers extend to the largest value no more than 1.5x IQR from the box. Source data are available online for this figure.

Taken together, our data indicate that KLRF1 might exert a supportive function for memory-like late-differentiated CD8⁺ T cells in maintaining their cytokine production capacity.

## CD4⁺ T cells are essential for KLRF1-driven IFN-γ and TNF production by CD8⁺ T Cells

Since KLRF1 was reported to stimulate NK cell cytokine release (Welte et al, 2006), we investigated whether the same applies to CD8⁺ T cells. Enriched total CD3⁺ T cells (comprising CD4⁺ and CD8⁺ T cells) were stimulated for 6 and 24 h with anti-CD3 in the presence of either a blocking anti-KLRF1 antibody or an isotype control and KLRF1 surface expression as well as TNF and IFN-γ production determined by flow cytometry (Fig. 2A). Plate-bound anti-KLRF1 induced a rapid and pronounced loss of surface KLRF1 on CD8⁺ T cells at both timepoints (Fig. 2B), accompanied by a significant reduction in the proportion of TNF- and IFN-γ producing CD8⁺ T cells in comparison to isotype-treated cells. To exclude that this effect was due to receptor cross-linking, we additionally performed neutralization experiments using soluble anti-KLRF1 (Fig. 2B). Soluble blockade also reduced surface KLRF1 expression at 6 and 24 h, although less strongly than the plate-bound antibody. The functional readout mirrored this difference in receptor modulation: cytokine production decreased early under plate-bound conditions, whereas the reduction became significant only at 24 h under soluble blockade once KLRF1 down-modulation was sustained. These results demonstrate that interference with KLRF1 signaling, independent of the blocking modality, diminishes TNF and IFN-γ production.

The positive effect of KLRF1 on TNF and IFN-γ production by CD8⁺ T cells is in line with previous findings, which reported a positive effect of KLRF1 on IFN-γ production by NK cells and CD8⁺ T cells (Welte et al, 2006; Kuttruff et al, 2009).

Activation-induced C-type lectin (AICL), encoded by C-type lectin domain family 2 member B (*CLEC2B*), is the only identified binding partner of KLRF1 (Welte et al, 2006). Activation-induced upregulation of AICL has been described in lymphocytes, including T and NK cells (Hamann et al, 1997), but expression has been reported also on monocytes, macrophages and granulocytes (Welte et al, 2006).

Next, we examined whether the KLRF1 effect is CD8⁺ T-cell intrinsic or depends on signals from CD4⁺ T cells present in the CD3⁺ cultures. Therefore, we repeated the experiments with purified CD8⁺ T cells. Similarly to stimulated total CD3⁺ T cells, adding the blocking anti-KLRF1 antibody abolished KLRF1 surface expression on CD8⁺ T cells (Fig. 2C). However, in the absence of co-cultured CD4⁺ T cells this was not accompanied by a significant

reduction in the proportion of cytokine producing CD8⁺ T cells. Thus, the reduction in cytokine production caused by anti-KLRF1 antibody treatment is not due to the modulation of KLRF1 expression on the cell surface but rather to the abolishment of KLRF1-mediated interaction between CD4⁺ and CD8⁺ T cells.

In summary, KLRF1 signaling can promote TNF and IFN-γ production by CD8⁺ T cells when co-cultured in the presence of CD4⁺ T cells.

## Stimulation-induced upregulation of AICL in CD4⁺ KLRF1⁺ T Cells

Since KLRF1 internalization reduced TNF and IFN-γ production by CD8⁺ T cells only in the presence of CD4⁺ T cells, we hypothesized that the putative KLRF1 ligand is expressed on CD4⁺ T helper cells during T cell stimulation.

We therefore analyzed the cell surface expression of AICL on CD4⁺ and CD8⁺ T cells during anti-CD3 and anti-CD28 antibody-mediated stimulation of PBMCs (Fig. 3A). AICL was predominantly expressed by stimulated CD4⁺ T cells and to a smaller degree by CD8⁺ T cells (Fig. 3B, left panel). The stimulation-dependent upregulation of AICL on CD4⁺ T cells was transient, peaking at 6 h and returning to baseline levels by 24 h (Fig. 3B, right panel). The proportion of CD4⁺ T cells upregulating AICL expression upon stimulation was reproducible but relatively small, 1% on average. This led us to investigate the distinct phenotypic profiles that characterize this population. The genes encoding AICL (*CLEC2B*) and KLRF1 (*KLRF1*) are adjacently located in the Natural Killer gene complex (NKC) in a tail-to-tail orientation (Bartel et al, 2013). For NK cells, co-regulation/-expression of AICL and KLRF1 has been reported (Bartel et al, 2013). Therefore, we hypothesized that, similar to NK cells, AICL expression in CD4⁺ T cells is primarily associated with KLRF1. To test this, we applied our previously established marker set and mapped AICL expression onto unstimulated and stimulated CD4⁺ T cells expressing KLRG1, KLRB1, GPR56, and KLRF1 (Appendix Fig. S1). Indeed, AICL upregulation upon stimulation was observed exclusively in KLRG1⁺ KLRF1⁺ subsets (Fig. 3C). Interestingly, GPR56 was inversely associated with AICL expression in CD4⁺ T cells, as CD4⁺ T cells co-expressing KLRF1 and GPR56 showed markedly lower levels of AICL.

To test whether a CD4⁺ T cell–derived AICL signal, rather than the mere presence of CD4⁺ T cells, is required for the increased cytokine production in CD8⁺ T cells, we FACS-sorted differentiated CD4⁺ KLRG1⁺ KLRF1⁺ AICL⁺ T cells and their CD4⁺ KLRG1⁺ KLRF1⁺ AICL⁻ counterparts. These subsets were co-cultured with autologous CD8⁺ T cells for 24 h, alongside a CD8⁺-

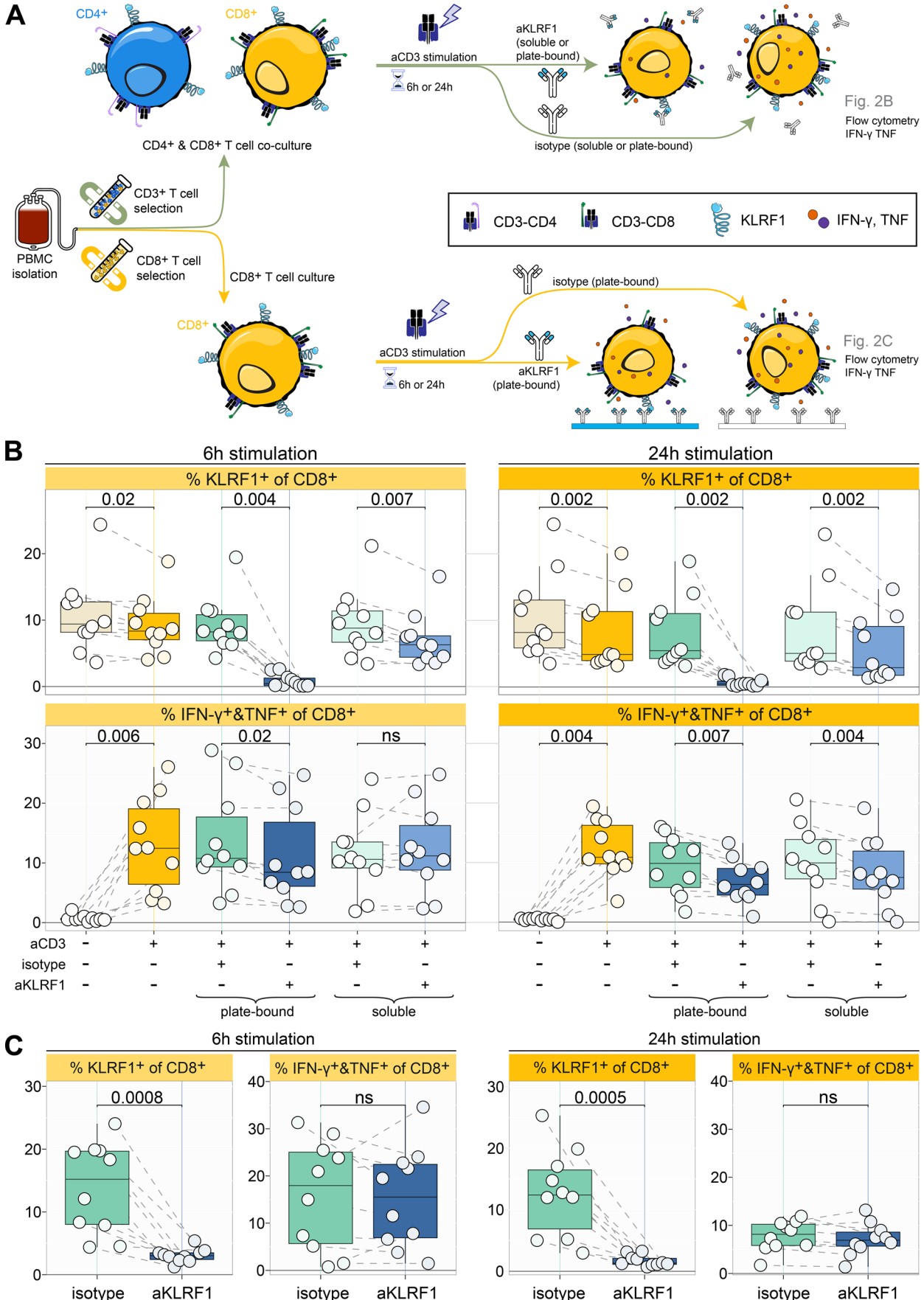

◄

**Figure 2. TNF and IFN-γ production by CD8⁺ T cells upon KLRF1 neutralization.**

(A) Experimental workflow summarizing the setup of the culture conditions and subsequent flowcytometric analysis of CD8⁺ T cells in Fig. 2. (B) KLRF1 expression and total TNF&IFN-γ production of CD8⁺ T cells. Cells were unstimulated (aCD3−), 6 h ($n = 10$, biological replicates) or 24 h ($n = 10$, biological replicates) stimulated (aCD3 +). Co-culture was stimulated together with either plate-bound or soluble anti-KLRF1 (aKLRF1 +) or the isotype control antibody (isotype +). For every experiment, the corresponding IgG1 isotype antibody control is shown. The Friedman rank-sum test was used to assess differences among groups. The Shapiro–Wilk test indicated significant deviation from a normal distribution under several conditions. Post-hoc paired Wilcoxon signed-rank test was performed with Benjamini & Hochberg adjustment across all comparisons within each plot. The relevant comparisons and exact $p$ values are shown in each plot; ns $p > 0.05$. Box plots show the median (center line) and interquartile range (box limits 25th to 75th percentile, 1x IQR), whiskers extend to the farthest values but no more than 1.5x IQR from the box. (C) KLRF1 expression and total TNF&IFN-γ production of CD8⁺ T cells stimulated for 6 h (left, $n = 10$, biological replicates) or 24 h (right, $n = 10$, biological replicates). CD8⁺ T cells were stimulated together with plate-bound anti-KLRF1 (aKLRF1) or IgG1 isotype antibody, respectively. Shown are the fraction of CD8⁺ KLRF1⁺ T cells and total TNF&IFN-γ production as in panel B. Tested for normality using the Shapiro–Wilk test, followed by a two-tailed *t*-test. Exact $p$ values are noted in the plot; ns $p > 0.05$. Box plots show the median (center line) and interquartile range (box limits 25th to 75th percentile, 1x IQR), whiskers extend to the minimum and maximum values. Source data are available online for this figure.

only culture that served as reference for cytokine production (Fig. 3D). CD8⁺ T cells co-cultured with AICL-expressing CD4⁺ T cells showed a trend toward increased frequencies of total TNF⁺ and/or IFN-γ⁺ cells compared with CD8⁺ T cells cultured alone (Fig. 3E). This effect was largely driven by a significant increase in TNF positive CD8⁺ T cells (Fig. 3F). In contrast, co-culture with AICL negative CD4⁺ T cells failed to enhance TNF or IFN-γ production and resulted in significantly lower proportions of TNF⁺ and/or IFN-γ⁺ producing CD8⁺ T cells compared with co-cultures containing AICL-expressing CD4⁺ T cells (Fig. 3E).

Our findings reveal that AICL, the only known ligand for KLRF1, is transiently upregulated in a small subset of stimulated CD4⁺ T cells, reaching maximal expression at 6 h. Notably, AICL upregulation occurs exclusively in CD4⁺ KLRF1⁺ T cell subsets, suggesting a co-regulation mechanism similar to that observed in NK cells. Interestingly, GPR56 was inversely associated with this process, as CD4⁺ T cells co-expressing KLRF1 and GPR56 exhibited significantly reduced AICL expression, indicating a potential regulatory interaction, similar to the inhibitory role of GPR56 in NK cells (Chang et al, 2016).

## Lack of activated AICL-expressing CD4⁺ T helper cells in lung adenocarcinoma tissue

Based on our in vitro findings indicating that the AICL-KLRF1 axis contributes to cytokine production in terminally differentiated CD8⁺ T cells, we hypothesized that this CD4⁺-CD8⁺ T cell interaction may be compromised in the tumor microenvironment, potentially promoting CD8⁺ T cell exhaustion and a reduction in Tpex subsets. Indeed, low expression of *CLEC2B* is associated with poor overall survival in different tumors, e.g., pancreatic adenocarcinoma (Li et al, 2022) and worse disease progression in melanoma (Zhang et al, 2023). Furthermore, AICL is significantly associated with the abundance of infiltrating immune cells and immunoregulation-related genes (Li et al, 2022), suggesting a potential role in supporting TIL function. This type of support is well-established in tissue-resident NK cells, where AICL-expressing myeloid cells have been shown to activate NK cells, enhancing their cytolytic capacity and cytokine production (Welte et al, 2006). In addition to myeloid cells, KLRF1-expressing NK cells have been reported to be enriched near, and activated by, AICL-expressing endothelial cells (Geldhof et al, 2022). This raises the possibility that AICL-expressing CD4⁺ T cells may play a similar role in supporting tissue-resident CD8⁺ T cells to sustain their cytokine production potential.

To explore the connection between the cellular microenvironment and the exhaustion phenotype of CD8⁺ TILs, we performed imaging mass cytometry (IMC) on lung adenocarcinoma and adjacent tissue sections assembled into a tissue microarray (Fig. 4A). We designed a 37-plex panel incorporating markers to define major immune compartments (CD19, CD3, CD4, CD8, TCRδ), along with markers associated with T cell activation (e.g., HLA-DR, CD25), tissue residency (CD69, CD103), and exhaustion (e.g., PD-1, TIM-3) (Appendix Table S5). To specifically examine interactions between AICL-expressing CD4⁺ T cells and KLRF1-expressing CD8⁺ T cells, we included antibodies targeting AICL and KLRF1. Segmentation masks were generated with the Bodenmiller workflow (Windhager et al, 2023) and single cell IMC data analyzed with an adapted SPECTRE (Ashhurst et al, 2022) code written in R language. We additionally used CK18 as a marker for tumor areas and created tumor masks in Fiji that were loaded into R for further analysis (Fig. 4A).

We applied a Boolean gating strategy to the IMC dataset to separate CD19⁺ B cells (Gate 1), CD19⁻ CD3⁺ T cells (Gate 2), CD19⁻ CD3⁻ CD45⁺ non-B/non-T immune cells (Gate 3), and CD19⁻ CD3⁻ CD45⁻ non-immune stromal and epithelial cells (Gate 4, Fig. EV2A; Appendix Fig. S2). We first checked AICL expression in each gated population and found that T cells produced significantly more AICL compared to non-T immune cells or CD45⁻ cells (Fig. EV2B). Because B cells were extremely rare and detected at appreciable frequencies in only one of the ten analyzed ROIs, they were excluded from further analysis.

The T cell and the remaining non-T immune subsets were subdivided with unsupervised clustering using PhenoGraph (Levine et al, 2015) according to the expression levels of 26 markers (T cells, Fig. 4B) and 18 markers, respectively (non-B/non-T immune cells, Fig. EV2C). Plotting the T cell subset in a UMAP embedding revealed a homogenous mixture of lung adenocarcinoma versus adjacent tissue samples (Fig. 4C) with the main T cell lineages (CD4⁺, CD8⁺, γδ T cells) well separated across the UMAP embedding (Fig. 4D,E).

We first verified that the progressive upregulation of KLRs was detectable within the CD3⁺ T cell subset of the IMC data. Specifically, we investigated whether AICL⁺ T cells co-express KLRF1. Plotting KLRG1 vs. KLRF1 expression, with cells color-coded by their AICL expression, confirmed that AICL expression was predominantly found in KLRG1⁺ KLRF1⁺ double-positive T cells (Fig. EV2D). We then tested our hypothesis that AICL expression in CD4⁺ T cells was decreased in lung adenocarcinoma as compared to adjacent tissue. Pooling all CD4⁺ T cell clusters confirmed significantly lower AICL levels in tumor tissue (Fig. 4F).

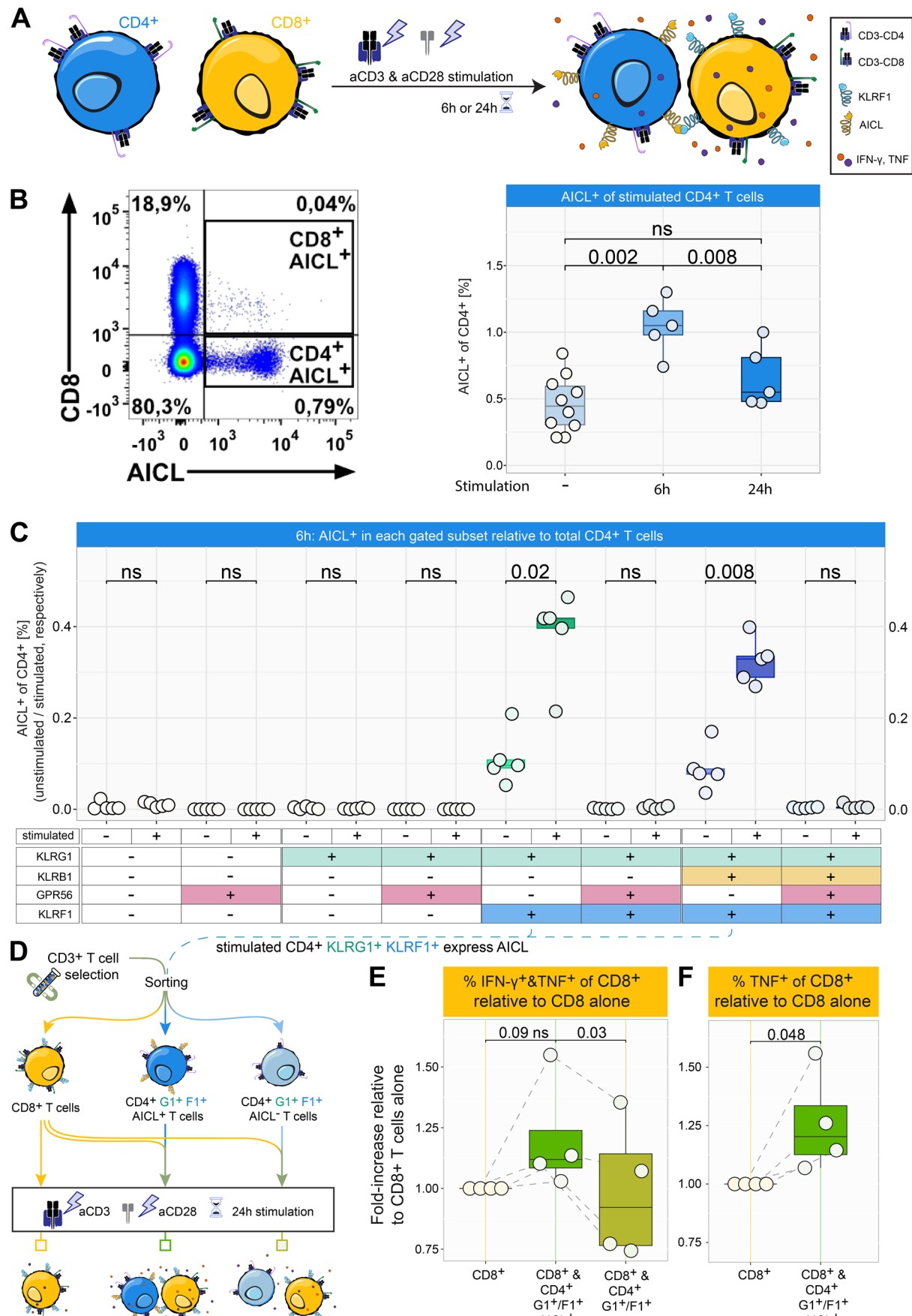

Figure 3.   Stimulation of CD4⁺ T cells results in upregulation of the KLRF1 ligand AICL.

**Figure 3.   Stimulation of CD4+ T cells results in upregulation of the KLRF1 ligand AICL.**

(A) Experimental workflow summarizing the setup of the culture conditions and subsequent flowcytometric analysis of CD4+ T cells in Fig. 3. (B) Surface expression of AICL of CD3+-enriched CD4+ T cells left unstimulated or stimulated with anti-CD3 and anti-CD28 for 6 or 24 h ($n = 5$, biological replicates). Tested for normality using the Shapiro–Wilk test, followed by a paired two-tailed $t$-test with Benjamini & Hochberg adjustment. Exact p-values are noted in the plot; ns $p > 0.05$. Box plots show the median (center line) and interquartile range (box limits 25th to 75th percentile, 1x IQR), whiskers extend to the minimum and maximum values. (C) CD3+-enriched T cells stimulated with anti-CD3 and anti-CD28 for 6 h and gated for CD4+ T cells ($n = 5$, biological replicates). Shown is the proportion of total gated CD4+ T cells that expressed the indicated combinations of KLRG1, KLRB1, GPR56, and KLRF1 (see Appendix Fig. S1). Tested for normality using the Shapiro–Wilk test, followed by a paired two-tailed $t$-test with Benjamini & Hochberg adjustment. Exact $p$ values are noted in the plot; ns $p > 0.05$. Box plots show the median (center line) and interquartile range (box limits 25th to 75th percentile, 1x IQR), whiskers extend to the farthest values but no more than 1.5x IQR from the box hinges. (D) Experimental design of the FACS-sorted enrichment of AICL-expressing CD4+ T cells. CD3+ T cells were enriched from PBMC, and differentiated CD4+ T cells were sorted into KLRG1+ / KLRF1+ AICL+ (G1+/F1+ AICL+) and KLRG1+/KLRF1+ AICL− subsets (G1+/F1+ AICL−). Sorted CD4+ subsets were then co-cultured at a 1:20 ratio with autologous CD8+ T cells for 24 h, alongside with a CD8+-only T cells alone to allow direct comparison of cytokine outcomes. (E) CD8+ TNF+/IFN-γ+ T cells expressed as fold change relative to the CD8-only condition ($n = 4$, biological replicates). Statistical comparisons were performed using a one-tailed $t$-test with Benjamini–Hochberg correction. Exact $p$ values are noted in the plot, also for ns $p > 0.05$. Box plots show the median (center line) and interquartile range (box limits 25th to 75th percentile, 1x IQR), whiskers extend to the farthest values but no more than 1.5x IQR from the box. (F) CD8+ TNF+ T cells were analyzed analogously ($n = 4$, biological replicates). Statistical comparison was performed using one-tailed $t$-test. Exact $p$ value is noted in the plot. Box plots show the median (center line) and interquartile range (box limits 25th to 75th percentile, 1x IQR), whiskers extend to the minimum and maximum values. Source data are available online for this figure.

AICL levels were consistently lower in the lung adenocarcinoma tissue, regardless of whether we probed T helper cells inside the tumor or the surrounding tumor stroma (Fig. EV2E,F).

Beyond overall AICL expression, we were particularly interested in shifts within the T cell compartment (Fig. 4G), especially among CD4+ T helper cells, which may interact with and support memory-like Tpex cells. Clustering of the T cell space resulted in nine different CD4+ T cell clusters (Fig. 4B). In the annotation of T cell clusters, we considered not only markers associated with known functional states but also the expression of KLRG1 (G), KLRF1 (F), and AICL (A). In addition to a FoxP3+ Treg cluster, we identified two clusters in which cells lacked KLRG1, KLRF1, and AICL expression (CD4+ G⁻F⁻A⁻CD31hi; CD4+ G⁻F⁻A⁻). The remaining six clusters were composed of cells expressing one or two of these markers and were further characterized by distinct marker profiles, including high PD-1 expression (CD4+ G⁻F⁻A^lo PD-1+), T-bet expression (CD4+ G+F^lo A^lo T-bet+), and high PD-L1 levels (CD4+ G⁻F⁻A^lo PD-L1hi). Additionally, cells belonging to three clusters expressed all three differentiation trajectory markers, including AICL, as well as high levels of HLA-DR alongside other activation or exhaustion markers (CD4+ G+F+A+, CD4+ G+F+A+ TIM-3+, CD4+CD8+ G+F+A+). Among these, the CD4+CD8+ double-positive T cluster was particularly notable, as it was significantly more abundant in non-tumor adjacent tissue compared to adenocarcinoma sections (Fig. 4G). In contrast, the PD-1 high cluster, which lacked activation markers and showed minimal AICL expression, was enriched in lung adenocarcinoma tissue samples.

Taken together, our findings suggest that activated AICL-expressing CD4+ T cells, which may support cytokine production in CD8+ T cells, are significantly reduced in lung adenocarcinoma tissue compared to adjacent non-tumor areas. This reduction is accompanied by an increase in PD-1hi CD4+ T cell subsets lacking activation markers and showing minimal AICL expression, which may contribute to an altered immune microenvironment in the tumor.

## Abolished interaction between CD8+ KLRF1+ T Cells and activated CD4+ AICL+ T helper cells in lung adenocarcinoma tissue

We observed that one KLRG1+ KLRF1+ AICL+ cluster showed expression of both, CD4 and CD8 (Fig. 4B). This could either result from a mixture of CD4+ and CD8+ T cells or reflect an enrichment of double-positive (CD4+CD8+) T cells. Examining the UMAP embedding, we noticed that the CD4+CD8+ double-positive cluster was divided into two subpopulations: one positioned between the CD4+ and CD8+ T cell clusters, and the other fully embedded within the CD4+ T cell clusters (Fig. 4E). To determine whether this cluster represented a mixture of distinct cell types with similar expression profiles, we plotted CD4 versus CD8 expression. This revealed one population expressing CD4 alone, and a second population consisting of both CD4+CD8+ double-positive and CD8+ single-positive T cells (Fig. 5A left). The clustering of CD4+ and CD8+ T cells within the same T cell cluster, suggests not only similar marker expression, but also spatial proximity and a potential interaction between these two T cell types. To further explore this interaction, we manually separated the cluster by gating on CD8+ T cells (Fig. 5A, left, red horizontal line) and recalculated the heatmap, designating the newly created clusters as "split" (Fig. 5A, right). Both the CD4 "split" and CD8 "split" clusters exhibited a similar phenotype, characterized by high KLRG1 and KLRF1 expression. While expression differences were subtle, split CD4+ T cells showed higher AICL expression, whereas split CD8+ T cells displayed increased levels of the degranulation marker CD107a. Among the identified CD8+ T cell clusters, only the split CD8+ T cells exhibited high HLA-DR expression, indicating a clear activation signature (Fig. 5B). The remaining CD8+ T cell clusters segregated into three KLRF1⁻ but HLA-DR+ subsets (CD8+ G⁻F⁻A+ CD69hi CD103hi, CD8+ G⁻F⁻A+ Ki67hi, CD8+ G⁻F⁻A+ CD38hi) and three KLRF1⁻ HLA-DR⁻ subsets (CD8+ G⁻F⁻A⁻ CD69+CD103+, CD8+ G+F⁻A+ GZMBhi, CD8+ G⁻F⁻A⁻). Confirming our findings, the split CD8+ T cell cluster was significantly more abundant in adjacent tissue samples compared to adenocarcinoma sections (Fig. 5C). And in line with previous reports (Huang et al, 2024; Tanoue et al, 2024) the split CD8+ T cell cluster was found significantly more in the tumor stroma than inside the tumor tissue (Fig. 5D).

We then used the split AICL+ CD4+ T cells as a reference to calculate distances to and direct neighbors of all CD8+ T cell clusters in non-affected tumor adjacent tissue. Distance measurements showed that the split CD4+ cluster was closest to the split CD8+ cluster (Fig. 5F, left; median distance = 603px). The second closest CD8+ T cell cluster, which also displayed activation characteristics and included proliferating cells (CD8+ G⁻F+A+ Ki67hi, median distance = 607px), was similarly positioned. In

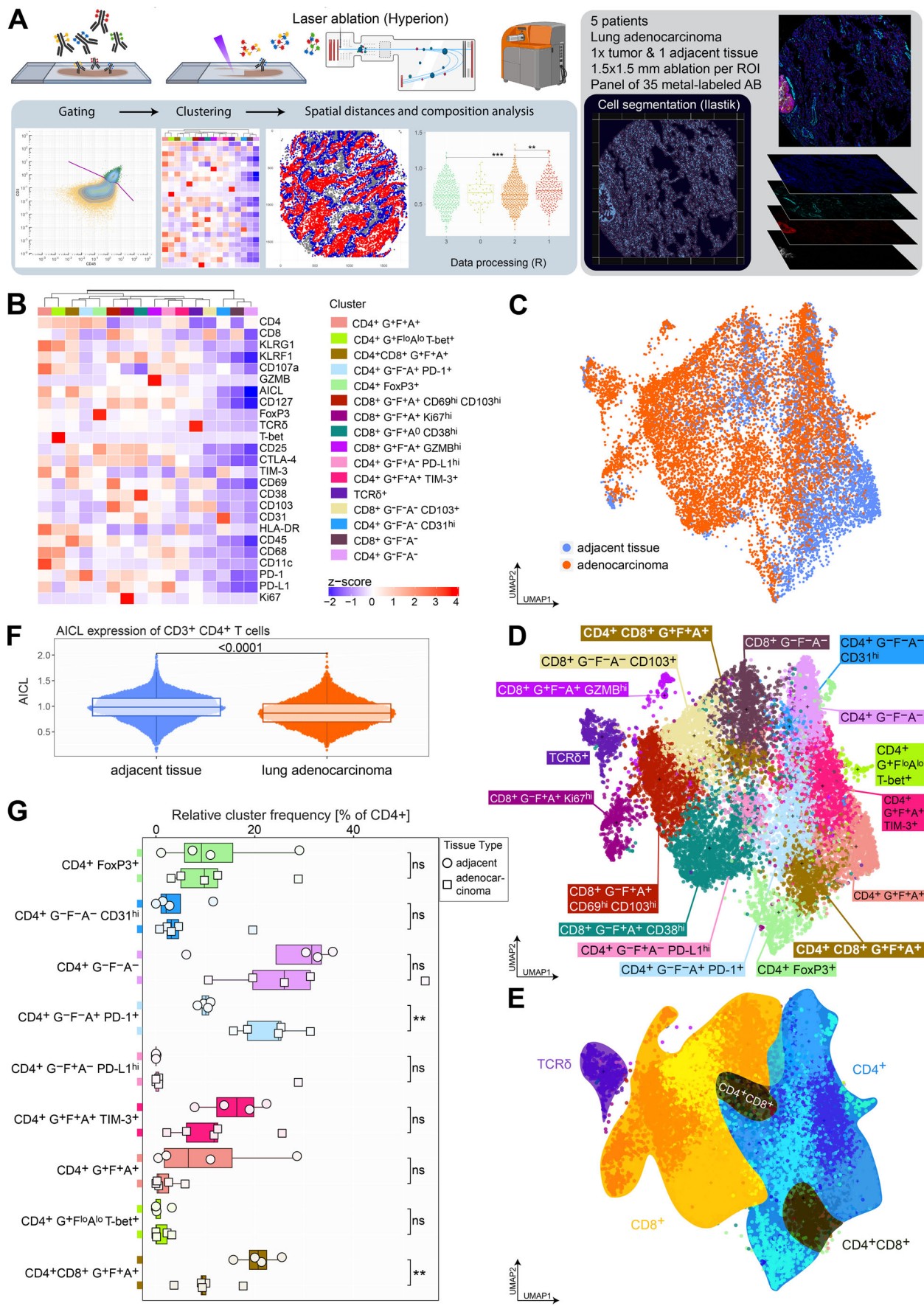

Figure 4.   IMC analysis revealed reduced AICL expression and decreased proportions of activated CD4$^+$ AICL$^+$ T cells in lung adenocarcinoma tissue.

(A) Schematic illustration of the imaging mass cytometry method and data processing workflow. (B) Heatmap displaying z-scored marker expression of the identified T cell clusters. Clusters were annotated by the expression lineage markers CD4, CD8, and TCR delta as well as their expression of KLRG1 (G), KLRF1 (F) and AICL (A), along with activation and exhaustion markers. Markers are colored by their respective z-scored values. (C–E) UMAP embeddings of gated T cells, color-coded by (C) tissue origin (lung adenocarcinoma versus adjacent tissue), separated into annotated T cell clusters, (D) T cell lineages, (E) artificially marking CD4$^+$ (blue), CD8$^+$ (yellow), γδ T cells (Tγδ, violet), and the CD4$^+$CD8$^+$ double-positive (black) cluster. (F) Bulk expression of arcsinh-transformed raw AICL by all CD4$^+$ T cells split by tissue (lung adenocarcinoma versus adjacent tissue, $n = 5$, biological replicates). Statistical analysis was performed with a two-tailed $t$-test (7488 CD4$^+$ T cells), $p$ value ****$p < 0.0001$. Box plots show the median (center line) and interquartile range (box limits 25th to 75th percentile, 1x IQR), whiskers extend 1.5x IQR from the box. (G) Patient-wise cluster frequencies of annotated CD4$^+$ clusters relative to all CD4$^+$ cells found in adjacent (circles) and lung adenocarcinoma tissue (squares). Two-tailed $t$-test, adjacent vs. adenocarcinoma tissue: CD4$^+$ G$^-$F$^-$A$^+$ PD1$^+$ **$p = 0.0075$; CD4$^+$CD8$^+$ G$^+$F$^+$A$^+$ **$p = 0.0096$; ns $p > 0.05$ ($n = 5$, paired tumor adjacent and adenocarcinoma samples, biological replicates). Box plots show the median (center line) and interquartile range (box limits 25th to 75th percentile, 1x IQR), whiskers extend to the farthest values but no more than 1.5x IQR from the box.

contrast, the distance to all other CD8$^+$ T cell clusters were significantly greater.

Consequently, we observed a significantly higher number of split CD8$^+$ T cells in direct proximity (within 20 μm) to split CD4$^+$ T cells (Fig. 5F, middle). While not statistically significant, a similar trend was observed when averaging the number of neighboring cells across all split CD4$^+$ T cells in each ROI (Fig. 5F, right).

In stark contrast, the silent CD4$^+$ PD-1$^+$ cluster, which was more prevalent in lung adenocarcinoma tissue, did not preferentially accumulate near any CD8$^+$ T cell cluster, including the two activated CD8$^+$ T cell clusters, as all median distances were above 800px (Fig. 5G).

AICL is expressed by multiple cell types, including monocytes, macrophages, neutrophils and NK cells (Hamann et al, 1997). To identify potential sources of AICL apart from the split CD4$^+$ G$^+$F$^+$A$^+$ T cell cluster within the tissue microenvironment, we performed unsupervised clustering on the CD19$^-$ CD3$^-$ CD45$^+$ non-B/non-T immune compartment (Gate 3). Cluster annotation was based on the expression patterns of lineage-relevant markers available in our IMC panel: clusters with high CD25 and CD127 but lacking CD68 and CD11c were annotated as ILC- or NK-like populations, while clusters expressing CD68, CD11c, and/or HLA-DR and lacking CD25 and CD127 were annotated as monocyte-, macrophage-, or dendritic cell-like populations. A small number of clusters could not be reliably assigned due to limited marker resolution and were therefore designated as "other" (Fig. EV2C). The UMAP embedding showed these annotated groups as well-separated phenotypic islands, confirming the validity of the clustering and annotation approach (Appendix Fig. S2B).

To evaluate whether any of these non-T immune populations might serve as AICL-expressing interaction partners for CD8$^+$ KLRF1$^+$ T cells, we quantified both AICL expression and median spatial distances for each Gate 3 cluster relative to the split CD8$^+$ G$^+$F$^+$A$^+$ cluster (Fig. EV2G). This analysis demonstrated that none of the non-T immune clusters expressed AICL at levels comparable to the split CD4$^+$ G$^+$F$^+$A$^+$ T cell cluster, and all were positioned considerably farther away from the split CD8$^+$ G$^+$F$^+$A$^+$ T cells. Notably, the myeloid clusters closest to CD8$^+$ KLRF1$^+$ T cells (clusters 311 and 319) displayed the lowest AICL expression of all Gate 3 clusters. Cluster 321 did not co-localize on the same ROI with any of the split CD8$^+$ G$^+$F$^+$A$^+$ T cells and is therefore missing in the distance analysis. Together, these findings indicate that CD4$^+$ AICL$^+$ T cells are the predominant AICL-expressing population in spatial proximity to CD8$^+$ KLRF1$^+$ T cells in non-tumor lung tissue.

Our findings suggest a spatially organized interaction between CD4$^+$ AICL$^+$ T cells and activated CD8$^+$ KLRF1$^+$ T cells in non-affected adjacent tissue, which seems to be disrupted in lung adenocarcinoma. These observations highlight a potential loss of functional CD4$^+$–CD8$^+$ T cell interactions in the tumor microenvironment, which may contribute to immune dysfunction and CD8$^+$ T cell exhaustion in lung adenocarcinoma.

## Reduced abundance and IFN/TNF expression of KLRF1$^+$ Tpex-like cells in lung adenocarcinoma

IMC analysis revealed increased abundance of and close interaction between the activated split CD4$^+$ AICL$^+$ and split CD8$^+$ KLRF1$^+$ T cell clusters in non-affected adjacent tissue. Next, we were interested in whether this is linked to a Tpex-like transcriptional phenotype and increased *IFN-γ* and *TNF* expression. A portion of the tissue samples used for imaging mass cytometry had previously undergone scRNA-seq analysis, which was published by Bischoff et al (Bischoff et al, 2021). To leverage these data for our study, we performed targeted amplification of the cDNA libraries using IFNG-specific primers with subsequent re-sequencing.

NK and cytotoxic CD8$^+$ T cells share a similar transcriptomic profile (Narni-Mancinelli et al, 2011). Consequently, the UMAP embedding of the annotated NK/T cell subset (Bischoff et al, 2021) exhibited overlapping expressions of markers typically attributed to NK cells (NKG7), γδT cells (*TRDC*) or cytotoxic T cells (*TRAC, CD3E, CD8*) (Fig. EV3A). Thus, clustering the NK/T cell subset yielded improper separation of NK cells from the cytotoxic CD8$^+$ T cells (Fig. EV3B) with both cell types expressing *KLRF1* as described in literature (Vitale et al, 2001; Kuttruff et al, 2009). Since the cluster was predominantly composed of T cells expressing *KLRF1* to a lower extent as compared to the NK cells (Fig. EV3C), improper separation of both cell types would have severely hampered the analysis of KLRF1$^+$ T cells.

Due to the lack of NK cell-exclusive genes (Fig. EV3A), we refrained from gating out the NK cells but instead gated on T cells by selecting *CD3E$^+$* and *TRAC$^+$* cells that additionally expressed either *CD3D* or *CD3G*. The gated T cells were clustered (Figs. 6A and EV3D) and classified into CD4$^+$ or CD8$^+$ T cells (Figs. 6B and EV3E,F). This stringent gating strategy effectively removed all NK and γδ T cells from the scRNA-seq data (Fig. 6C).

We next aimed to characterize the CD8$^+$ KLRF1$^+$ T cell clusters. Given that our in vitro data demonstrated the ability of CD8$^+$ KLRF1$^+$ T cells to produce IFN-γ and TNF, we hypothesized that these cells resemble early dysfunctional Tpex rather than Tex cells

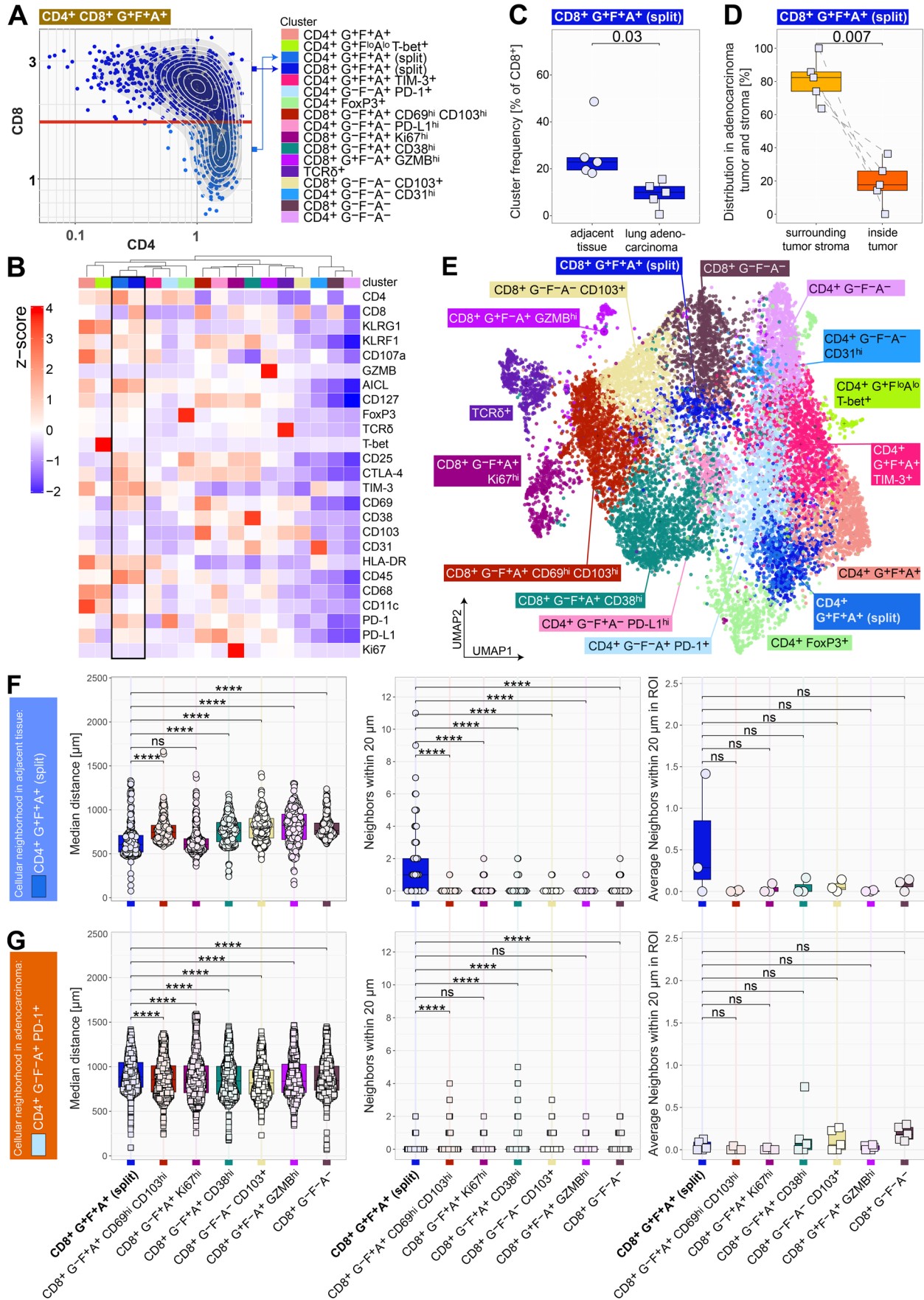

Figure 5. Spatial analysis of CD4$^+$ and CD8$^+$ T cell subsets in lung adenocarcinoma and adjacent tissue.

(A) Manual separation of the CD4$^+$CD8$^+$ double-positive cluster (see Fig. 4B, brown cluster) by gating on CD8$^+$ T cells (red line) and designating the clusters as "split". (B, E) Recalculated the heatmap and UMAP embedding with both split clusters. (C) Patient-wise cluster frequency of split CD8$^+$ KLRG1$^+$ KLRF1$^+$ AICL$^+$ (G$^+$F$^+$A$^+$) cluster relative to all CD8$^+$ cells found in adjacent (circles) or tumor tissue (squares). $n = 5$, biological replicates. Statistical analysis was performed with a two-tailed $t$-test. Exact $p$ value is noted in the plot. Box plots show the median (center line) and interquartile range (box limits 25th to 75th percentile, 1x IQR), whiskers extend to the farthest values but no more than 1.5x IQR from the box. (D) Proportional distribution of the split CD8$^+$ KLRG1$^+$ KLRF1$^+$ AICL$^+$ (G$^+$F$^+$A$^+$) cluster within lung adenocarcinoma tissues ($n = 5$, biological replicates). Fractional composition in the stroma surrounding the tumor (orange) and inside the tumor (red). Statistical analysis was performed with a paired two-tailed $t$-test. **$p = 0.0068$. Box plots show the median (center line) and interquartile range (box limits 25th to 75th percentile, 1x IQR), whiskers extend to the farthest values but no more than 1.5x IQR from the box. (F, G) Neighborhood analysis of cells belonging to the split CD4$^+$ G$^+$F$^+$A$^+$ cluster (panel E, $n = 5$ biological replicates, 272 split CD4$^+$ AICL$^+$ T cells found on 3 ROIs) and the CD4$^+$ KLRG1$^-$ KLRF1$^-$ AICL$^+$ PD-1$^+$ T cell cluster (panel F, $n = 5$ biological replicates, 806 CD4$^+$ PD-1$^+$ T cells found on five ROIs). Box plots show the median (center line) and interquartile range (box limits 25th to 75th percentile, 1x IQR), whiskers extend to the farthest values but no more than 1.5x IQR from the box. Left: Median distance of each cell to CD8$^+$ T cell clusters; Middle: Number of CD8$^+$ T cells within 20-μm distance to each cell; Right: Number of CD8$^+$ T cells within 20-μm distance, averaged over all cells in each ROI. For panels (E, F), the Wilcoxon rank-sum test with Benjamini & Hochberg adjustment. ****$p < 0.0001$.

with an imprinted exhaustion phenotype. To test this, we analyzed marker gene expression in CD8$^+$ T cell clusters using additional Tpex- and Tex-specific markers (highlighted in blue and red in Fig. 6D). Indeed, CD8$^+$ KLRF1$^+$ clusters 4 and 14 exhibited a transcriptional profile characteristic of Tpex, marked by presence of *TCF7*, *IL7R*, *CXCR5*, and *TBX21* but absence of *TOX*, *CTLA4*, *HAVCR2* and *GZMB* expression. Since IMC data suggested that cytotoxic CD8$^+$ KLRF1$^+$ T cells reside in a reservoir outside the tumor (Fig. 5C), we further confirmed that KLRF1$^+$ Tpex-like clusters were significantly reduced in lung adenocarcinoma tissue samples (Fig. 6E).

Given that IFN-γ and TNF production by KLRF1$^+$ Tpex-like T cells was associated with the presence of CD4$^+$ T cells, we further hypothesized that loss of KLRF1$^+$ Tpex-like cells could negatively affect cytokine expression in the tumor. Indeed, analyzing *TNF* and *IFN-γ* expression levels in both KLRF1$^+$ clusters 4 and 14 showed notably dampened levels in lung adenocarcinoma tissue (Fig. 6F).

To validate our findings and thus the Tpex-like phenotype and enhanced cytokine expression potential of KLRF1$^+$ T cells in tumor-adjacent tissue areas, we leveraged publicly available databases. For this, we analyzed two lung cancer datasets: a smaller dataset by Laughney et al (Laughney et al, 2020) and a larger, multi-center dataset from the Human Tumor Atlas Network (HTAN) MSK (Chan et al, 2021) (Fig. EV4). Both datasets were analyzed using the same gating strategy as described above, focusing on CD8$^+$ KLRF1$^+$ T cell clusters (Fig. EV4A,B). The results again showed that KLRF1$^+$ T cell clusters predominantly expressed *TCF-1* while lacking *TOX*, *CTLA4*, and *HAVCR2*, further reinforcing their Tpex-like phenotype (Fig. EV4C,D). In all three datasets, *KLRF1* also identified effector T cells, as evidenced by high *TBX21* expression in CD8$^+$ KLRF1$^+$ T cell clusters (Fig. EV4C,D).

At the same time, we observed a significant loss of CD8$^+$ KLRF1$^+$ T cells in tumor tissues across all three datasets (Fig. EV4E). However, in the smaller Laughney dataset, CD8$^+$ KLRF1$^+$ T cell clusters were primarily detected in adjacent non-tumor samples, limiting statistical analysis of *IFN-γ* and *TNF* expression due to an insufficient number of carcinoma samples containing CD8$^+$ KLRF1$^+$ T cell clusters ($n = 1$, Fig. EV4G). Nonetheless, the HTAN MSK dataset confirmed the impact of the tumor microenvironment on cytokine production, as KLRF1-expressing CD8$^+$ T cell clusters in carcinoma tissue samples also exhibited impaired cytokine production (Fig. EV4H).

Taken together, the loss of CD8$^+$ KLRF1$^+$ T cells observed in IMC data was also confirmed by scRNA-seq analysis of the same cohort. Tpex- and Tex-specific marker expression indicated that KLRF1$^+$ T cell clusters exhibited a Tpex-like transcriptional profile rather than a Tex phenotype. Since KLRF1$^+$ Tpex-like cells and AICL$^+$ T helper cells were significantly reduced in carcinoma samples in IMC data, we examined whether this lost interaction impacted cytokine production in the remaining Tpex cells. Indeed, *IFN-γ* and *TNF* expression in CD8$^+$ KLRF1$^+$ T cell clusters was significantly reduced in carcinoma samples. This finding was further validated in two independent publicly available datasets, where CD8$^+$ KLRF1$^+$ T cells were similarly diminished, along with a corresponding decrease in cytokine expression in carcinoma tissue samples.

## Epigenetic profiling defines a Tpex-like chromatin state in CD8$^+$ KLRF1$^+$ T cells

To further refine the differentiation state of KLRF1-expressing CD8$^+$ T cells beyond transcriptional signatures, we next examined their chromatin-accessibility landscape. As epigenetic programs are more stable than gene expression and provide a robust framework to distinguish progenitor-exhausted from terminally exhausted T cell states, we reanalyzed a multimodal dataset integrating scRNA-seq, ATAC-seq, and antibody-derived tags (ADT) (Thomson et al, 2023). This approach allowed us to assess whether the transcriptionally defined Tpex-like features of CD8$^+$ KLRF1$^+$ T cells are supported by a corresponding progenitor-like epigenetic profile. To accurately define the CD8$^+$ T cell compartment, we did not rely solely on the automated annotations provided in the original dataset, as these occasionally assigned CD4 identity to cells with a clear CD8 transcriptional signature. Instead, we applied an extended version of the gating strategy used consistently throughout our manuscript. We first identified T cells, followed by gating on CD8$^+$ T cells with exclusion of γδ T-cell markers (Fig. EV3 and method part). Applying this stringent gating yielded a CD8$^+$ T cell subset that included cells annotated as CD4$^+$ by the automated method but whose expression pattern matched a CD8$^+$ phenotype (Fig. EV5A). In turn, this ensured that the full CD8$^+$ T cell population was retained and that no CD8$^+$ T cells were inadvertently omitted from our multimodal analyses.

The gated CD8$^+$ T cells were clustered using a weighted-nearest-neighbor framework that integrates scRNA-seq PCA and ATAC LSI components. This analysis showed that CD8$^+$ KLRF1$^+$ T cells, marked by strong *KLRF1* expression, clustered almost exclusively in a single population (cluster 7, Fig. EV5B,C), whereas the most

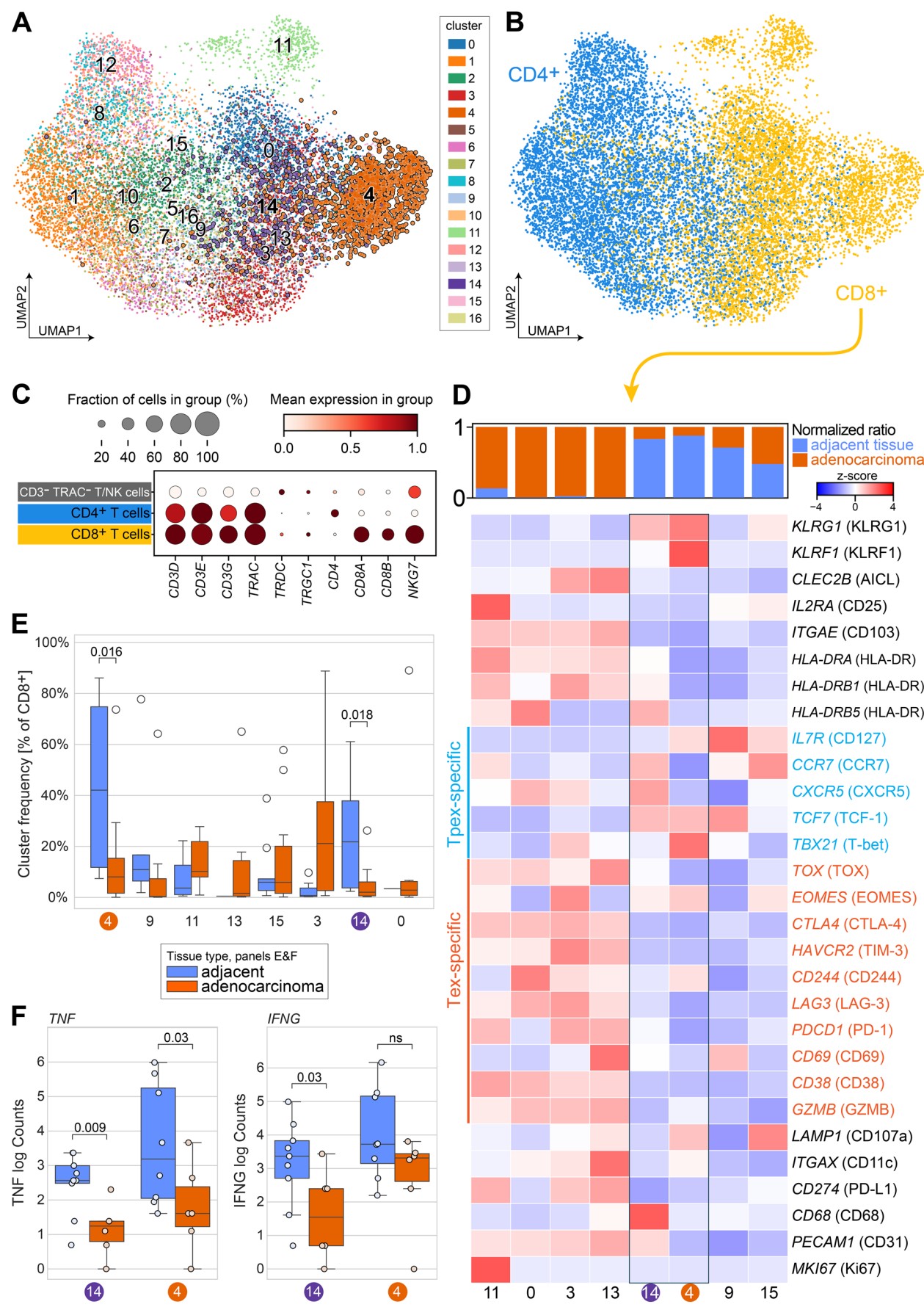

**Figure 6. Tpex-like CD8⁺ KLRF1⁺ T cells are reduced and show decreased cytokine expression in lung adenocarcinoma tissue.**

(A) UMAP embedding of gated T cells, color-coded by Leiden clusters. KLRF1⁺ clusters 14 and 4 are highlighted. (B) UMAP embedding with clusters assigned based on CD4 and CD8 expression (see Fig. EV3E) and color-coded by CD4⁺ and CD8⁺ T cells. (C) Dot plot representation of CD4⁺, CD8⁺ T cells, and remaining cells that were gated out (gray, see Fig. EV3F). (D) Marker expression heatmap of the CD8⁺ clusters along with the respective distribution among lung adenocarcinoma and adjacent non-tumor samples (top). Tpex- and Tex-specific markers are highlighted in blue and red, respectively. Clusters 14 and 4 were *KLRF1⁺ TCF-1⁺ TOX⁻*. (E) Changes in the composition of CD8⁺ clusters. Statistical analysis was performed with a one-tailed Wilcoxon rank-sum test for the *KLRF1⁺* clusters 4 and 14. Exact *p* values are noted in the plot, other clusters were not tested ($n = 12$, biological replicates). Box plots show the median (center line) and interquartile range (box limits 25th to 75th percentile, 1x IQR), whiskers extend to the farthest values but no more than 1.5x IQR from the box. Only data points beyond the whiskers are drawn. (F) *TNF* (left) and *IFN-γ* (right) expression of *KLRF1⁺* T cell clusters 14 and 4 in lung adenocarcinoma and adjacent non-tumor tissue. Statistical analysis was performed with a one-tailed Wilcoxon rank-sum test ($n = 12$, biological replicates). ns $p > 0.05$. Box plots show the median (center line) and interquartile range (box limits 25th to 75th percentile, 1x IQR), whiskers extend to the farthest values but no more than 1.5x IQR from the box. All data points are drawn.

terminally differentiated KLRF1⁻ cells fell into cluster 3. Both clusters together with cluster 2 were distinct from naïve and central-memory CD8⁺ T cells.

We next used the multimodal nature of the dataset to characterize the differentiation state of clusters 2, 7, and 3. ADT signals (e.g., CCR7, CD27, and CD45RA) and a heatmap of differentially expressed genes revealed that clusters 7 and 3 represent the most differentiated effector memory cells (Fig. EV5D). Cluster 3 showed higher proportions and higher expression of CD45RA and lower expression of CD27, consistent with a T_EMRA-like profile, whereas cluster 7 exhibited a more memory-like phenotype. At the transcriptional level, both clusters expressed effector genes such as *NKG7*, *GZMH*, *GZMB*, and *PRF1*, whereas cluster 2 displayed lower effector-gene expression. To further distinguish potential Tpex-like from Tex-like states, we compiled a curated list of human Tex and Tpex marker genes from the literature (Brummelman et al, 2018; Li et al, 2018; Sade-Feldman et al, 2018; Thommen et al, 2018; Utzschneider et al, 2018; Khan et al, 2019; Miller et al, 2019; Scott et al, 2019; Yost et al, 2019; Beltra et al, 2020; Galletti et al, 2020; Sekine et al, 2020; Gabriel et al, 2021; Zheng et al, 2021; Sun et al, 2023) (Appendix Table S8) and generated corresponding module scores. UMAP visualization demonstrated that cells in cluster 7 exhibited higher Tpex scores, whereas cluster 3 cells showed elevated Tex scores (Fig. EV5E). Statistical comparison confirmed significantly higher Tpex scores in cluster 7 and significantly higher Tex scores in cluster 3 (Fig. EV5F). Finally, we examined chromatin accessibility using ATAC gene-score comparisons between clusters 7 and 3 (Fig. EV5G). The KLRF1⁺ cluster showed significantly higher accessibility at loci associated with memory and progenitor states, including *IL7R*, *CXCR5*, and *TCF7*, while the exhaustion-associated *ENTPD1* locus was markedly more accessible in the KLRF1⁻ cluster. These epigenetic patterns reinforce the conclusion that CD8⁺ KLRF1⁺ T cells retain memory-associated, progenitor-like features.

Taken together, the transcriptomic profiles, protein-marker signals, Tex/Tpex gene-module scoring, and ATAC accessibility differences collectively support the interpretation that CD8⁺ KLRF1⁺ T cells exhibit Tpex-like rather than Tex-like characteristics.

## Discussion

In this study, we identify a previously unrecognized CD4⁺–CD8⁺ T cell communication axis that supports cytokine production in differentiated memory or Tpex CD8⁺ T cells. Building on our

earlier findings in CD4⁺ T cells, we demonstrate that progressive acquisition of killer-like receptors, particularly KLRF1, also marks the functional differentiation of human CD8⁺ T cells. CD8⁺ KLRF1⁺ T cells, which transcriptionally and epigenetically resemble Tpex cells, maintain IFN-γ and TNF production, with KLRF1 signaling contributing to a measurable but modest enhancement of cytokine output.

We show that activated CD4⁺ T cells can provide the KLRF1 ligand AICL and thereby contribute to enhanced cytokine production. Profiling of lung adenocarcinoma and adjacent tissue revealed that these interactions occur primarily in non-tumor regions, where AICL-expressing CD4⁺ and CD8⁺ KLRF1⁺ T cells are enriched. In contrast, both populations are diminished within the tumor microenvironment.

Cytokines are considered potentially useful signaling molecules for a sustained cancer-immunity cycle and to trigger tumor-directed responses by T cells through the induction of integrins (Harjunpää et al, 2019). Even though proportions of tumor-reactive T cells are highly variable among cancer types and amongst patients, high densities of tissue-resident CD8⁺ memory and cytotoxic T cells are generally associated with better response to ICB (Fridman et al, 2012), a longer disease-free survival and/or overall survival (Fridman et al, 2012). It is worth noting that tumor infiltration per se does not necessarily correlate with antitumor reactivity, since only a small fraction of tumor-infiltrating T cells are tumor-reactive (Scheper et al, 2019) or functional. As described before, tumor-reactive CD8⁺ T cells suffer from persistent antigen load inside the tumor and are composed of at least two populations termed Tpex and Tex (Kallies et al, 2020). Whereas Tex cells have adopted a terminally exhausted phenotype, Tpex are responsive to extrinsic signals such as blockade of co-inhibitory receptors (Im et al, 2016; Miller et al, 2019; Siddiqui et al, 2019). Since IFN-γ-related genes correlate with responsiveness to, e.g., anti-PD-1 therapy (Ayers et al, 2017; Higgs et al, 2018; Karachaliou et al, 2018), Tpex are assumed to be important in successful immunotherapeutic intervention (Miller et al, 2019) and have therefore gained substantial attention. Thus, identifying ligand–receptor pairs and the corresponding signal-providing cells within the tumor microenvironment that regulate cytokine production by Tpex cells is of critical importance. This, in turn, would open new venues for targeted therapeutic strategies to maximize cytokine production in tumors.

We recently published a set of four markers—KLRG1, KLRB1, GPR56, and KLRF1—that progressively emerge during CD4⁺ T helper cell differentiation (Truong et al, 2019). In the present study, we examined the expression of these markers in CD8⁺ T cells to

identify those characteristics of late-differentiated cells that promote the maintenance of cytokine production.

In vitro activated CD8$^+$ T cells collected from PBMC first upregulate KLRG1, followed by KLRB1. Both markers are expressed at an early memory stage and have been shown to positively correlate with overall survival, disease progression or response to immunotherapy (Yang et al, 2021; Cheng et al, 2022). Subsequent upregulation of GPR56 and KLRF1 marks the late-differentiated subsets $T_{EM}$ and $T_{EMRA}$ (Fig. 1A). Interestingly, GPR56 has been shown to attenuate cytotoxicity in NK cells and is therefore considered an immune checkpoint for CD8$^+$ T cells (Bilemjian et al, 2022). Our findings support a negative role for GPR56 in cytokine production by late-differentiated CD8$^+$ T cells (Figs. 1D and EV1B), consistent with previous reports linking GPR56 expression to terminal differentiation and T cell exhaustion (Kared et al, 2024). Notably, in lung cancer, the proportion of GPR56$^+$ lymphocytes increases during disease progression (Liu et al, 2023). In contrast, KLRF1 is known to positively regulate NK and T cell functionality (Welte et al, 2006; Kuttruff et al, 2009). This suggested that the AICL–KLRF1 axis functions as a signaling pathway supporting the maintenance of cytokine production in late-differentiated CD8$^+$ T cells and might therefore be characteristic of Tpex cells. Indeed, in line with previous findings, blocking KLRF1 caused a reduction of IFN-γ and TNF by CD8$^+$ T cells (Fig. 2B). This was only observed in the presence of CD4$^+$ T helper cells (Fig. 2C), indicating that they must express the ligand for KLRF1. AICL, the currently only known ligand for KLRF1, was detected in T and B lymphocytes and myeloid cells, including monocytes and granulocytes, but has been primarily investigated on NK cells (Hamann et al, 1997) and has been shown to induce NK cell-mediated cytolytic activity (Vitale et al, 2001; Welte et al, 2006). In fact, we detected a transient upregulation of AICL during anti-CD3 and anti-CD28 antibody-mediated stimulation of CD4$^+$ T cells (Fig. 3B). Further phenotyping revealed that AICL upregulation was restricted to KLRF1-expressing CD4$^+$ T cells (Fig. 3C), supporting co-regulation of *CLEC2B* and *KLRF1*, as described for NK cells (Bartel et al, 2013). Consistently, AICL expression positively correlates in certain cancers with clinical outcome and is discussed as a prognostic marker (Guo et al, 2021).

In contrast, little is known about the role of KLRF1 or the relevance of AICL–KLRF1-mediated interactions between CD4$^+$ and CD8$^+$ T cells within the tumor microenvironment. To investigate this, we used IMC and studied the spatial distribution of AICL and KLRF1 expressing CD4$^+$ and CD8$^+$ T cells in lung adenocarcinoma and adjacent non-affected tissue samples. Notably, we observed that a substantial fraction of AICL$^+$ and KLRF1$^+$ CD4$^+$ and CD8$^+$ T cells, which shared a similar phenotype, clustered together as AICL$^+$ KLRF1$^+$ double-positive T cells (Fig. 4B). These cell populations were significantly less abundant in lung adenocarcinoma tissue samples compared to adjacent non-affected samples (Figs. 4G and 5C). Subsequent gating-based separation into CD4$^+$ and CD8$^+$ T cell subsets revealed that these two populations not only established direct contacts in non-tumor tissue (Fig. 5F), but that CD4$^+$ AICL$^+$ T helper cells were in closer proximity to CD8$^+$ KLRF1$^+$ T cells than to any other CD8$^+$ T cell subset.

Given the loss of close interactions and the reduction of KLRF1$^+$ T cells in adenocarcinoma tissue, we hypothesized that these cells

may resemble a Tpex-like phenotype (Utzschneider et al, 2013; Miller et al, 2019). To further characterize CD8$^+$ KLRF1$^+$ T cells and gain insights into their cytokine profile, we reanalyzed our previously published scRNA-seq dataset from the same patient cohort (Bischoff et al, 2021). This analysis confirmed a contraction of KLRF1$^+$ T cells within tumor tissue and revealed that KLRF1$^+$ T cell clusters were indeed *TCF-1$^+$*, *TOX$^-$*, *TIM-3$^-$*, and *GZMB$^-$*—an expression pattern consistent with a Tpex phenotype. Furthermore, CD8$^+$ KLRF1$^+$ T cells from adjacent non-tumor tissue exhibited significantly higher expression of *IFNG* and *TNF* compared to their counterparts in adenocarcinoma samples. Importantly, we could validate our findings by reanalyzing two additional publicly available datasets (Laughney et al, 2020; Chan et al, 2021). These findings position KLRF1 as a novel marker of Tpex cells with potential diagnostic and therapeutic relevance, which needs further investigation in future studies.

Recent studies have highlighted that exhausted CD8$^+$ T cells exhibit greater functional heterogeneity than previously appreciated. Although Tex cells are often characterized by reduced effector function, several reports demonstrate that specific Tex subsets can retain or regain expression of IFN-γ, TNF, granzyme B, or perforin (Minnie et al, 2024). Conversely, Tpex cells remain the major source of IL-2 and polyfunctional cytokines, while terminal Tex populations typically show dominant IFN-γ expression but diminished TNF production (Miller et al, 2019; Beltra et al, 2020; Wang et al, 2022). This reinforces that exhaustion states form a functional continuum, not rigid categories. Consistent with this continuum, our scRNA-seq data show that KLRF1$^+$ Tpex-like cells are enriched for TNF expression, whereas KLRF1$^-$ terminal clusters preferentially express late cytotoxic markers. Importantly, multi-modal scRNA/ATAC-seq analyses further revealed that CD8$^+$ KLRF1$^+$ T cells maintain accessibility at *IL7R*, *CXCR5*, and *TCF7* loci—hallmarks of progenitor-like states—while exhaustion-associated loci such as *ENTPD1*, encoding for CD39, are more accessible in the KLRF1$^-$ clusters. Together, these transcriptional and epigenetic signatures align with a memory-like, Tpex-like phenotype and support the notion that KLRF1 marks a subset of late-differentiated CD8$^+$ T cells that retain functional potential despite chronic antigen stimulation.

Linking the IMC and scRNA-seq data through the identification of KLRF1-expressing CD8$^+$ T cells allowed us to draw conclusions about the spatial dynamics of Tpex versus Tex CD8$^+$ T cell populations. Our observation that CD8$^+$ KLRF1$^+$ Tpex-like cells are more frequently found in the surrounding tumor stroma compared to the tumor core (Fig. 5D) is consistent with recent reports (Huang et al, 2024; Tanoue et al, 2024).

The findings presented here further underscore the importance of CD4$^+$ T cell help in preventing CD8$^+$ T cell dysfunction under conditions of chronic antigen stimulation (Church et al, 2014; Busselaar et al, 2020). While previous studies have primarily focused on the role of CD4$^+$ T helper cells in instructing dendritic cells (Busselaar et al, 2020; Gressier et al, 2023), our data highlight a direct interaction between CD4$^+$ and CD8$^+$ T cells.

In summary, our findings reveal a previously unrecognized CD4$^+$–CD8$^+$ T cell communication axis mediated by the AICL–KLRF1 interaction that contributes to the maintenance of cytokine competence in Tpex-like CD8$^+$ T cells. This highlights a novel mechanism of CD4$^+$ T cell help beyond dendritic cell

instruction and identifies KLRF1 as a promising marker with potential diagnostic and therapeutic relevance in the context of tumor immunity.

# Methods

### Reagents and tools table

| Reagent/resource | Reference or source | Identifier or catalog number |
|---|---|---|
| **Experimental models** | | |
| **Recombinant DNA** | | |
| **Antibodies** | | |
| Flow/Sorting stains: | This study | See also Appendix Tables S2–4 |
| Purified anti-human CD3 | BD Biosciences | UCHT1 |
| Purified anti-human CD28 | BD Biosciences | CD28.2 |
| Ultra-LEAF™ purified anti-human IgG1 isotype control | BioLegend | QA16A12 |
| Ultra-LEAF™ purified anti-human NKp80 (KLRF1) | BioLegend | 5D12 |
| PerCP-CD8a | BioLegend | RPA-T8 |
| PE-KLRB1 (CD161) | Miltenyi Biotec | 191B8 |
| BV421-KLRB1 (CD161) | BioLegend | HP-3G10 |
| PE-Vio770-KLRF1 (NKp80) | Miltenyi Biotec | 4A4.D10 |
| APC-KLRG1 | Miltenyi Biotec | REA261 |
| BV785-CD3 | BioLegend | OKT3 |
| VioBright FITC-GPR56 | Miltenyi Biotec | REA467 |
| PE-Dazzle594-IFN-γ | BioLegend | 4S.B3 |
| BV510- IFN-γ | BioLegend | 4S.B3 |
| AF700-TNF | BioLegend | Mab11 |
| PE-AICL | LSBio | - |
| IMC panel | This study | See also Appendix Table S5 |
| CD38 | Abcam | ab226034 |
| CD19 | Invitrogen | 14-0194-82 |
| Granzyme B | Abcam | ab219803 |
| KLRF1 | Abcam | ab198928 |
| β-Actin | Santa Cruz | sc-47778 |
| HLA-DR | Invitrogen | 14-9956-82 |
| GPR56 | Santa Cruz | sc-390192 |
| CD107a | Cell Signalling | 9091BF |
| p53 | Cell Signalling | 48818 |
| PD-L1 | Fluidigm | 3150031D |
| CD31 | Fluidigm | 3151025D |
| CD45 | Fluidigm | 3152018D |
| CD103 | Abcam | ab271889 |
| Tim-3 | Fluidigm | 3154024D |
| KLRG1 | R&D Systems | MAB70293 |
| CD4 | Abcam | ab181724 |

| Reagent/resource | Reference or source | Identifier or catalog number |
|---|---|---|
| CD324 | Fluidigm | 3158029D |
| CD68 | Invitrogen | 17-0688-82 |
| T-Bet | Cell Signalling | 13232 |
| CA19-9 | Abcam | ab3982 |
| CD8 | BioLegend | 372902 |
| AICL | Abcam | ab235626 |
| Ki67 | BD Biosciences | 556003 |
| PD-1 | Abcam | 186928 |
| CTLA-4 | Abcam | ab237712 |
| CD11c | Abcam | ab216655 |
| CD127 | Fluidigm | 3168026D |
| TCR delta | Santa Cruz | sc-100289 |
| CD3 | Cell Signalling | 85061BF |
| CD69 | Abcam | ab234512 |
| CK18 | Abcam | ab7797 |
| FoxP3 | Invitrogen | 14-4777-82 |
| aSMA | Abcam | ab240654 |
| CD25 | Fluidigm | 3175036D |
| **Oligonucleotides and other sequence-based reagents** | | |
| **Chemicals, enzymes and other reagents** | | |
| DPBS | Gibco | 14190-169 |
| Pancoll | PAN-Biotech GmbH | |
| FBS (Lot 605c) | Sigma-Aldrich | discont. |
| FBS superior | Sigma-Aldrich | SO615-500ML |
| DMSO | Sigma-Aldrich | 276855-100 ML |
| RPMI 1640 | VLE, Biochrom | discont. |
| RPMI 1640 | PAN-Biotech GmbH | P04-18525 |
| Penicillin/Streptomycin | Biochrom | discont. |
| Penicillin/Streptomycin | Gibco | 15140-122 |
| Pan T cell Isolation Kit | Miltenyi Biotec | 130-096-535 |
| CD8+ T Cell Isolation Kit | Miltenyi Biotec | 130-096-495 |
| Brefeldin A | Sigma-Aldrich | B7651-5MG |
| Zombie UV Fixable Viability Kit | BioLegend | 423107 |
| Beriglobin | CSL Behring GmbH | discont. |
| Cytofix/Cytoperm | BD Biosciences | 51-2090KZ |
| Perm/Wash | BioLegend | 421002 |
| AO/PI Assay_CellDrop FL | DeNovix | CD-AO-PI-1-5-C |
| Sodium Azide | Sigma-Aldrich | S2002-100G |
| Xylene | VWR | 28973.363 |
| Ethanol | Carl Roth | K928.4 |
| Tris-HCl | Carl Roth | 9090.2 |
| EDTA | Sigma-Aldrich | E5134-100G |
| Tween-20 | Carl Roth | 9127.1 |

| Reagent/resource | Reference or source | Identifier or catalog number |
|---|---|---|
| BSA Fraction V | Serva | 11930.03 |
| Goat serum | Invitrogen | 100000 C |
| **Software** | | |
| FlowJo | FlowJo LLC | V10 |
| MCD Viewer | Fluidigm | 1.0.560.6 |
| CellProfiler | Carpenter et al, 2006 | 4.2.6 |
| Ilastik | Berg et al, 2019 | 1.4.0 |
| ImageJ | Schneider et al, 2012 | 1.54 f |
| R | | 4.5 |
| **Other** | | |
| BD LSRFortessa flow cytometer | BD Biosciences | |
| BD AriaFusion | BD Biosciences | |
| Nanodrop 1000 | Thermo Fisher Scientific | |
| Hyperion Tissue Imager | Fluidigm | |
| Helios Mass Cytometer | Fluidigm | |
| CellDrop FL | DeNovix | |

## In vitro studies

Fresh blood was collected from 11 voluntary, healthy blood donors and processed immediately. Additionally, frozen peripheral blood mononuclear cells (PBMC) were used. They were obtained from five anonymous, healthy blood donors (DRK LB Berliner Rotes Kreuz e.V.) and processed within 2 h. Sample collection and preparation was done with permission from the Ethics committee of the Charité—Universitätsmedizin Berlin (EA2/067/15 and EA/2/020/14). All donors provided their written consent.

## IMC and scRNA-seq

Lung adenocarcinoma and adjacent tissues were obtained from 12 patients undergoing primary surgery. Patients were aware of the planned research and agreed to the use of tissue. Research was approved by vote EA4/164/19 of the ethics committee at Charité—Universitätsmedizin Berlin. Further information can be found in Bischoff et al (Bischoff et al, 2021). We used cDNA from all 12 patients for the targeted sequencing (method part is found below) and scRNA-seq analysis in this study. For the IMC analysis, we transferred material of 5 patients (p018, p019, p023, p024, p027) plus a tonsil core serving as a positive control (11 cores in total) into one TMA. Patient IDs and IMC ablations are listed in the Appendix Table S6.

## In vitro studies—isolation and preparation of PBMC

For the isolation of PBMC, human blood was first diluted 1:2 in DPBS (Gibco, Thermo Fisher Scientific, UK) at room temperature (RT), followed by standard density gradient centrifugation using Pancoll (PAN-Biotech GmbH, Aidenbach, Germany). PBMCs were either used directly or cryopreserved in 1 mL freezing medium consisting of FBS (heat-inactivated, Sigma-Aldrich, St. Louis, USA) and 10% DMSO (Sigma-Aldrich, St. Louis, USA), and stored in liquid nitrogen.

For experiments using unseparated PBMCs, the concentration was adjusted to $1 \times 10^6$ cells per mL with culture medium consisting of RPMI 1640 (VLE, Biochrom, Berlin, Germany, since 11/2020 Pan Biotec), 10% FBS Lot 605c, and 1% penicillin/streptomycin (10,000 U/mL/10,000 µg/mL, Biochrom, Berlin, Germany).

## In vitro studies—T cell separation

For $CD3^+$ enrichment, the Pan T cell Isolation Kit and for $CD8^+$ enrichment, the $CD8^+$ T Cell Isolation Kit (both Miltenyi Biotec, Bergisch Gladbach, Germany) were used according to the manufacturer's protocol. Magnetic activated cell sorting (MACS) was handled with MACS buffer consisting of DPBS, 0.5% BSA, and 2 mM EDTA (both Sigma-Aldrich, St. Louis, USA). Isolated T cells were counted, and the concentration was adjusted to $1 \times 10^6$ cells per mL with culture medium.

## In vitro studies—cell stimulation

For cell stimulation, anti-CD3 alone or anti-CD3 collectively with either anti-KLRF1 or the IgG1 isotype control antibody were added in 50 µL DPBS per well in a 96-well round-bottom plate, accordingly (Appendix Table S1). The antibodies were plate-bound by incubation overnight at 4 °C or for at least 2 h in a humidified atmosphere (5% $CO_2$, 37 °C, ≥95% RH). Before cell addition, wells were carefully washed twice with 200 µL DPBS.

For conventional stimulation, 2 µg/mL anti-CD28 in 200 µL cell suspension ($2 \times 10^5$ cells) per well was added. Cells were incubated for 6 or 24 h in a humidified atmosphere (5% $CO_2$, 37 °C, ≥95% RH). During the last 4 h of stimulation, 4 µl Brefeldin A (0.5 mg/ml, 2 µg/well, Sigma-Aldrich, St. Louis, USA) was added to each well.

## In vitro studies—flow cytometry staining

Cells were harvested, centrifuged (5 min, $300 \times g$, 4 °C), and washed once with 1 mL DPBS. Samples were stained with the Zombie UV Fixable Viability Kit (BioLegend, San Diego, USA) for 15 min at 4 °C. After viability staining, cells were washed in 1 mL FACS buffer (5 min, $300 \times g$, 4 °C) consisting of DPBS, 2% heat-inactivated FBS (Sigma-Aldrich, St. Louis, USA), and 0.1% sodium azide (SERVA, Heidelberg, Germany).

For Fc receptor blocking, cells were incubated with 3.2 mg/mL Beriglobin ® (CSL Behring GmbH, Marburg, Germany) in FACS buffer at 4 °C. The antibody panel used for receptor inhibition experiments is listed in Appendix Table S2; the panel for AICL upregulation is listed in Appendix Table S3.

Following blocking, cells were stained with the surface antibody mix (prepared in Beriglobin solution) in a final volume of 40 µL and incubated for 20 min at 4 °C. Then, cells were washed with 1 mL FACS buffer (5 min, $300 \times g$, 4 °C), fixed, and permeabilized using 200 µL Cytofix/Cytoperm (BD Biosciences, Heidelberg, Germany) for 20 min at 4 °C.

Cells were washed twice with Perm/Wash buffer (BioLegend, San Diego, USA) (5 min, $500 \times g$, 4 °C for each wash). Intracellular

staining was performed using a prepared antibody mix in Perm/Wash buffer (final volume: 40 μL), and cells were incubated for 30 min at 4 °C. After staining, cells were washed again twice with Perm/Wash buffer (5 min, 500 × g, 4 °C) and resuspended in 200 μL FACS buffer.

Samples were measured on a BD LSRFortessa flow cytometer (BD Biosciences, Heidelberg, Germany) either immediately or after overnight storage at 4 °C in the dark. Data were analyzed using FlowJo software version 10 (FlowJo LLC, Ashland, USA).

## In vitro studies –FACS sorting

Freshly thawed PBMC from buffy coats were enriched for $CD3^+$ T cells as mentioned in the section "T cell separation" of the methods part. $CD3^+$ T cells were incubated in 3.2 mg/ml Beriglobin ($4 \times 10^6$ cells per 50 μl) for at least 5 min and the antibody mix in DPBS supplemented with 2% FBS was added for surface staining with 50 μl per $4 \times 10^6$ cells. Cells were stained for 20 min at 4 °C in the dark and washed twice with DPBS + 2% FBS. Shortly before sorting 4′,6-diamidine-2-phenylindole dihydrochloride (DAPI, Roche, Penzberg, Germany) was added (1:250) and the following T-cell subpopulations were sorted using the flow cytometer BD AriaFusion (BD Biosciences, Germany): CD8 + T cells, CD4+ $AICL^+$ $KLRG1^+/KLRF1^+$ and $AICL^-$ $KLRG1^+/KLRF1^+$ T cells. Antibodies used for surface staining are listed in the Appendix Table S4.

## IMC—antibody-metal conjugation and titration

The conjugation of metals to the relevant antibodies was performed following the Maxpar X8 antibody labeling protocol from Fluidigm. Final concentrations were measured with the Nanodrop 1000 (Thermo Fisher Scientific). Titrations of untested conjugated antibodies was done staining freshly cut FFPE slides with four human pancreatic duct adenocarcinoma (PDAC) and four human tonsil cores (tissue preparation is reported in the next paragraph). Sequential dilutions of the antibody mix were put on one pair of PDAC-tonsil cores. At the Flow & Mass Cytometry Core Facility (BIH-Charité, Berlin), selected regions of interest (ROIs) were ablated with the Hyperion Tissue Imager and analyzed by time-of-flight mass spectrometry with the Helios Mass Cytometer (both from Fluidigm). The signal was visualized with the MCD Viewer data processing software (Fluidigm) and the optimal dilution chosen for subsequent staining.

## IMC—TMA preparation and staining

The tissue sections were sliced from the paraffin block and baked at 60 °C for 1 h at the iPATH Core Unit (Charité, Berlin). The freshly cut tissue sections were de-paraffinized in two changes of fresh xylene for 8 min each. Consequently, tissue sections were rehydrated in two changes of 100% ethanol for 3 min each, and one change of 95, 80, and 70% ethanol for 2 min each. The slides were subsequently rinsed twice in distilled water for 5 min. The samples were then immersed in pre-warmed antigen retrieval buffer (10 mM Tris base, 1 mM EDTA, 0.05% Tween-20, pH 9.0) in a container and incubated for 30 min in a water bath at 95 °C. After this step, the slides were cooled by placing the container in ice-water for 30 min and subsequently rinsed with distilled water and

PBS with 0.1% Tween-20 (PBS-T) for 5 min each. The tissue was encircled with a hydrophobic pen and blocked with blocking buffer (3% BSA, 5% goat serum in PBS) in a hydration chamber for 1 h at RT. In the meantime, the antibody mix was prepared by spinning the antibodies at $13,000 \times g$ for 2 min and pipetting from the top of the vial to avoid antibody aggregates. Details of the antibodies and dilution factors can be found in Appendix Table S5. The antibody mix (in 1% BSA in PBS) was pipetted onto the sections and incubated at 4 °C overnight. On the following day, the sections were rinsed twice with PBS-T for 5 min each and twice with PBS for 8 min each. Then, the slides were incubated twice with Intercalator (Ir191-Ir193 at 1:400) in PBS in a hydration chamber for 30 min each at RT. Finally, the sections were washed once with PBS for 5 min and dipped in deionized water three times before air drying for at least 20 min. Finally, the selected regions of interest (ROIs) were ablated pixel by pixel with the Hyperion Tissue Imager and analyzed by time-of-flight mass spectrometry with the Helios Mass Cytometer (both Fluidigm).

## IMC—processing of Hyperion output

ROIs were processed and segmented with the Bodenmiller pipeline (Windhager et al, 2023) by propagation of the nuclear signals. In short, a python script was used to extract the ion counts and pre-process the raw data (hot pixel filtering) into images that were then read by CellProfiler (Carpenter et al, 2006) to produce cropped images in .h5 format which were loaded into Ilastik (Berg et al, 2019) for pixel-based training of the cellular objects nuclei, cytoplasm and background. The pixel-based probabilities for these three layers were exported for the entire ROI and loaded into CellProfiler again, where the nuclear probabilities were used to create objects that were expanded until they met either the expanding neighboring cells or the background.

IMC raw .mdc files, segmentation masks and integrated data at the cellular level can be downloaded from the Zenodo repository (https://doi.org/10.5281/zenodo.13947395).

## IMC—Generation of tumor masks

To spatially separate the cells in each ROI, Euclidean distance masks were created with a macro written for ImageJ (Schneider et al, 2012). Tumor areas were identified by summing up CK18 and alpha-SMA channels. The composite image was first Gaussian-filtered with 0.8 sigma and then automatically thresholded using the Otsu algorithm. Objects were then smoothened by using open with 5 iterations with a count of 1 adjacent pixels. Objects smaller than 200 px were then filtered out before the remaining objects were dilated with 9 iterations and a count of 1 adjacent pixel. Finally, the mask was inverted and the Euclidean distance map calculated.

## IMC—data processing with adapted SPECTRE code

Cellular segmentation masks and processed marker channels were both read as .tiff files into R and processed with an adapted version of the SPECTRE code (Ashhurst et al, 2022). The adaptations in that code allowed for additional pre-processing steps, such as cell-wise spillover correction according to the spillover matrix provided by the in-house cytometry core facility.

Cells were divided using a Boolean gating strategy into four subsets (Gate 1 to 4) using the spillover-corrected and asinh-transformed signals of CD19, CD3, and CD45 (B cells, non-B T cells, remaining immune and non-immune cells). Depending on the marker, we either used simple signal thresholds or polygonal gate lines to gate on the corresponding populations (Fig. EV2A; Appendix Fig. S2A). We tested different asinh co-factors and compared the heatmaps and found that a co-factor of two yields the most meaningful data. For clustering and dimension reduction, we z-scored the marker signals over the entire dataset and used Phenograph to cluster (cytofkit package version 0.99.0), UMAP (uwot package version 0.1.14), and the complex heatmap engine (version 2.12.0) to depict expression levels of the clustered populations. After individual clustering of each subset, Euclidean distance maps were loaded as matrices into R. The Spatial position of each cell (center of mass) was then mapped onto the supplied maps, and cells were flagged accordingly.

The code to process raw data can be downloaded from the github repo SawitzkiGroup/KLRF1-Tpex. Processed data can be downloaded from the Zenodo repository mentioned earlier, https://doi.org/10.5281/zenodo.13947395.

## scRNA-seq—targeted sequencing of IFN-γ in lung adenocarcinoma cDNA of Bischoff et al

The non-targeted single cell measurements published earlier (Bischoff et al, 2021) were used to extract the combined coverage information of the .bam files around the IFNG locus (±10 kb). We used the coverage information to derive the most suitable inner and outer primers by applying the TAP-seq R package according to the package vignette (https://github.com/argschwind/TAPseq) (Schraivogel et al, 2020). The resulting inner and outer primers are provided in the Appendix Table S7.

All primers (outer, inner, ReadOne) were ordered from IDT in desalted format; the inner primer was modified with 5'-GTGACTGGAGTTCAGACGTGTGCTCTTCCGATCT-3' as a PCR handle at the 5' end.

The targeted library was conducted in three PCR steps:

(1) For the outer PCR, we used 10 ng amplified cDNA (Step 2.3 of 10x Genomics 3'scRNA), 2.5 μL 100 μM outer primer, 4 μL 10 μM partial Read 1 (sequence from 10x Genomics manual: 5'-CTACACGACGCTCTTCCGATCT-3',) 50 μL KAPA HiFi HS RM (Roche, KK2601). The PCR was performed with the following cycle numbers: 1x (95 °C, 3 min), 10x (98 °C, 20 s, 67 °C, 60 s, 72 °C, 60 s), 1x (72 °C, 5 min). The PCR product was cleaned using SPRIselect beads (Beckman Coulter) as a double-sided size selection using 0.55x and 1.2x as the first and second step, respectively.

(2) For the inner PCR, the whole 30 μL Eluate of the outer PCR was combined with 2.5 μL 100 μM inner primer, 4 μL 10 μM partial Read 1, 50 μL KAPA HiFi HS RM and water to reach a total volume of 100 μL. Afterwards, a PCR using the same cycle scheme as in the outer PCR was conducted. For double-sided size selection, we used Cleanup 0.6x and 1.2x.

(3) For the indexing PCR, 10 ng of the cleaned PCR product were mixed with 20 μL dual Index TT Set A from 10x genomics and 50 μL KAPA HiFi HS RM in a 100 μL reaction. The PCR was performed with the following cycle numbers: 1x (95 °C, 3 min),

1x (98 °C, 45 s), 10x (98 °C, 20 s, 54 °C, 30 s, 72 °C, 20 s), 1x (72 °C, 1 min) followed by cleanup using SPRIselect (0.55x as 1st and 1.2x as 2nd). Library quality was assessed using the TAPE station.

## Public scRNA-seq data processing

The analysis of scRNA-seq data was carried out in Python using the scanpy package (Wolf et al, 2018) (v1.9.6). Raw cellranger count data output was downloaded from Codeocean as deposited by the original authors (Bischoff et al, 2021). Only matching tumor and normal tissue samples from patients with diagnosed lung adenocarcinoma (M8140/3), namely, were selected for subsequent analysis. T/NK cells were selected as annotated by the original authors, thereby keeping cells passing the original quality control thresholds (500–10,000 genes expressed, 1000–100,000 UMIs counted, less than 30% mitochondrial reads, and less than 5% hemoglobin reads). HTAN MSK data for T cells was obtained from CZI cellxgene (url: https://cellxgene.cziscience.com/collections/62e8f058-9c37-48bc-9200-e767f318a8ec) and the data of Laughney et al directly from NCBI gene expression omnibus (identifier: GSE123903). For more information on how those datasets were processed, please see the associated processing Jupyter notebooks.

We recalculated principle component analysis (PCA), k-nearest neighbor graph, and UMAP embedding (McInnes et al, 2020). Counts were normalized using scanpy.pp.normalize_total, and expression was not scaled except for (PCA) or if otherwise specified. To reliably remove NK cells from the T cells, we gated for the latter as $CD3E^+$ $CD3D/G^+$ $TRAC^+$ cells as mentioned in the text instead of using clustering, since both cell types formed a continuous phenotype in scRNA-seq (see Figs. 6C and EV3B).

## Characterization of KLRF1+ cells in scRNA-seq data

Cells were clustered using the Leiden algorithm implemented by the package leidenalg (Traag et al, 2019) (v0.10.1, 10.5281/zenodo.1469356) with resolution 1.3. Cluster-wise average CD4 and CD8A/B gene expression was calculated. The clusters were separated by which marker was dominant, grouping them into $CD4^+$ and $CD8^+$ T clusters. Heatmaps were generated using cluster-wise marker expression, then marker-wise z-scoring across $CD8^+$ clusters and plotted with PyComplexHeatmap (Ding et al, 2023). Cluster 4 and Cluster 14 showed markedly higher KLRF1 expression compared to the other $CD8^+$ T cell clusters (log2-fold change = 5.6, adjusted $p$ value = 2.9e-120 with Wilcoxon rank test).

## Public multimodeal scRNA-seq + ATAC data processing

The analysis of the multimodal dataset (Thomson et al, 2023) was carried out in R using the Seurat (Hao et al, 2024) and ArchR (Granja et al, 2021) packages using the deposited datasets on the zenodo repository https://doi.org/10.5281/zenodo.10372038. T cells in single_positive_diet_v3.rds (v4, https://doi.org/10.5281/zenodo.17101381) were gated based on $CD3E > 0$ & ($CD3D>0$ | $CD3G>0$) and $TRAC>0$ (Fig. EV3, this gating strategy was used consistently throughout the manuscript), followed by gating on $CD8^+$ T cells using the condition ($CD8A>0$ | $CD8B>0$ | ($CD8A>0$ & $CD8B>0$)) combined with ($TRDC=0$ & $TRGC1=0$ & $TRGC2=0$) and $CD4=0$

(exclusion of γδ and CD4+ T cell markers). The gated CD8+ T cells were then clustered with the Leiden algorithm implemented in the Seurat package with resolution 0.6 using the weighted-nearest-neighbor framework that integrates scRNA-seq PCA and ATAC LSI components. Clusters and UMAP coordinates were then mapped into the ArchR project obtained from the same zenodo repository (v3 https://doi.org/10.5281/zenodo.16922998) using pool_id, pbmc_sample_id, and barcodes columns of the Seurat meta.data. CD4+ T cells present in the scRNA-seq data were mapped as "other" and Arrow files were processed according to the ArchR pipeline by adding group coverages and reproducible peak sets and adding the peak matrix. Marker genes from the GeneScoreMatrix were then calculated comparing the clusters 7 and 3.

## Data availability

The imaging mass cytometry (IMC) and processed scRNA-seq data from this publication have been deposited in the Zenodo database https://zenodo.org/records/13947395 and assigned the identifier 13947395.

The source data of this paper are collected in the following database record: biostudies:S-SCDT-10_1038-S44319-026-00732-5.

## Peer review information

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

## Acknowledgements

We would like to acknowledge the technical support of the BIH Cytometry Core Facility. This work was supported by the German Research Foundation (DFG) SA1383/3-1, KFO BECAUSE-Y project SA1383/4-1 and CRC 1444 project 427826188 to BS, RTG2424 to NB; the German Federal Ministry of Education and Research (BMBF) FKZ 01EP2201 NKSG, FKZ 01KX2121 RAPID, FKZ 01KX2021 COVIM to BS, FKZ 02NUK086E to NB; the European Research Council: MSCA TOP-GUT project B8000303 to BS; the Volkswagen Foundation: Swarm Learning project 8000354 to BS; and a Charite' 3R project to BS.

## Author contributions

**Matthias Barone**: Conceptualization; Software; Formal analysis; Visualization; Writing—original draft; Writing—review and editing. **Stefan Peidli**: Software; Investigation; Writing—original draft; Writing—review and editing. **Anika Neuschulz**: Software; Formal analysis; Writing—review and editing. **Karla Riesterer**: Investigation; Writing—original draft. **Christina Iwert**: Investigation; Methodology; Writing—original draft. **Laia Junquera**: Formal analysis; Investigation; Writing—original draft. **Somesh Sai**: Software; Formal analysis. **Olufemi Bolaji**: Software; Validation. **Diana Bakoueva**: Investigation. **Christine Appelt**: Validation; Investigation; Writing—original draft; Writing—review and editing. **Benedikt Obermayer**: Software; Formal analysis; Validation. **Bertram Klinger**: Investigation. **Alexandra Trinks**: Investigation; Methodology. **Anja Sieber**: Investigation; Methodology. **Nils Blüthgen**: Conceptualization; Supervision; Funding acquisition; Investigation; Methodology; Writing—original

draft; Writing—review and editing. **Birgit Sawitzki**: Conceptualization; Supervision; Funding acquisition; Writing—original draft; Project administration; Writing—review and editing.

Source data underlying figure panels in this paper may have individual authorship assigned. Where available, figure panel/source data authorship is listed in the following database record: biostudies:S-SCDT-10_1038-S44319-026-00732-5.

## Funding

## Disclosure and competing interests statement

The authors declare no competing interests.

# Expanded View Figures

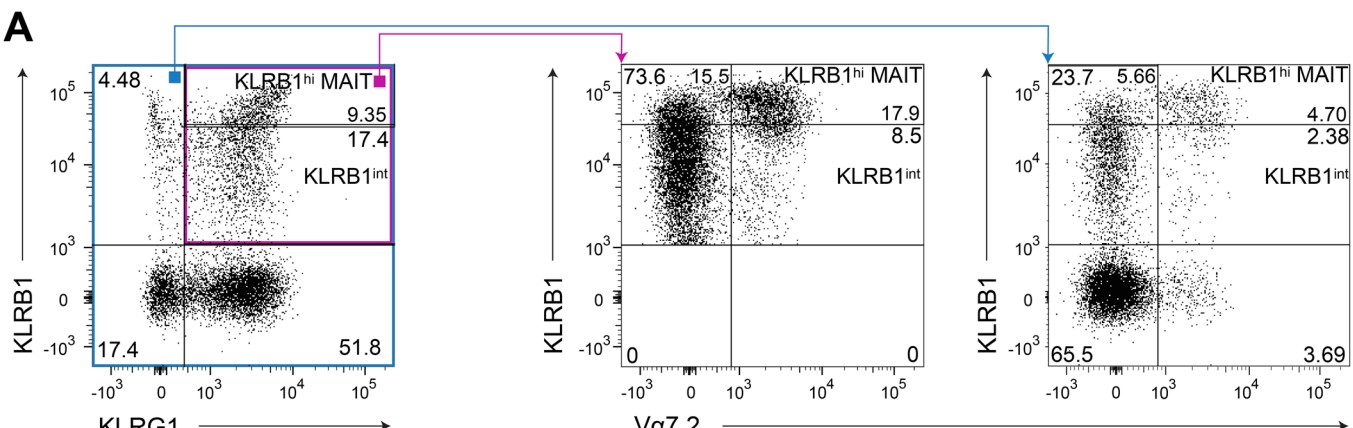

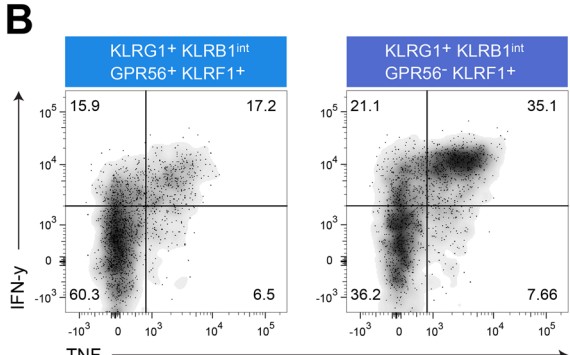

**Figure EV1.  Identification and exclusion of KLRBhi CD8⁺ MAIT cells from the analysis in Fig. 1.**

(A) Exemplary dot plots showing KLRB1 versus Vα7.2 staining of pre-gated KLRB1⁺ cells of CD8⁺ T cells from PBMC to identify and exclude KLRB1high-expressing MAIT cells. (B) Exemplary dot plots belonging to the analysis in Fig. 1D showing the TNF and IFN-γ producing CD8⁺ KLRG1⁺ KLRB1int GPR56⁺ KLRF1⁺ T cells compared to their GPR56- counterparts. An additional shading was added to the dot plots to highlight the populations.

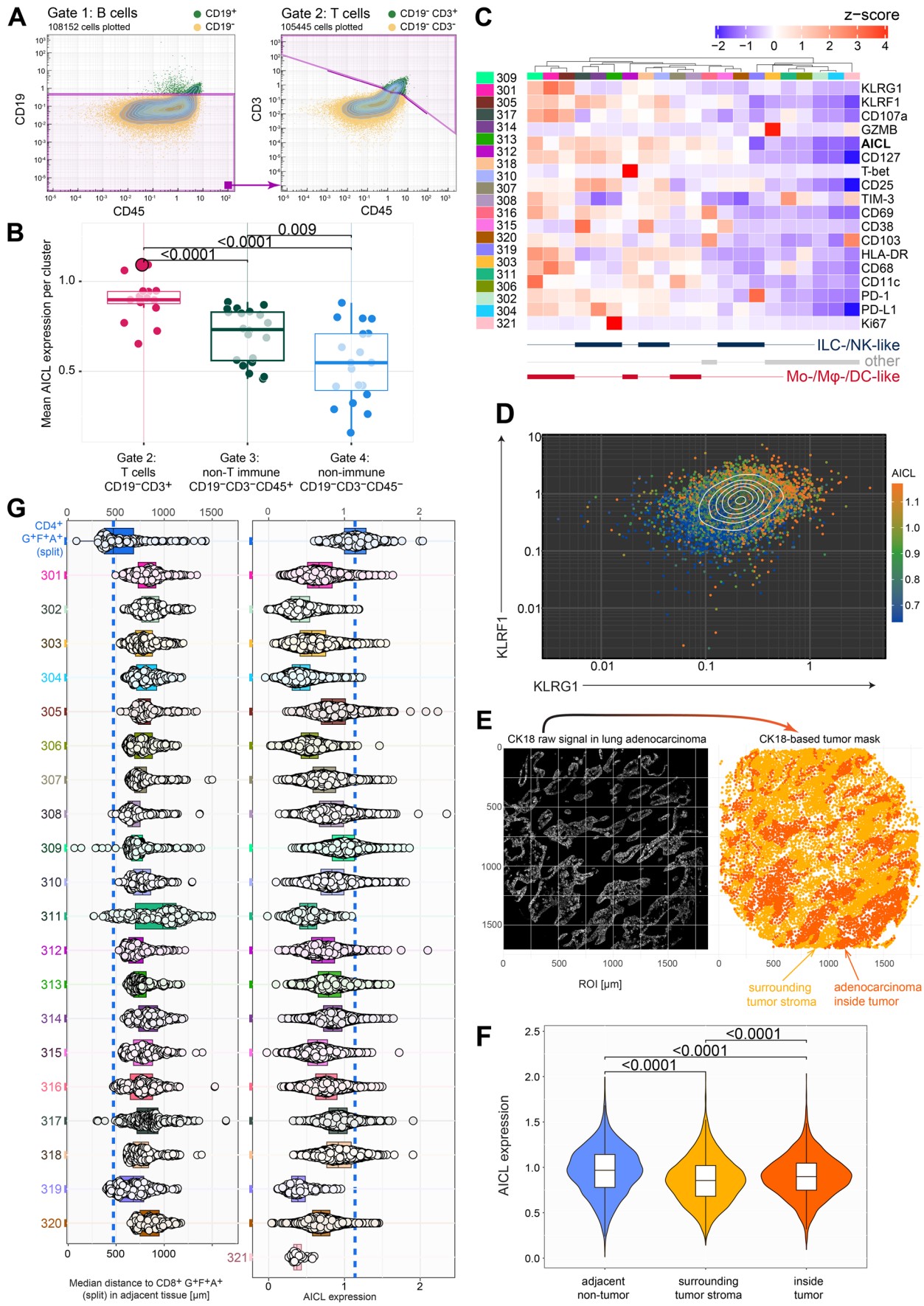

**Figure EV2.  IMC cell gating strategy, quality control, and analysis of the non-B/non-T immune cell subset.**

(A) Boolean gating strategy employed to identify T cells (CD19⁻ CD3⁺) used for this study. B cells were removed by gating on CD19 marker expression, and the remaining cells were gated for CD3⁺ CD45⁺ double-positive cells using the polygonal line as shown. (B) AICL expression across the T cell (Gate 2), non-B/non-T immune cell (Gate 3), and remaining non-immune stromal and epithelial cell (Gate 4) clusters in tumor and adjacent tissue samples combined. Plotted are cluster-wise mean AICL expressions of all 16 (Gate 2), 20 (Gate 3), and 19 (Gate 4) clusters ($n = 5$, biological replicates). In Gate 2, the CD4⁺CD8⁺ G⁺F⁺A⁺ T cell cluster is highlighted with a bigger dot size. Statistical analysis was performed by the Wilcoxon rank-sum test with Benjamini and Hochberg adjustment. Exact *p* values are noted in the plot. Box plots show the median (center line) and interquartile range (box limits 25th to 75th percentile, 1x IQR), whiskers extend to the farthest values but no more than 1.5x IQR from the box. (C) Heatmap displaying *z*-scored marker expression of the identified non-B/non-T immune cell clusters (Gate 3). Cluster assignment is shown on the bottom of the heatmap: (blue) ILC-/NK-like: innate lymphoid and natural killer cell-like; (red) myeloid clusters Mo-/Mφ-/DC-like: monocyte-, macrophage-, dendritic cell-like; (gray) other: other immune cell clusters. (D) Dot plot of KLRG1 versus KLRF1 marker expressions of all gated CD3⁺ T cells, color-coded by AICL marker expression (color gradient clipped at 15ᵗʰ to 85th percentile of AICL expression). (E) The CK18 raw signal was used to create lung adenocarcinoma tumor masks for each ROI. The masks were loaded into R to map cell centroid positions. Shown is a representative ROI with the CK18 signal (white) aligned with the cell positions marked in red/orange for cells mapped inside/outside of the binary mask. The same color-code was also used in Fig. 5D. (F) AICL expression of CD4⁺ T cells in adjacent non-tumor samples (blue). Cells in lung adenocarcinoma tissue are sectioned into tumor (red) and surrounding tumor stroma (orange). Same color-code as in panel (E); $n = 5$, biological replicates. Statistical analysis was performed with a two-tailed *t*-test with Benjamini and Hochberg adjustment. Exact *p* values are noted in the plot. Box plots show the median (center line) and interquartile range (box limits 25th to 75th percentile, 1x IQR), whiskers extend to 1.5x IQR from the box G) Left: Neighborhood analysis of cells belonging to the split CD8⁺ G⁺F⁺A⁺ cluster probing median distances to CD4⁺ G⁺F⁺A⁺ T cells (split, see Fig. 5A) and cells of the non-B/non-T immune cell clusters (Gate 3, panel C). Right: AICL expression of the cells probed in the neighborhood analysis. Box plots show the median (center line) and interquartile range (box limits 25th to 75th percentile, 1x IQR), whiskers extend to 1.5x IQR from the box. Median distances and AICL expression of the split CD4⁺ G⁺F⁺A⁺ T cells are highlighted as a blue dashed line in the corresponding plots (both plots $n = 5$, biological replicates).

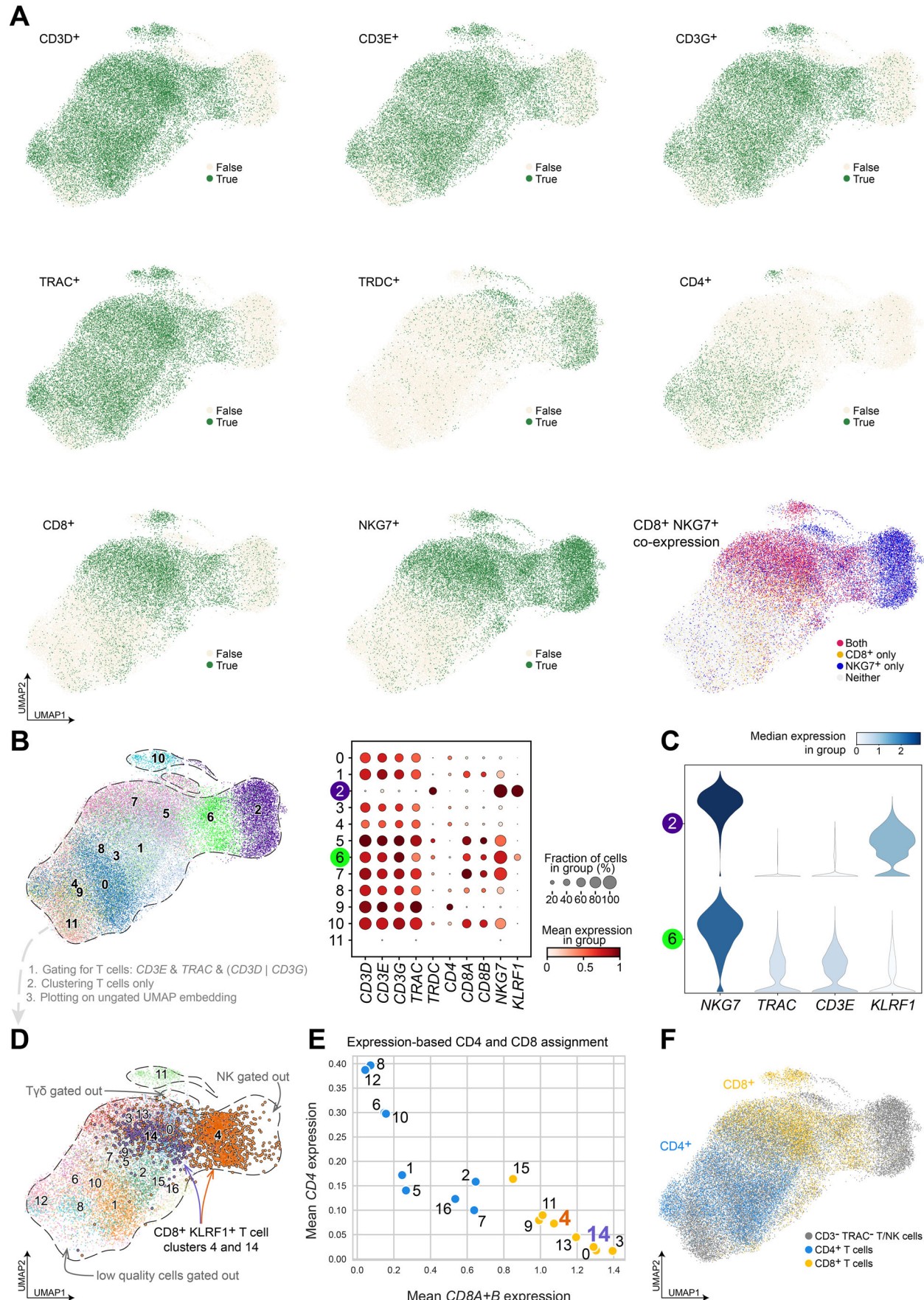

◀   **Figure EV3.   scRNA-seq T cell gating strategy of annotated NK/T cells in Bischoff et al.**

(A) Markers typically associated with NK cells (*NKG7*), γδ T cells (*TRDC*), and T cells (*CD3, CD4, CD8,* and *TRAC*) were mapped onto the UMAP embedding, with each marker plotted separately. Cells expressing these genes are highlighted in green. Bottom right: Cells expressing both *CD8* and *NKG7* are shown in red, while cells with single reads of either *CD8* (yellow) or *NKG7* (blue) are distinguished. (B) Clustering of the entire NK/T cell space predominantly yielded a T cell cluster (6, green) and an NK cell cluster (2, violet). A dot plot highlights a notable proportion of *TRDC*+ and *NKG7*+ impurities within cluster 6. (C) Violin plots depict gene expression of NK and T cell markers in clusters 2 and 6 of ungated NK/T cells, showing that cluster 6, primarily composed of T cells, expresses lower KLRF1 levels than cluster 2. (D) The NK/T cell subset was gated on T cells expressing *CD3E*+ and *TRAC*+ along with either *CD3D*+ or *CD3G*+. The gated T cells were re-clustered and plotted using the UMAP coordinates from panel B. The *KLRF1*+ clusters 14 and 4, referenced in Fig. 6, are highlighted. (E) T cell clusters were assigned based on *CD4* and *CD8* expression and color-coded by main lineage, with clusters 14 and 4 (*KLRF1*+) highlighted as in Fig. 6. (F) Using the color scheme from panel (E), CD4+ (blue) and CD8+ (yellow) T cells are plotted along with gated-out cells (gray).

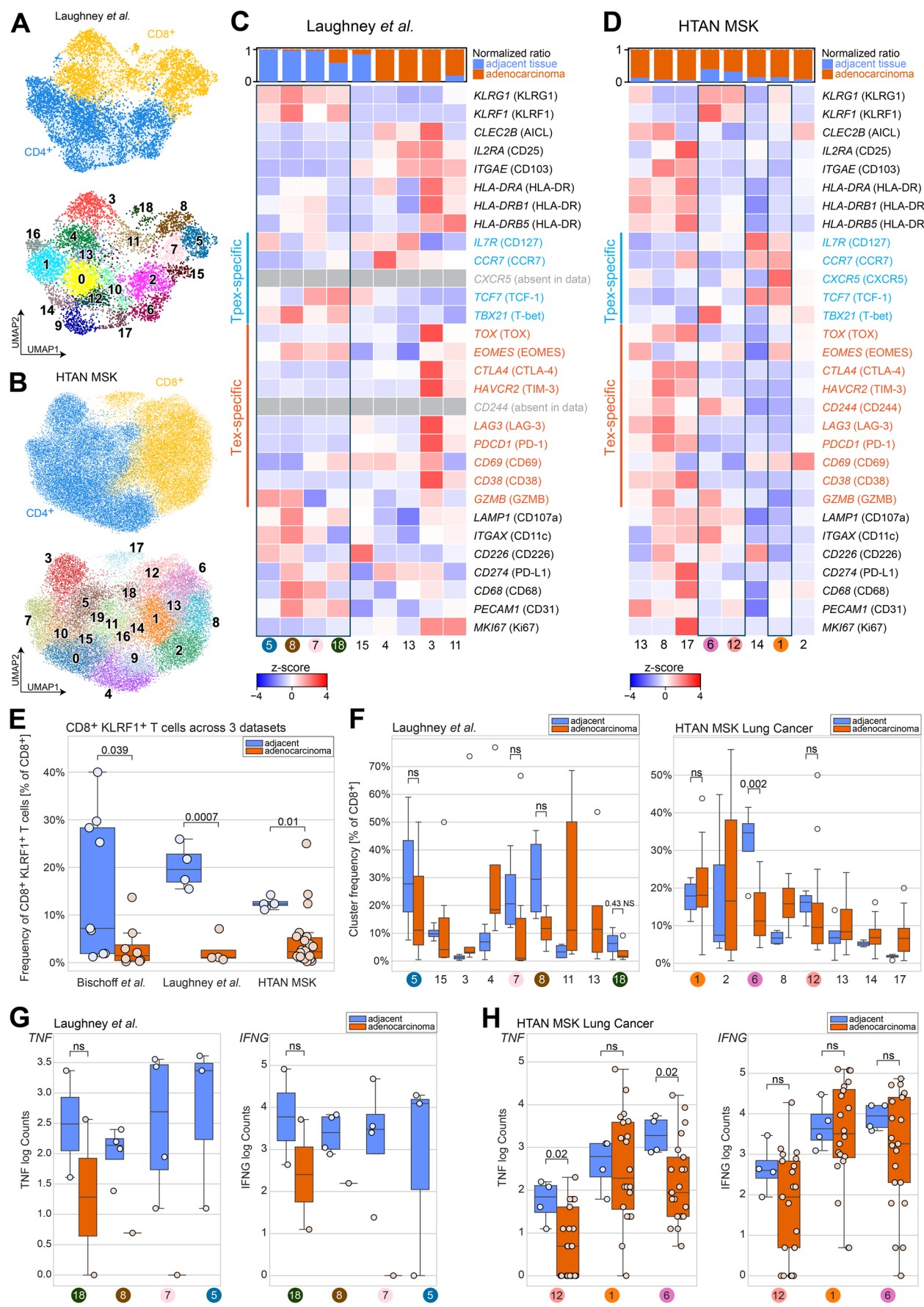

Figure EV4.  CD8$^+$ T cell analysis in Laughney et al and HTAN MSK lung cancer scRNA-seq datasets.

Both datasets (Laughney et al, 2020; Chan et al, 2021) were gated using the same strategy used throughout the manuscript by identifying T cells based on *CD3E*>0 and (*CD3D*>0 | *CD3G*>0) together with *TRAC*>0. (Figs. 6 and EV3B,D) to exclude NK cells before analysis. (**A**, **B**) UMAP embeddings of both datasets, color-coded by major CD4$^+$ and CD8$^+$ lineage markers and Leiden clusters. (**C**, **D**) Heatmaps of CD8$^+$ T cell clusters identified in both datasets, with markers grouped by activation and exhaustion. KLRF1$^+$ clusters are highlighted with boxes. (**E**) Frequencies of *KLRF1*$^+$ CD8$^+$ T cells relative to all CD8$^+$ T cells in all three datasets, separated into carcinoma tissue (orange) and adjacent non-tumor (blue) tissue samples (Bischoff $n = 12$, Laughney $n = 9$, HTAN $n = 26$, all biological replicates). Statistical analysis was performed by the Wilcoxon rank-sum test. Exact $p$ values are noted in the plot. Box plots show the median (center line) and interquartile range (box limits 25th to 75th percentile, 1x IQR), whiskers extend to the farthest values but no more than 1.5x IQR from the box. All data points are drawn. (**F**) Patient-wise frequencies of CD8$^+$ T cell clusters relative to all CD8$^+$ T cells, separated by carcinoma (orange) and adjacent non-tumor normal (blue) tissue (Laughney $n = 9$, HTAN $n = 26$, both datasets using biological replicates). Statistical analysis was performed with a one-tailed Wilcoxon rank-sum test. Exact $p$ value is noted in the plot; ns $p > 0.05$. Box plots show the median (center line) and interquartile range (box limits 25th to 75th percentile, 1x IQR), whiskers extend to the farthest values but no more than 1.5x IQR from the box. Only data points beyond the whiskers are drawn. (**G**, **H**) Cytokine production of CD8$^+$ KLRF1$^+$ T cell clusters, separated by carcinoma (orange) and adjacent non-tumor normal (blue) tissue (Laughney $n = 9$, HTAN $n = 26$, both biological replicates). Statistical analysis was performed by a one-tailed Wilcoxon rank-sum test. Exact $p$ values are noted in the plot; ns $p > 0.05$. Box plots show the median (center line) and interquartile range (box limits 25th to 75th percentile, 1x IQR), whiskers extend to the farthest values but no more than 1.5x IQR from the box. All data points are drawn.

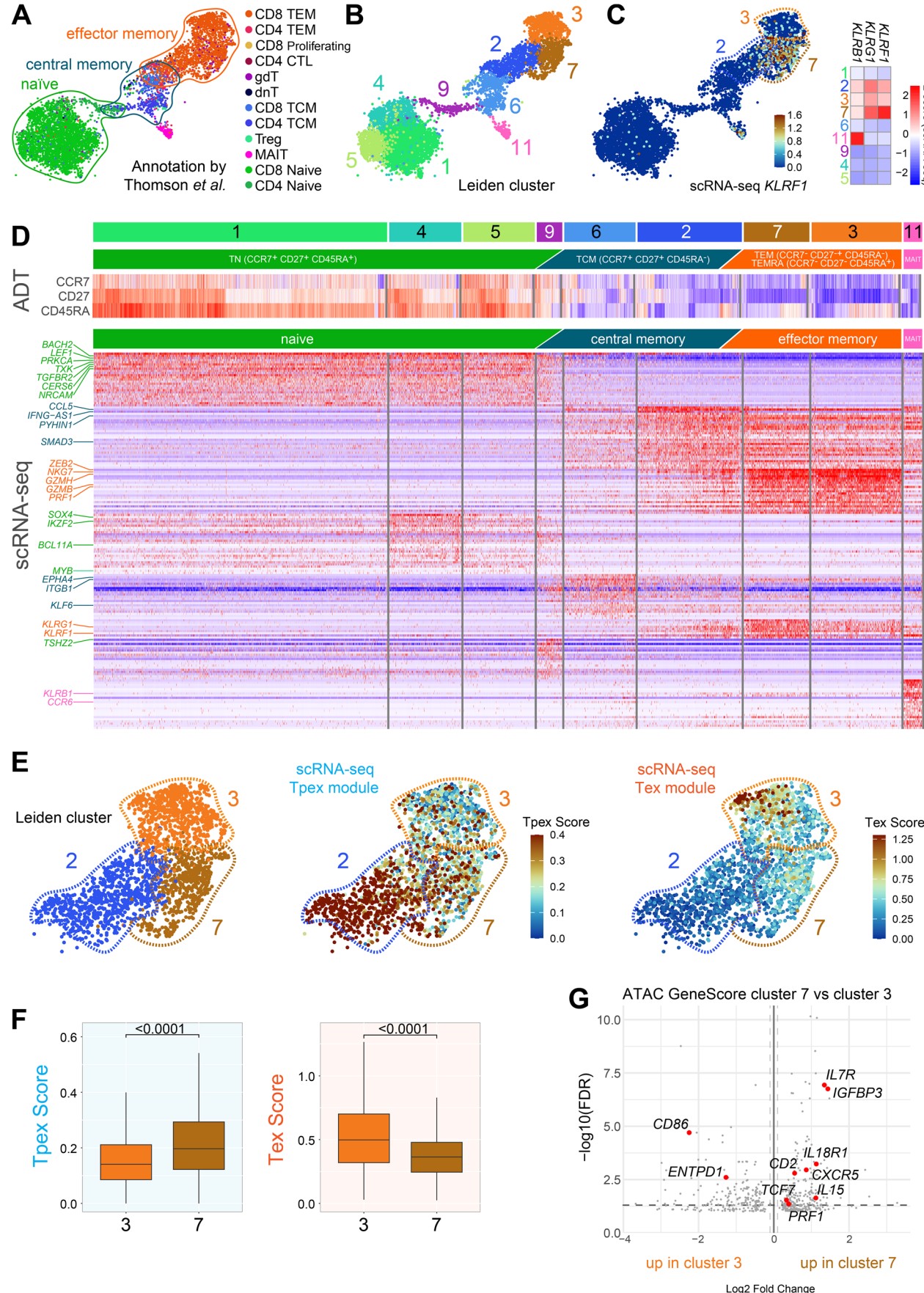

**Figure EV5. CD8⁺ T cell analysis in Thomson et al trimodal scRNA-seq, ADT, and ATAC dataset.**

The dataset (Thomson et al, 2023) was gated using an extended version of the gating strategy to identify CD8⁺ T cells while excluding CD4⁺ and γδ T cells (see methods part). (A) UMAP embedding of gated CD8⁺ T cells, color-coded by the predicted T cell types as reported by Thomson et al (predicted.t-celltype.l2). Differentiation stages are colored in similar tones: naïve (green), central memory T cells (TCM, blue), mucosal-associated invariant T cells (MAIT, pink), effector memory (TEM) and cytotoxic T lymphocytes (CTL, both orange). Minor populations were colored separately: regulatory T cells (Treg, aquamarine), double-negative T cells (dnT, black), γδ T cells (gdT, turquoise). (B) UMAP embedding color-coded by Leiden clusters determined by the weighted-nearest neighbor (WNN) graph constructed from scRNA-seq PCA and ATAC LSI embeddings. (C) UMAP embedding color-coded by the KLRF1 expression and a z-scored heatmap of KLRs transcription levels used in our study. (D) Antibody-derived tags used in Thomson et al to gate into naïve T cell subsets plotted as a z-scored heatmap across the calculated Leiden clusters. Bottom: Differential gene expression list with top 40 genes per cluster. Genes used to align and annotate the clusters are highlighted in the same color-code as in panel (A). (E) UMAP embeddings of Tex and Tpex module scores calculated from the gene sets shown in the Appendix Table S8 (F) Comparison of Tpex and Tex scores of the clusters 3 and 7 ($n = 16$, biological replicates. Cluster 3 730 cells; cluster 7 530 cells). Statistical analysis was performed by a two-tailed $t$-test. Exact $p$ values are noted in the plot. Box plots show the median (center line) and interquartile range (box limits 25th to 75th percentile, 1x IQR), whiskers extend to the farthest values but no more than 1.5x IQR from the box. (G) Differential ATAC gene scores between clusters 7 and 3. Gene scores enriched in the KLRF1⁺ cluster 7 are log2FC >0. Loci associated with memory/Tpex states and exhaustion, and FDR <0.05 are highlighted.

