## [Peer Review File · EMBO Reports]

The AICL-KLRF1 axis supports CD4-CD8 T cell communication and cytokine competence in pre-exhausted CD8+ T cells

Matthias Barone, Stefan Peidli, Anika Neuschulz, Karla Riesterer, Christina Iwert, Laia Junquera, Somesh Sai, Olufemi Bolaji, Diana Bakoueva, Christine Appelt, Benedikt Obermayer, Bertram Klingner, Alexandra Trinks, Anja Sieber, Nils Blüthgen, and Birgit Sawitzki

Corresponding author(s): Birgit Sawitzki (birgit.sawitzki@charite.de)

Review Timeline:

Submission Date:	28th Apr 25
Editorial Decision:	6th Jun 25
Revision Received:	8th Dec 25
Editorial Decision:	13th Jan 26
Revision Received:	29th Jan 26
Accepted:	19th Feb 26

Editor: Deniz Senyilmaz Tiebe / Achim Breiling

Transaction Report:

Dear Birgit,

Thank you for transferring your manuscript to EMBO Reports, which was now seen by three referees, whose reports are copied below.

Referees express interest in the proposed effect of AICL-KLRF1 mediated CD4⁺-CD8⁺ T Cell Interaction in cytokine production in pre-exhausted T Cells. However, they also raise some concerns that need to be addressed to consider publication here. In particular, both referees #1 and #2 requests additional support for the proposed role of KLRF1 activation in CD8⁺ T cell activation, which is a very important point to address.

I find the reports informed and constructive, and believe that addressing the concerns raised will significantly strengthen the manuscript. As the reports are below, and I think all points need to be addressed, I will not detail them here. Please contact me if you have questions or comments regarding the revision for further discussion (also by video chat).

Given these recommendations, we would like to invite you to revise your manuscript with the understanding that the referee concerns (as in their reports) must be fully addressed and their suggestions taken on board. Please address all referee concerns in a complete point-by-point response. Acceptance of the manuscript will depend on a positive outcome of a second round of review. It is EMBO reports policy to allow a single round of major experimental revision only and acceptance or rejection of the manuscript will therefore depend on the completeness of your responses included in the next, final version of the manuscript.

We realize that it is difficult to revise to a specific deadline. In the interest of protecting the conceptual advance provided by the work, we recommend a revision within 3 months. Please discuss the revision progress ahead of this time with me if you require more time to complete the revisions, or if you have questions or comments regarding the revision (also by video chat).

1. A data availability section providing access to data deposited in public databases is missing (where applicable).
2. Your manuscript contains statistics and error bars based on $n=2$. Please use scatter plots in these cases.

You can submit the revision either as a Scientific Report or as a Research Article. For Scientific Reports, the revised manuscript can contain up to 5 main figures and 5 Expanded View figures, and it should not exceed 27000 characters. If the revision leads to a manuscript with more than 5 main figures it will be published as a Research Article. In this case the Results and Discussion section should be separate. If a Scientific Report is submitted, these sections have to be combined. This will help to shorten the manuscript text by eliminating some redundancy that is inevitable when discussing the same experiments twice. In either case, all materials and methods should be included in the main manuscript file.

4) a .docx formatted letter INCLUDING the reviewers' reports and your detailed point-by-point responses to their comments. As part of the EMBO publication's Transparent Editorial Process, EMBO reports publishes online a Review Process File (RPF) to

accompany accepted manuscripts. This File will be published in conjunction with your paper and will include the referee reports, your point-by-point response and all pertinent correspondence relating to the manuscript.

<https://www.embopress.org/page/journal/14693178/authorguide#transparentprocess>

5) a complete author checklist, which you can download from our author guidelines

<https://www.embopress.org/page/journal/14693178/authorguide>. Please insert information in the checklist that is also reflected in the manuscript. The completed author checklist will also be part of the RPF.

6) Please note that all corresponding authors are required to supply an ORCID ID for their name upon submission of a revised manuscript (<<https://orcid.org/>>). Please find instructions on how to link your ORCID ID to your account in our manuscript tracking system in our Author guidelines

<<https://www.embopress.org/page/journal/14693178/authorguide#authorshipguidelines>>

7) Before submitting your revision, primary datasets produced in this study need to be deposited in an appropriate public database (see <https://www.embopress.org/page/journal/14693178/authorguide#datadeposition>). Please remember to provide a reviewer password if the datasets are not yet public. The accession numbers and database should be listed in a formal "Data Availability" section placed after Materials & Method (see also

<https://www.embopress.org/page/journal/14693178/authorguide#datadeposition>). Please note that the Data Availability Section is restricted to new primary data that are part of this study. * Note - All links should resolve to a page where the data can be accessed. *

Additional information on source data and instruction on how to label the files are available:

<https://www.embopress.org/page/journal/14693178/authorguide#sourcedata>

9) Our journal encourages inclusion of *data citations in the reference list* to directly cite datasets that were re-used and obtained from public databases. Data citations in the article text are distinct from normal bibliographical citations and should directly link to the database records from which the data can be accessed. In the main text, data citations are formatted as follows: "Data ref: Smith et al, 2001" or "Data ref: NCBI Sequence Read Archive PRJNA342805, 2017". In the Reference list, data citations must be labeled with "[DATASET]". A data reference must provide the database name, accession number/identifiers and a resolvable link to the landing page from which the data can be accessed at the end of the reference. Further instructions are available at <http://www.embopress.org/page/journal/14693178/authorguide#referencesformat>

10) Regarding data quantification (see Figure Legends:

<https://www.embopress.org/page/journal/14693178/authorguide#figureformat>)

11) The journal requires a statement specifying whether or not authors have competing interests (defined as all potential or actual interests that could be perceived to influence the presentation or interpretation of an article). In case of competing interests, this must be specified in your disclosure statement. Further information: <https://www.embopress.org/competing->

interests

12) Please also note our reference format:

13) All Materials and Methods need to be described in the main text using our 'Structured Methods' format, which is required for all research articles. According to this format, the Methods section includes a Reagents and Tools Table (listing key reagents, experimental models, software and relevant equipment and including their sources and relevant identifiers) followed by a Methods and Protocols section describing the methods using a step-by-step protocol format. The aim is to facilitate adoption of the methodologies across labs. More information on how to adhere to this format as well as a downloadable template (.docx) for the Reagents and Tools Table can be found in our author guidelines:

I look forward to seeing a revised version of your manuscript when it is ready. Please let me know if you have questions or comments regarding the revision.

Kind regards,

Deniz

Deniz Senyilmaz Tiebe, PhD

Senior Scientific Editor

EMBO Reports

Referee #1:

This article reports that the surface molecule KLRF1 marks human CD8 T cells that are at an advanced stage of differentiation where they possess limited/diminishing effector functions. It is shown that CD8 T cells expressing KLRF1 are associated with solid tumors, but that such cells preferentially reside in stromal compartments outside the tumor parenchyma. Moreover, KLRF1+ CD8 T cells that associate with solid tumors have certain characteristics that may suggest that they are precursor exhausted cells, a cell type with stem cell like capacity that is thought to be responsible for responsiveness to checkpoint inhibition. Evidence is provided, finally, to support the idea that CD4 T cells may provide direct help to such CD8 T cells via activating KLRF1.

The manuscript contains a wealth of data, some of it from cutting edge experimental techniques, including Imaging Mass Cytometry, and cleverly mines existing single cell RNAsequencing data sets. Much of the analysis is well done. In principle, the identification of the KLRF1+ differentiation stage, and especially the possibility that CD4 T cells help CD8 T cells via this molecule, is of interest to the immunological community.

Major comments:

1. I am not fully sold on the idea that CD4 T cells provide help to KLRF1+ CD8 via activation of that surface molecule. I am convinced by the data that a subset of CD4+ T cells expresses the ligand for KLRF1 (CLEC2B/AICL) and that these cells are found in proximity of KLRF1+ CD8 T cells in tumor associated tissue. However, the functional data to support this relationship (in Figure 2) are not fully convincing in my view. The argument given by the authors is that antibody mediated blockade of KLRF1 results in (modestly) reduced production of IFN γ and TNF by in vitro stimulated KLRF1+ CD8 T cells, specifically when CD4 T cells are present in the culture (Fig 2b), but not when these cells are absent (Fig 2c). The problem I have with this interpretation is that the difference between the effect in Figure 2b and 2c is not that pronounced and may be influenced by the fact that in Figure 2c much fewer donors were tested. The fact that the difference in cytokine production in figure 2C is not significant does not mean that there is no effect. I would say there is a trend towards inhibition of cytokine production in Figure 2c as well. However, here only 5 donors were examined, much less than were tested in figure 2B. Perhaps therefore 1 outlier has an outside effect in Figure 2C making the effect seem non-significant. I wonder if the experimental set up cannot be improved to generate more convincing data. The proportion of CD4 T cells that expresses CLEC2B/AICL is very low (1%), implying that most of the KLRF1+ CD8 T cells probably are not "found" by CLEC2B/AICL+ CD4 T cells. The experiment might be more robust if some kind of enrichment could be done. I would imagine that MACS isolation of the relevant populations might be possible without blocking all KLRF1 and CLEC2B/AICL, if the antibodies to the molecules are first coupled to the beads (only a small number of beads is required to select a cell, leaving most of the molecules free for interaction).

Although the finding that CLEC2B/AICL⁺ CD4⁺ T cells are found in the vicinity of KLRF1⁺ CD8⁺ T cells is interesting, the manuscript seems to ignore the fact that this ligand is expressed by many other cells, such as myeloid cells and, in fact, activated KLRF1⁺ CD8⁺ T cells themselves (according to the expression data in SI Figure 4 and in Figure 5b-the split CD8 positive population also expresses CLEC2B/AICL). Does the imaging Mass Cytometry show that CD4⁺ T cells are the only cells expressing CLEC2B/AICL in the vicinity of KLRF1⁺ CD8⁺ T cells?

Minor comments:

2. The use of arrows in figure 1B seems to indicate a developmentally progressive relationship between cells with these even phenotypes, which may or may not have a basis in reality. It might be better to be less committal about this.
3. The conclusion that KLRF1 might exert supportive function for memory-like late differentiated CD8⁺ T cells in maintaining cytokine production seems a little odd, given that KLRF1 acquisition is associated with reduced cytokine production in Fig 1C.
4. Why are gates put so high on the KLRs in 1A?
5. Were total PBMC stimulated or total T cells or isolated CD8 in Fig 1c? I could not find this information.
6. I am confused about the experimental set up in Figure 2c. According to the legend, anti KLRF1 was plate bound. Does that trigger the receptor? However, according to text, anti KLRF1 is supposed to block. I would expect the antibody to be added solubly then. Also, the way I understand the manuscript, the idea is that KLRF1⁺ CD8⁺ T cells are helped by CD4⁺ T cells to produce cytokines. To support that, you would then want to examine expression of cytokines by KLEF1⁺ CD8⁺ T cells. However, in this experiment, those cells cannot be identified anymore as the receptor is blocked (presumably) by the antibody. Therefore, the anti KLRF1⁺ antibody could theoretically reduce cytokine production by KLRF1⁻ CD8⁺ T cells. It seems to me that this experimental set up is not optimal to test the hypothesis.
7. Furthermore, percentages may not be the best reflection of the effect of KLRF1 blockade in Figure 2c. How about MFI? If the receptor promotes production of these cytokines, perhaps you get more expression per cell, rather than more cells producing.
8. The following phrases seem overstatements: "Interestingly, GPR56 appears to negatively regulate AICL upregulation in CD4⁺ T cells," and : "GPR56 appears to negatively influence this process" (both page 12). The only justified conclusion is that there is a negative correlation between expression of the two.
9. Figure 3. The percentages in 3C are difficult to interpret. What proportion of the gated populations expresses these markers? This is important to interpret the effects from this ligand on the KLRF1⁺ CD8⁺ T cells. Also, example gatings would be helpful to be able to judge the quality of the results.
10. Figure 4F does seem to show some differences in expression, and it may be significant statistically, but is it also significant biologically? I have to admit that I find the value on the Y axis difficult to interpret.
11. There is an incorrect reference in the second paragraph on page 24 to supplementary image figure 6A. This does not exist. I think this should be supplementary figure 3A.
12. I have some doubt about whether the KLRF1⁺ CD8⁺ T cells truly are T_{pex} cell as they seem to lack expression of TOX, according to SI Fig 4.

Referee #2:

The manuscript submitted by Barone et al reports that terminally differentiated effector memory CD8⁺ T cells upregulate KLRF1 expression, enabling CD8-CD4 T cell interaction through the KLRF1-AICL axis that allow CD8⁺ T cells retain their capacity to produce cytokines like IFN γ and TNF α . Integrated spatial proteomics and single-cell transcriptomics studies of lung adenocarcinoma and adjacent tissues reveal that KLRF1⁺ CD8⁺ T cells exhibit T_{pex} gene signature, and that localized interactions between AICL⁺ CD4⁺ and KLRF1⁺ CD8⁺ T cells are identified in non-tumor regions but rare in tumor cores. The report is interesting because it may identify additional diagnostic markers to distinguish different subsets of exhausted T cells, and uncover AICL-KLRF1 as a potential therapeutic target to facilitate CD4-CD8 interactions and enhance CD8⁺ T cell functionality. However, a number of issues need to be addressed to enhance the rigor of the report.

1. The report is largely descriptive in nature. The major findings, like CD4⁺ T Cells promote IFN- γ and TNF production by CD8⁺ T Cells through AICL-KLRF1 axis, and direct CD4-CD8 interactions through AICL-KLRF1 prevent CD8 T cell exhaustion, are based on correlative studies. Overall, the data shown are indicative but not definitive.
2. In Fig. 2, a blocking anti-KLRF1 antibody was used to neutralize KLRF1 in CD8 T cells. However, the anti-KLRF1 antibody was plate-coated during T cell culture. The loss of KLRF1 on CD8 T cells seemed to be due to internalization of surface KLRF1 upon engagement with coated anti-KLRF1 antibody. This experimental setting is different from the typical Ab neutralization experiment in which the neutralizing Ab is in free form. The plate-coated Ab may cross-link KLRF1 expressed on the surface CD8 T cells, causing signal transmission similar to what AICL would do. The results should be validated by a different approach, for example, use of soluble aKLRF1 Ab, blocking of AICL, knockdown of KLRF1 or AICL, etc. In addition, the differences seen in the % of IFN γ +TNF α + CD8 T cells between the isotype and aKLRF1 conditions are really subtle (median ~10% vs ~8% at 6 hr), questioning whether there is any biological significance.
3. The claim that KLRF1+CD8⁺ T cells are T_{pex} seemed to be solely dependent on gene signature derived from scRNAseq data. It is known that the spectrum of T cell exhaustion is critically defined by epigenetic landscapes. Without epigenetic profiling

of the KLRF1+ CD8 T cells, it is premature to define these cells as Tpex cells.

4. Some recent studies demonstrate that even terminally exhausted CD8 T cells can express effector molecules such as IFN γ , TNF α and Granzyme B, even at levels higher than Tpex cells. The authors should at least include this pool of literature to discuss the discrepancies.

Minor point:

In Fig. 1B, it is confusing to see the arrows between the dot plots. Please specify whether the arrows indicate the sequential expression of the markers by CD8 T cells upon stimulation or are arbitrarily placed by the authors for comparison purpose.

Referee #3:

This is a carefully executed study showing that CD4+ T cells contribute to the maintenance of pre-exhausted CD8+ T cells through interaction of AICL with KLRF1. Moreover the authors elegantly demonstrate that this interaction fails in tumor tissue.

I only have some minor details, primarily on the references. I noticed them as they refer to my own papers and propose that the authors cross-check other references as well.

- Fig1.B-D shows cytokine production. Please, clarify in text and legend whether/how cells have been stimulated.
- We already showed activation-induced, transient expression of AICL in T cells (hence its name) in Hamann et al, Immunogenetics 1997.
- Define NKC
- The reference in the text to SI Fig.6A should be corrected into Fig.6A (without SI).
- The key paper for GPR56 as checkpoint is Chang et al, Cell Rep 2016 (PMID: 27184850). For the full history, see Hsiao et al, BCPT 2023 (PMID 36750420).
- The sentence in the Discussion about AICL cites Vitale et al, EJI 2021. I assume this should be Welte et al, Nat Commun 2006.

Point by Point Reply: EMBOR-2025-61822-T

We would like to thank all reviewers for their time and the effort spent on reviewing our work. Their insightful comments and valuable suggestions have helped us to further strengthen our study. We addressed the reviewer's criticism, have added new data and adapted the manuscript text following the reviewers' advice.

In brief, we have added the following data and made the following changes:

- Expanded functional validation of the AICL-KLRF1 axis
 - Inclusion of *soluble* anti-KLRF1 blocking antibody experiments (Figure 2B).
 - Inclusion of *sorted AICL⁺ vs. AICL⁻ CD4⁺ T cell* co-culture experiments (new Figure 3D-F).
- Increased sample size in key neutralization experiments (now n=10 in both Figure 2B and Figure 2C), ensuring equal statistical power.
- Addition of multimodal scRNA-seq/ATAC re-analysis (new Figure EV 6), supporting that CD8⁺ KLRF1⁺ T cells exhibit a more T_{pex}-like chromatin landscape.
- Restructuring and textual refinement, including rephrasing statements about the role of GPR56

Reviewer's Comments:

Referee #1:

This article reports that the surface molecule KLRF1 marks human CD8 T cells that are at an advanced stage of differentiation where they possess limited/diminishing effector functions. It is shown that CD8 T cells expressing KLRF1 are associated with solid tumors, but that such cells preferentially reside in stromal compartments outside the tumor parenchyma. Moreover, KLRF1⁺ CD8 T cells that associate with solid tumors have certain characteristics that may suggest that they are precursor exhausted cells, a cell type with stem cell like capacity that is thought to be responsible for responsiveness to checkpoint inhibition. Evidence is provided, finally, to support the idea that CD4 T cells may provide direct help to such CD8 T cells via activating KLRF1.

The manuscript contains a wealth of data, some of it from cutting edge experimental techniques, including Imaging Mass Cytometry, and cleverly mines existing single cell RNAsequencing data sets. Much of the analysis is well done. In principle, the identification of the KLRF1⁺ differentiation stage, and especially the possibility that CD4 T cells help CD8 T cells via this molecule, is of interest to the immunological community.

Major comments:

1. *I am not fully sold on the idea that CD4 T cells provide help to KLRF1⁺ CD8 via activation of that surface molecule. I am convinced by the data that a subset of CD4⁺ T cells expresses the ligand for KLRF1 (CLEC2B/AICL) and that these cells are found in proximity of KLRF1⁺ CD8 T cells in tumor associated tissue. However, the functional*

data to support this relationship (in Figure 2) are not fully convincing in my view. The argument given by the authors is that antibody mediated blockade of KLRF1 results in (modestly) reduced production of IFN γ and TNF by in vitro stimulated KLRF1+ CD8 T cells, specifically when CD4 T cells are present in the culture (Fig 2b), but not when these cells are absent (Fig 2c). The problem I have with this interpretation is that the difference between the effect in Figure 2b and 2c is not that pronounced and may be influenced by the fact that in Figure 2c much fewer donors were tested. The fact that the difference in cytokine production in figure 2C is not significant does not mean that there is no effect. I would say there is a trend towards inhibition of cytokine production in Figure 2c as well. However, here only 5 donors were examined, much less than were tested in figure 2B. Perhaps therefore 1 outlier has an outsize effect in Figure 2C making the effect seem non-significant. I wonder if the experimental set up cannot be improved to generate more convincing data. The proportion of CD4 T cells that expresses CLEC2B/AICL is very low (1%), implying that most of the KLRF1+ CD8 T cells probably are not "found" by CLEC2B/AICL+ CD4 T cells. The experiment might be more robust if some kind of enrichment could be done. I would imagine that MACS isolation of the relevant populations might be possible without blocking all KLRF1 and CLEC2B/AICL, if the antibodies to the molecules are first coupled to the beads (only a small number of beads is required to select a cell, leaving most of the molecules free for interaction).

Response:

We appreciate the reviewer's careful evaluation of the functional data. In response, we implemented two major experimental improvements, each of which substantially strengthens the mechanistic conclusion that CD4⁺ AICL⁺ T cells support cytokine production in CD8⁺ KLRF1⁺ T cells.

1. Balanced donor numbers across experimental conditions

Previously, Figure 2C included only 5 donors, limiting statistical comparison with Figure 2B. We therefore repeated all experiments (see Rebuttal figure 1 below), resulting in:

- Figure 2B (CD3⁺ T-cell co-culture): n = 10 donors
- Figure 2C (CD8⁺ T cells alone): n = 10 donors

After equalizing the sample size KLRF1 blockade consistently reduced IFN- γ /TNF production in co-cultures containing CD4⁺ T cells, but no significant reduction was observed when CD8⁺ T cells were stimulated without CD4⁺ T cells.

This reinforces the original conclusion that CD4⁺ T cells are required for KLRF1-dependent enhancement of cytokine production.

*Rebuttal Figure 1: Updated KLRF1 neutralization data corresponding to Figure 2B–C. (B) CD3⁺ T-cell co-cultures were stimulated for 6h or 24h and treated with either plate-bound or soluble anti-KLRF1 antibody or the respective isotype controls. Post-hoc paired Wilcoxon signed-rank test was performed with Benjamini & Hochberg adjusted p values across all comparisons and the relevant comparisons are shown in the plot: * p < 0.05, ** p < 0.01 (C) In parallel, isolated CD8⁺ T cells were stimulated under the plate-bound anti-KLRF1 or isotype conditions. Shown are the percentages of CD8⁺ KLRF1⁺ T cells and TNF⁺/IFN- γ ⁺ CD8⁺ T cells. Tested for normality using the Shapiro-Wilk test, followed by a t-test to compare the two groups: *** p < 0.001. All conditions in panels B&C were repeated to achieve n = 10 donors per subgroup.*

2. AICL⁺ vs. AICL⁻ CD4⁺ T cell sorting experiment (new Figure 3D&E, see Rebuttal figure 2 below)

To further strengthen the functional evidence for a CD4⁺ T cell–derived AICL signal and to directly address the reviewer’s concern regarding the low frequency of CD4⁺ AICL⁺ T cells, we performed an enrichment experiment in which we FACS-sorted AICL-expressing versus

AICL-negative CD4⁺ T cell subsets and tested their ability to support cytokine production by CD8⁺ T cells.

Experimental design

We first enriched total CD3⁺ T cells from PBMC. Pre-gating on KLRG1 and / or KLRF1 expressing CD4⁺ T helper cells, cells were then sorted into two distinct subsets according to surface AICL expression:

- CD4⁺ KLRG1⁺ and/or KLRF1⁺ AICL⁺ T helper cells, and
- CD4⁺ KLRG1⁺ and/or KLRF1⁺ AICL⁻ T helper cells, which served as a AICL negative control population matched for other differentiation state-specific surface markers.

Sorted CD4⁺ subsets were subsequently co-cultured at a 1:20 ratio with autologous CD8⁺ T cells. In parallel, we included control conditions consisting of CD8⁺ T cells alone to allow direct comparison of cytokine outcomes.

Results

Despite being phenotypically equivalent in terms of differentiation markers (KLRG1 and KLRF1), only the AICL expressing CD4⁺ T helper cells were able to promote a robust increase in cytokine production by CD8⁺ T cells (see Rebuttal Figure 2A).

Specifically:

- CD8⁺ T cells co-cultured with CD4⁺ AICL⁺ T cells showed a trend towards an increase in the frequency of total TNF or IFN- γ positive cells after 24 hours of stimulation in comparison to CD8⁺ T cells cultured alone (Rebuttal Figure 2B). This was largely due to a significant increase in TNF positive CD8⁺ T cells (Rebuttal Figure 2C).
- In contrast, co-culture with CD4⁺ AICL⁻ T cells failed to enhance TNF or IFN- γ production and resulted in significantly lower proportions of total TNF or IFN- γ expressing CD8⁺ T cells compared to co-cultures with CD4⁺ AICL⁺ T cells (Rebuttal Figure 2B).

This enrichment experiment provides causal evidence for the AICL–KLRF1 interaction between CD4⁺ and CD8⁺ T cells, respectively. Furthermore, the helper effect is not a general property of differentiated CD4⁺ T helper cells but is dependent on the presence of AICL.

These newly performed experiments and their corresponding findings have been incorporated into the revised manuscript and are presented on pages 10–11 as well as in Figure 3.

Rebuttal Figure 2: AICL⁺ versus AICL⁻ CD4⁺ T cell sorting and co-culture experiment corresponding to revised Figure 3D–F. CD3⁺ T cells were enriched from PBMC, and differentiated CD4⁺ T cells were sorted into AICL⁺ KLRG1⁺/KLRF1⁺ and AICL⁻ KLRG1⁺/KLRF1⁺ subsets (AICL⁻ G1⁺/F1⁺). Sorted CD4⁺ populations were co-cultured with autologous CD8⁺ T cells at a 1:20 ratio for 24h alongside a CD8-only control. (A) Schematic of sorting and co-culture workflow. (B) CD8⁺ TNF⁺/IFN-γ⁺ T cells expressed as fold change relative to the CD8-only condition. (C) CD8⁺ TNF⁺ T cells analyzed analogously. Statistical comparisons were performed using one-tailed t-tests with Benjamini–Hochberg correction.

2. Although the finding that CLEC2B/AICL⁺ CD4 T cells are found in the vicinity of KLRF1⁺ CD8⁺ T cells is interesting, the manuscript seems to ignore the fact that this ligand is expressed by many other cells, such as myeloid cells and, in fact, activated KLRF1⁺ CD8 T cells themselves (according to the expression data in SI Figure 4 and in Figure 5b-the split CD8 positive population also expresses CLEC2B/AICL). Does the imaging Mass Cytometry show that CD4⁺ T cells are the only cells expressing CLEC2B/AICL in the vicinity of KLRF1⁺ CD8⁺ T cells?

Response:

We agree with the reviewer that AICL is expressed by multiple cell types, including monocytes, macrophages, neutrophils, and NK cells (Hamann *et al.*, 1997). Therefore, it was essential to determine which AICL-expressing cells are present in our IMC dataset and, crucially, which of these are spatially positioned to interact with CD8⁺ KLRF1⁺ T cells in human lung tissue.

To address this, we performed a comprehensive re-analysis of the IMC data, now included in Supplementary Figure 3, using a Boolean gating strategy to partition all cells into four major compartments (see Rebuttal Figure 3A):

- Gate 1: CD19⁺ B cells
- Gate 2: CD19⁻CD3⁺ T cells
- Gate 3: CD19⁻CD3⁻CD45⁺ non-B/non-T immune cells (myeloid lineage, ILCs, NK-like populations)
- Gate 4: CD19⁻CD3⁻CD45⁻ non-immune stromal and epithelial cells

This allowed us to systematically quantify AICL expression and distances to CD8⁺ KLRG1⁺ KLRF1⁺ T cells in every immune compartment and to directly compare them to the AICL expression and spatial proximity of the CD4⁺ AICL⁺ T cell cluster identified earlier.

We found that B cells (Gate 1) constituted only a minute fraction of total cells, and were present at detectable abundance in only 1 out of 10 ROIs across the five patient samples analyzed (Rebuttal Figure 3B). Given their rarity and negligible contribution to overall AICL expression as described by Hamann et al., they were excluded from downstream evaluation.

AICL expression was highest in the T cell compartment (Gate 2), particularly in the CD4⁺ AICL⁺ T cell cluster (Rebuttal Figure 3C). All myeloid, NK-like, and ILC-like clusters (Gate 3) exhibited consistently lower AICL intensity than CD4⁺ AICL⁺ T cells. Non-immune stromal and epithelial cells (Gate 4) showed negligible AICL expression. Thus, no other immune or non-immune population reached the AICL levels observed in CD4⁺ T cells, making them the dominant AICL⁺ subset in our lung tissue IMC dataset.

Rebuttal Figure 3: Boolean gating strategy and AICL expression across major cellular compartments in the IMC dataset. (A) Gating used to define CD19⁺ B cells (Gate 1), CD19⁻CD3⁺ T cells (Gate 2), CD19⁻CD3⁻CD45⁺ non-B/non-T immune cells (Gate 3), and CD19⁻CD3⁻CD45⁻ non-immune cells (Gate 4). (B) Fraction of B cells in each region of interest (green). Gates 2–4 are shown collectively (violet). ROIs are ordered by tissue type. (C) Mean AICL expression per IMC cluster within each gated subset. In Gate 2, the CD4⁺ AICL⁺ T cell cluster is highlighted by an enlarged dot.

To further characterize the non-B/non-T immune compartment (Gate 3), we performed unsupervised clustering followed by manual annotation based on established lineage

markers available in our IMC panel. This analysis revealed that Gate 3 consisted of two major immunological groups, along with several unassignable clusters (see heatmap in Rebuttal Figure 4A):

1. ILC- and NK-like clusters

These clusters were defined by high expression of CD25 and CD127, combined with absence of CD68 or CD11c. This phenotype is characteristic of innate lymphoid populations and NK-like cells. In the heatmap (Figure EV 2 / Rebuttal Figure 4A), these clusters are highlighted by a blue bar beneath the z-scored expression profile.

2. Myeloid clusters (monocyte-, macrophage-, and DC-like)

These clusters showed high CD68, CD11c, and/or HLA-DR expression and lacked CD25 and CD127, consistent with monocytes, macrophages, and dendritic cells. These clusters are marked by a red bar beneath the heatmap.

3. “Other” immune cell clusters

Several clusters displayed low or mixed expression of the above markers. Due to the limited marker set available in the IMC panel (e.g., absence of dedicated granulocyte markers), these clusters could not be reliably assigned to a known immune lineage and were therefore annotated conservatively as “other”, indicated by a gray bar beneath the heatmap.

The corresponding UMAP representation (Rebuttal Figure 4B) illustrates that these annotated groups form separated phenotypic islands.

We next assessed whether any of these Gate 3 clusters could serve as plausible AICL-expressing partners for CD8⁺ KLRF1⁺ T cells. For this, we quantified both AICL expression levels and median spatial distances of each cluster to the CD8⁺ KLRF1⁺ T cell cluster (Rebuttal Figure 4C). Inclusion of the CD4⁺ AICL⁺ T cell cluster in the same analysis served as a biologically validated reference for comparison.

This analysis showed that all Gate 3 clusters expressed markedly lower levels of AICL than CD4⁺ AICL⁺ T cells (a pattern already evident in Rebuttal Figure 3C), none of the Gate 3 clusters were spatially close to the CD8⁺ KLRF1⁺ T cell cluster, and importantly, the myeloid clusters that were physically closest to CD8⁺ KLRF1⁺ T cells (e.g., clusters 311 and 319) paradoxically displayed the very lowest AICL expression of all Gate 3 clusters.

Thus, Gate 3 populations not only fail to match CD4⁺ AICL⁺ T cells in ligand abundance, but they also lack the necessary spatial proximity for meaningful ligand–receptor interaction. In contrast, the CD4⁺ AICL⁺ T cell cluster uniquely combines high AICL expression with direct spatial adjacency to CD8⁺ KLRF1⁺ T cells.

Taken together, the phenotypic annotation of Gate 3 and the combined expression–distance analyses provide compelling evidence that myeloid and innate lymphoid/NK-like cells are unlikely to contribute to KLRF1 engagement in situ, and that CD4⁺ AICL⁺ helper T cells are the predominant AICL source relevant for CD8⁺ KLRF1⁺ T cell support in non-tumor lung tissue.

Rebuttal Figure 4: Spatial analysis of non-B/non-T immune cells (Gate 3) in relation to CD8⁺ KLRF1⁺ T cells. (A) Z-scored heatmap of Gate 3 clusters. ILC-/NK-like clusters (CD25⁺CD127⁺CD68⁻) are annotated by dark-blue bars; monocyte-, macrophage-, and DC-like clusters (CD25⁻CD127⁻CD68⁺) by red bars; unassignable clusters by gray bars. (B) UMAP embedding of Gate 3 showing separation of ILC/NK-like versus myeloid clusters. (C) Left: median spatial distance of each Gate 3 cluster to the CD8⁺ KLRF1⁺ T cell cluster, with the CD4⁺ AICL⁺ T cell cluster included as a reference (first cluster in each plot). Right: mean AICL expression per cluster. Vertical dashed lines indicate the median distance (left) and median AICL expression (right) of the CD4⁺ AICL⁺ T cell cluster.

The analysis shown in Rebuttal Figures 3 and 4 has been incorporated into Figure EV 2 and Appendix Figure S2, and the revised manuscript now includes dedicated paragraphs explaining the gating, clustering, and annotation of the relevant populations. To ensure clarity, the paragraph begins by describing the Boolean gating strategy (Page 13):

“We applied a Boolean gating strategy to the IMC dataset to separate CD19⁺ B cells (Gate 1), CD19⁻ CD3⁺ T cells (Gate 2), CD19⁻ CD3⁻ CD45⁺ non-B/non-T immune cells (Gate 3), and CD19⁻ CD3⁻ CD45⁻ non-immune stromal and epithelial cells (Gate 4, Fig. EV 2A). We first checked AICL expression in each gated population and found that T cells produced significantly more AICL compared to non-T immune cells or CD45⁻ cells (Fig. EV 2B). Because B cells were extremely rare and detected at appreciable frequencies in only one of the ten analyzed ROIs, they were excluded from further analysis”

The next paragraphs then explain how Gate 3 was annotated after clustering and finally reports the spatial and expression analyses and conclusions (Pages 16 and 17):

“AICL is expressed by multiple cell types, including monocytes, macrophages, neutrophils and NK cells (Hamann *et al.*, 1997). To identify potential sources of AICL apart from the split CD4⁺ G⁺F⁺A⁺ T cell cluster within the tissue microenvironment, we performed unsupervised clustering on the CD19⁻ CD3⁻ CD45⁺ non-B/non-T immune compartment (Gate 3). Cluster annotation was based on the expression patterns of lineage-relevant markers available in our IMC panel: clusters with high CD25 and CD127 but lacking CD68 and CD11c were annotated as ILC- or NK-like populations, while clusters expressing CD68, CD11c, and/or HLA-DR and lacking CD25 and CD127 were annotated as monocyte-, macrophage-, or dendritic cell-like populations. A small number of clusters could not be reliably assigned due to limited marker resolution and were therefore designated as ‘other’ (Fig. EV 2C). The UMAP embedding showed these annotated groups as well-separated phenotypic islands, confirming the validity of the clustering and annotation approach.

To evaluate whether any of these non-T immune populations might serve as AICL-expressing interaction partners for CD8⁺ KLRF1⁺ T cells, we quantified both AICL expression and median spatial distances for each Gate 3 cluster relative to the split CD8⁺ G⁺F⁺A⁺ cluster (Fig. EV 2G). This analysis demonstrated that none of the non-T immune clusters expressed AICL at levels comparable to the split CD4⁺ G⁺F⁺A⁺ T cell cluster, and all were positioned considerably farther away from the split CD8⁺ G⁺F⁺A⁺ T cells. Notably, the myeloid clusters closest to CD8⁺ KLRF1⁺ T cells (clusters 311 and 319) displayed the lowest AICL expression of all Gate 3 clusters. Cluster 321 did not co-localize on the same ROI with any of the split CD8⁺ G⁺F⁺A⁺ T cells and is therefore missing in the distance analysis. Together, these findings indicate that CD4⁺ AICL⁺ T cells are the predominant AICL-expressing population in spatial proximity to CD8⁺ KLRF1⁺ T cells in non-tumor lung tissue.”

Minor comments:

1. *The use of arrows in figure 1B seems to indicate a developmentally progressive relationship between cells with these even phenotypes, which may or may not have a basis in reality. It might be better to be less committal about this.*

Response:

We fully agree that the arrows in the original version of Figure 1B could inadvertently suggest a sequential developmental trajectory, which was not experimentally tracked in our study. To avoid this misinterpretation, we removed all arrowheads between the different phenotypically defined subpopulations. We also added clarifying text to the figure legend explicitly stating that these subsets represent phenotype-based categories, not reconstructed differentiation pathways. This ensures that readers will not infer a directional developmental model from these visual cues.

For convenience, the panel is shown here:

Rebuttal Figure 5: Updated version of **Figure 1B**. Arrows have been removed to avoid implying a developmental sequence.

- The conclusion that *KLRF1* might exert supportive function for memory-like late differentiated CD8 T cells in maintaining cytokine production seems a little odd, given that *KLRF1* acquisition is associated with reduced cytokine production in Fig 1C.

Response:

We thank the reviewer for this observation.

Terminally differentiated CD8⁺ T cells (T_{EMRA}) generally show reduced cytokine production compared to earlier memory subsets. It is important to note, however, that the TEMRA compartment is functionally heterogeneous, comprising both cytokine-poor and cytokine-competent cells, as supported by our data and previous work.

KLRF1 expression increases progressively with differentiation and is most frequent within T_{EMRA} cells. This explains why, when viewing all CD8⁺ T cell subsets together, higher *KLRF1* prevalence coincides with lower cytokine production, reflecting the overall decline in effector function that accompanies terminal differentiation rather than a direct impact of *KLRF1* itself.

At the same time, our functional perturbation experiments show that blocking *KLRF1* signaling leads to a reduction in TNF and IFN-γ production in CD8⁺ T cells. This demonstrates that endogenous *KLRF1* engagement contributes to maintaining the residual cytokine-producing capacity present in this compartment.

Taken together, these findings indicate that while T_{EMRA} cells as a whole produce fewer cytokines, *KLRF1* signaling plays a supportive role in sustaining cytokine output within this late-differentiated pool. We have clarified this point in the revised manuscript to avoid potential misinterpretation.

- Why are gates put so high on the KLRs in 1A?

Response:

We thank the reviewer for this comment. We assume that the question refers mainly to the upper inset gate in the KLRB1 plot, which is higher than the other KLR gates in Figure 1A. This gate was intentionally positioned to identify and exclude KLRB1^{high} mucosal-associated invariant T (MAIT) cells, which represent a distinct T cell lineage and would otherwise confound analysis of KLR expression within conventional CD8⁺ T cells.

The placement of this MAIT-specific gate is supported by the data shown in the Figure EV 1A, where we performed co-staining for the invariant TCR chain Va7.2—a canonical MAIT

cell marker. In this figure, the V α 7.2⁺ cells align precisely with the KLRB1^{high} population, confirming that the upper KLRB1 gate in Figure 1A captures MAIT cells rather than conventional CD8⁺ T cells.

In contrast, the gates for the other KLR receptors (KLRG1, KLRF1, and GPR56) were set between clearly negative and clearly positive events, and these gates are not positioned unusually high. Their placement reflects bimodal expression patterns and follows standard gating practice to cleanly isolate positive populations.

To avoid confusion, we have clarified this in the revised figure legend, explicitly noting that the upper KLRB1^{high} gate is used for MAIT cell exclusion, while all other KLR gates are drawn at the midpoint between negative and positive subsets. We have also annotated the MAIT cells in Figure 1A and Figure EV 1A to better guide the reader.

4. *Were total PBMC stimulated or total T cells or isolated CD8 in Fig 1c? I could not find this information.*

Response:

We thank the reviewer for pointing this out. The cells analyzed in Figure 1C were derived from CD3⁺-enriched PBMCs, from which CD8⁺ T cells were subsequently gated. To make this explicit, we have updated the Figure 1A legend to read: **“Exemplary dot plots of pre-gated CD8⁺ T cells from unstimulated CD3⁺-enriched PBMCs.”**. This wording now clearly specifies the source and preparation of the cells used in Figure 1A–C.

5. *I am confused about the experimental set up in Figure 2c. According to the legend, anti **KLRF1 was plate bound**. Does that trigger the receptor? However, according to text, anti KLRF1 is supposed to block. I would expect the antibody to be **added solubly then**. Also, the way I understand the manuscript, the idea is that KLRF1+ CD8 T cells are helped by CD4 T cells to produce cytokines. To support that, you would then want to examine expression of cytokines by KLEF1+ CD8 T cells. However, in this experiment, those cells cannot be identified anymore as the receptor is blocked (presumably) by the antibody. Therefore, the anti KLRF1+ antibody could theoretically reduce cytokine production by KLRF1- CD8 T cells. It seems to me that this experimental set up is not optimal to test the hypothesis.*

Response:

We thank the reviewer for raising these important points. Following this suggestion, we performed additional neutralization experiments using soluble anti-KLRF1 antibody to complement the original plate-bound approach. This now allows a comparison of the two blocking strategies.

Using soluble anti-KLRF1 (N = 10 donors at 6 hours and 24 hours; see Rebuttal Figure 1), we observed that surface KLRF1 is already significantly reduced at 6 hours, and remains reduced at 24 hours, although in both cases the magnitude of down-modulation is lower than with plate-bound antibody. In contrast, plate-bound anti-KLRF1 induces a more pronounced and rapid loss of detectable KLRF1 due to antibody cross-linking and receptor internalization.

Importantly, the **functional readout parallels the degree of receptor modulation**:

- under plate-bound conditions, where KLRF1 is strongly reduced at 6 hours, the decrease in TNF/IFN- γ production is already evident at this early timepoint;
- under soluble conditions, where the modulation is **weaker**, the reduction in cytokine-producing CD8⁺ T cells becomes statistically significant **only later**, once receptor engagement has been sustained.

Thus, both blocking strategies lead to down-modulation of surface KLRF1 and converge on the same biological conclusion—interference with KLRF1 reduces cytokine production—with the timing of the cytokine effect reflecting the **extent** of KLRF1 loss.

The reviewer also notes that after blockade the KLRF1⁺ cells can no longer be identified by flow cytometry. This is correct and expected: antibody-mediated receptor modulation is a well-documented phenomenon, including for other clinically used blocking antibodies such as anti-PD-1 (Saad *et al.*, 2024). Because the experimental question here is whether interference with KLRF1 signaling affects cytokine production, rather than comparing KLRF1⁺ and KLRF1⁻ cells within the same sample, the loss of detectable KLRF1 does not invalidate the experimental design. Instead, it confirms that the receptor is effectively engaged/neutralized. Importantly, the functional readout is measured across all CD8⁺ T cells in each condition, allowing us to assess how disruption of KLRF1 impacts the cytokine response of the terminally differentiated CD8⁺ T cell population as a whole.

These newly performed experiments and their corresponding findings have been incorporated into the revised manuscript and are presented on page 8 as well as in Figure 2.

6. *Furthermore, percentages may not be the best reflection of the effect of KLRF1 blockade in Figure 2c. How about MFI? If the receptor promotes production of these cytokines, perhaps you get more expression per cell, rather than more cells producing.*

Response:

We thank the reviewer for this constructive suggestion. To complement the frequency-based analyses in Figure 2, we quantified geometric mean fluorescence intensity (gMFI) for TNF and IFN- γ within the corresponding cytokine-positive CD8⁺ T-cell subsets. These analyses are provided in Rebuttal Figure 6.

We evaluated gMFI at 6 hours and 24 hours under KLRF1 blocking conditions in both CD3⁺ T-cell co-cultures and isolated CD8⁺ T-cell cultures. Overall, TNF gMFI changed only marginally upon KLRF1 blockade. A significant reduction was observed only in co-cultures (Rebuttal Figure 6A) at 6 hours, while all other TNF gMFI comparisons showed no meaningful differences.

For IFN- γ , isolated CD8⁺ T cells (Rebuttal Figure 6B) showed a modest reduction in gMFI at 6 hours. This trend was considerably more pronounced in co-cultures at 24 hours, where KLRF1 blockade resulted in a clear and significant decrease in IFN- γ gMFI.

Together, these analyses indicate that the main effect of KLRF1 blockade is a reduction in the frequency of cytokine-producing CD8⁺ T cells, with additional, more limited effects on per-cell cytokine intensity that depend on the cytokine, culture condition, and timepoint. These gMFI findings support the Figure 2 results and the overall conclusion.

*Rebuttal Figure 6. Geometric mean fluorescence intensity (gMFI) of IFN- γ and TNF in cytokine-positive CD8⁺ T cells after 6h and 24h stimulation. CD3⁺ T-cell co-cultures (A) and isolated CD8⁺ T cells (B) were treated with plate-bound anti-KLRF1 antibody or IgG1 isotype control. gMFI values were calculated for the IFN- γ ⁺ and TNF⁺ subsets as indicated. t-test with Benjamini & Hochberg adjusted p values: * $p < 0.05$. Boxplots show the median and interquartile range; whiskers indicate values within $1.5 \times IQR$.*

7. The following phrases seem overstatements: "Interestingly, GPR56 appears to negatively regulate AICL upregulation in CD4⁺ T cells," and: "GPR56 appears to negatively influence this process" (both page 12). The only justified conclusion is that there is a negative correlation between expression of the two.

Response:

We thank the reviewer for pointing this out and agree that our original wording could be interpreted as implying a regulatory mechanism that is not supported by the data. As suggested, we have revised both statements to accurately reflect the observational nature of the finding. The sentences now read:

Sentence 1 (replacing: "GPR56 appears to negatively regulate AICL upregulation..."):

"Interestingly, GPR56 expression was inversely associated with AICL expression in CD4⁺ T cells, as CD4⁺ T cells co-expressing KLRF1 and GPR56 showed markedly lower levels of AICL."

Sentence 2 (replacing: "GPR56 appears to negatively influence this process..."):

“GPR56 was inversely associated with this process, as CD4⁺ T cells co-expressing KLRF1 and GPR56 exhibited significantly reduced AICL expression, indicating a potential regulatory interaction, similar to the inhibitory role of GPR56 in NK cells (Chang *et al.*, 2016).”

This phrasing avoids implying causality and aligns with the conclusions justified by our experiments.

8. *Figure 3. The percentages in 3C are difficult to interpret. What proportion of the gated populations expresses these markers? This is important to interpret the effects from this ligand on the KLRF1+ CD8 T cells. Also, example gatings would be helpful to be able to judge the quality of the results.*

Response:

We thank the reviewer for this helpful comment. We agree that the original version of Figure 3C did not clearly convey what the percentages represent. To address this, we have revised both the figure legend and the y-axis labeling to ensure unambiguous interpretation.

First, we clarify that all percentages shown in Figure 3C refer to the proportion of AICL⁺ cells within the total gated CD4⁺ T cell population. In other words, the values indicate how many CD4⁺ T cells in each KLR/GPR56-defined subset upregulate AICL, relative to the entire CD4⁺ compartment. This presentation allows a direct comparison of how stimulation affects the distribution of AICL⁺ cells across these subsets.

Second, to avoid confusion between stimulated and unstimulated conditions, we now state explicitly that unstimulated and stimulated samples were normalized separately.

Third, in response to the reviewer’s request, we have updated the figure legend so that it now reads:

“CD3⁺-enriched T cells were stimulated with [...] and gated for CD4⁺ T cells. Shown is the proportion of total gated CD4⁺ T cells that expressed the indicated combinations of KLRG1, KLRB1, GPR56, and KLRF1.”

Fourth, the y-axis label has been adjusted for clarity and now reads: “% AICL⁺ of total CD4⁺ (unstimulated / stimulated, respectively)”.

Finally, to address the reviewer’s request for representative gating quality, we have added example gating plots to Appendix Figure S1, which depicts the gating into the two subpopulations which upregulate AICL the most, namely the CD4⁺ KLRG1⁺ KLRB1⁺ GPR56⁻ KLRF1⁺ and CD4⁺ KLRG1⁺ KLRB1 int. GPR56⁻ KLRF1⁺ T cells.

For convenience, Rebuttal Figure 7 below shows the main Figure 3C and Appendix Figure S1. To guide the reader, we used the same color-codes for the populations and markers as in Figure 3C.

Main Figure 3C:

Appendix Figure S1:

Rebuttal Figure 7: Updated version of **Figure 3C** (top panel) with the revised y-axis label (“% AICL⁺ of total CD4⁺”). Representative gating for the two CD4⁺ KLR/GPR56-defined subsets that predominantly

upregulate AICL is shown in the **Appendix Figure 1** (bottom panel). After gating on live CD4⁺ T cells, subsets were identified as KLRG1⁺ KLRB1⁻ GPR56⁻ KLRF1⁺ and KLRG1⁺ KLRB1⁺ GPR56⁻ KLRF1⁺, and AICL⁺ cells within these subsets were quantified as a fraction of total CD4⁺ T cells. Color-coding of the markers as well as the gated AICL⁺ populations matches that used in Figure 3C.

9. *Figure 4F does seem to show some differences in expression, and it may be significant statistically, but is it also significant biologically? I have to admit that I find the value on the Y axis difficult to interpret.*

Response:

We thank the reviewer for this question. Bulk expression of the gene encoding for AICL, *CLEC2B*, has been reported as biologically meaningful in several cancer contexts, where higher *CLEC2B* levels correlate positively with clinical outcome, e.g., (Guo *et al.*, 2021). This provides a rationale for examining AICL expression at the population level in our dataset.

We also agree that the y-axis labeling in the original version was not sufficiently intuitive. In mass cytometry datasets such as CyTOF or IMC, marker intensities are typically displayed using an arcsinh transformation to stabilize variance across a wide dynamic range while preserving linearity at low signal levels. Because the arcsinh transformation is already described in the figure legend, repeating it on the y-axis was unnecessary and reduced the clarity of the plot. We therefore revised the y-axis label to the simpler and more interpretable “AICL”. The plot title specifies that the values reflect AICL expression in CD4⁺ T cells, making the panel more accessible without changing the underlying data.

10. *There is an incorrect reference in the second paragraph on page 24 to supplementary image figure 6A. This does not exist. I think this should be supplementary figure 3A.*

Response:

We thank the reviewer for noticing this mistake. The correct reference was indeed Supplementary Fig. 3A, and we have updated the manuscript accordingly to reflect this correction.

11. *I have some doubt about whether the KLRF1⁺ CD8 T cells truly are Tpex cell as they seem to lack expression of TOX, according to SI Fig 4.*

Response:

We thank the reviewer for this comment. Absence of *TOX* expression in the CD8⁺ KLRF1⁺ T cell cluster does not contradict a Tpex-like phenotype. Indeed, *TOX* expression typically marks terminally exhausted (Tex) cells and not progenitor-exhausted (Tpex) cells as it is acquired after the progenitor-exhausted stage. Thus, Tpex cells are not expected to express *TOX*, and *TOX* negativity is in line with a progenitor-like state, e.g. (Kallies, Zehn and Utzschneider, 2020).

In our lung tissue dataset, CD8⁺ KLRF1⁺ T cells express *TCF7* and *CD127*, while showing low *GZMB* and low *TOX*, a transcriptional profile characteristic of memory-like or progenitor-

like CD8⁺ T cells rather than terminally exhausted cells. To further assess this phenotype, we reanalyzed multimodal scRNA-seq/ATAC data (Thomson *et al.*, 2023). Compared with terminally differentiated CD8⁺ KLRF1⁻ T cells, the KLRF1⁺ subset exhibits greater chromatin accessibility at loci associated with memory/Tpex biology (*IL7R*, *CXCR5*, *TCF7*) and reduced accessibility at exhaustion-associated loci such as *ENTPD1*. These results, now presented in Figure EV 5, provide additional epigenetic support for a Tpex-like identity (see response to comment 3 raised by reviewer 2).

Because functional assays to confirm fully defined Tpex behavior cannot be performed on human lung samples, we refer to these cells as “**Tpex-like.**” Nonetheless, the combined transcriptional and chromatin features strongly indicate that CD8⁺ KLRF1⁺ T cells more closely resemble a progenitor-exhausted-like population than a terminally exhausted one.

These newly performed experiments and their corresponding findings have been incorporated into the revised manuscript and are presented on pages 22ff as well as in Figure EV5.

Referee #2:

The manuscript submitted by Barone et al reports that terminally differentiated effector memory CD8⁺ T cells upregulate KLRF1 expression, enabling CD8-CD4 T cell interaction through the KLRF1-AICL axis that allow CD8⁺ T cells retain their capacity to produce cytokines like IFN γ and TNF α . Integrated spatial proteomics and single-cell transcriptomics studies of lung adenocarcinoma and adjacent tissues reveal that KLRF1⁺ CD8⁺ T cells exhibit T_{pex} gene signature, and that localized interactions between AICL⁺ CD4⁺ and KLRF1⁺ CD8⁺ T cells are identified in non-tumor regions but rare in tumor cores. The report is interesting because it may identify additional diagnostic markers to distinguish different subsets of exhausted T cells, and uncover AICL-KLRF1 as a potential therapeutic target to facilitate CD4-CD8 interactions and enhance CD8⁺ T cell functionality. However, a number of issues need to be addressed to enhance the rigor of the report.

- 1. The report is largely descriptive in nature. The major findings, like CD4⁺ T Cells promote IFN- γ and TNF production by CD8⁺ T Cells through AICL-KLRF1 axis, and direct CD4-CD8 interactions through AICL-KLRF1 prevent CD8 T cell exhaustion, are based on correlative studies. Overall, the data shown are indicative but not definitive.*

Response:

We thank the reviewer for this important point. We agree that the spatial proteomics and scRNA-seq analyses of lung adenocarcinoma samples are descriptive in nature. Their purpose is to define where CD4⁺ AICL⁺ and CD8⁺ KLRF1⁺ T cells reside within the tissue microenvironment and to map their potential interactions. To address the reviewer's concern that these analyses alone are correlative, we have substantially strengthened the mechanistic component of the study during revision.

- New AICL⁺ vs. AICL⁻ CD4⁺ T-cell sorting experiment (new Figure 3D-F):

As part of the revision, we performed a new functional experiment designed to directly test whether AICL expression in CD4⁺ T cells is required to support cytokine production in CD8⁺ T cells.

We FACS-sorted AICL positive and AICL negative CD4⁺ T helper cells (matched for differentiation markers KLRG1 and/or KLRF1) and co-cultured them with autologous CD8⁺ T cells at defined ratios (see also response to comment 1 by reviewer 1 & Rebuttal Figure 1).

- Only the AICL expressing CD4⁺ T cells enhanced TNF and IFN- γ production in CD8⁺ T cells.
- Co-culture with AICL negative CD4⁺ T cells failed to enhance cytokine output and resulted in significantly lower TNF/IFN- γ frequencies than AICL⁺ co-cultures.

This new experiment provides direct causal evidence that the helper effect is AICL-dependent, rather than a general property of differentiated CD4⁺ T cells.

These newly performed experiments and their corresponding findings have been incorporated into the revised manuscript and are presented on pages 10–11 as well as in Figure 3.

- Strengthened KLRF1 neutralization experiments (Figure 2):

We also expanded the previously included functional blocking experiments by:

- increasing donor numbers to N = 10,
- including both plate-bound and soluble anti-KLRF1 antibodies.

These additions were performed to build on the original blocking data already present in the manuscript. The soluble antibody experiments, added during revision, confirm that reduced cytokine production after KLRF1 interference is not an artifact of antibody cross-linking. Both neutralization methods converge on the same outcome—reduced TNF/IFN- γ production in CD8⁺ T cells, specifically in the presence of CD4⁺ T cells.

These newly performed experiments and their corresponding findings have been incorporated into the revised manuscript and are presented on page 10 as well as in Figure 2.

Together, the new AICL-sorting experiment and the strengthened KLRF1 neutralization data provide mechanistic support for the model derived from the descriptive omics analyses: CD4⁺ T-cell–derived AICL engages KLRF1 to help preserve cytokine-producing capacity in terminally differentiated CD8⁺ T cells.

These functional experiments now directly substantiate the central findings of the manuscript.

2. *In Fig. 2, a blocking anti-KLRF1 antibody was used to neutralize KLRF1 in CD8 T cells. However, the anti-KLRF1 antibody was plate-coated during T cell culture. The loss of KLRF1 on CD8 T cells seemed to be due to internalization of surface KLRF1 upon engagement with coated anti-KLRF1 antibody. This experimental setting is different from the typical Ab neutralization experiment in which the neutralizing Ab is in free form. The plate-coated Ab may cross-link KLRF1 expressed on the surface CD8 T cells, causing signal transmission similar to what AICL would do. The results should be validated by a different approach, for example, use of soluble aKLRF1 Ab, blocking of AICL, knockdown of KLRF1 or AICL, etc. In addition, the differences seen in the % of IFN γ +TNF α + CD8 T cells between the isotype and aKLRF1 conditions are really subtle (median ~10% vs ~8% at 6 hr), questioning whether there is any biological significance.*

Response:

We thank the reviewer for these important points. To address the concern regarding the plate-bound anti-KLRF1 antibody and potential receptor cross-linking, we performed new experiments using soluble anti-KLRF1 antibody, as suggested.

We repeated the blocking experiment with soluble anti-KLRF1, analyzing 10 donors at 6 hours and 24 hours in parallel with the plate-bound approach. These new data are included in the revised Figure 2B (see also response to comment 1 raised by reviewer 1, and Rebuttal Figure 1).

In these new experiments, both plate-bound and soluble anti-KLRF1 caused down-modulation of surface KLRF1, though with distinct magnitudes. Soluble antibody already produced a significant decrease in detectable KLRF1 at 6 hours and maintained this reduction at 24 hours, whereas plate-bound antibody consistently induced a more

pronounced loss of surface receptor. This graded reduction in KLRF1 was mirrored by the functional data: under plate-bound conditions—where KLRF1 down-modulation is strongest—the decrease in TNF/IFN- γ producing CD8⁺ T cells appeared early, while the weaker but significant modulation produced by soluble antibody translated into a later-onset reduction in cytokine production. The alignment between receptor modulation and functional output supports the conclusion that both formats effectively interfere with KLRF1 signaling.

These newly performed experiments, and corresponding findings have been added to the revised manuscript and are discussed on page 8 and presented in the revised Figure 2.

We acknowledge that the observed reduction is modest. This is consistent with the fact that only a minority of CD8⁺ T cells express KLRF1 and that the overall frequency of cytokine-producing cells under these stimulation conditions is relatively low. Importantly, we do not claim that the AICL–KLRF1 interaction is the only pathway positively regulating cytokine output in CD8⁺ T cells.

3. *The claim that KLRF1+CD8+ T cells are T_{pex} seemed to be solely dependent on gene signature derived from scRNAseq data. It is known that the spectrum of T cell exhaustion is critically defined by epigenetic landscapes. Without epigenetic profiling of the KLRF1+ CD8 T cells, it is premature to define these cells as T_{pex} cells.*

Response:

We thank the reviewer for this comment. We agree that assigning a definitive T_{pex} identity requires more than transcriptional signatures and that epigenetic landscapes are critical for distinguishing progenitor-exhausted (T_{pex}) from terminally exhausted (T_{ex}) states. For this reason, we have consistently referred to the CD8⁺ KLRF1⁺ T cell population as “**T_{pex}-like**” throughout the manuscript.

To further evaluate whether these cells display memory-/progenitor-like properties beyond transcriptional signatures, we re-analyzed a multimodal scRNA-seq + ATAC dataset (Thomson *et al.*, 2023). The resulting analyses are now provided in Figure EV 5 (Rebuttal Figure 8).

To define the CD8⁺ T cell compartment, we did not rely solely on the automatically generated T-cell annotations provided in the Thomson *et al.* (Thomson *et al.*, 2023) dataset, as these annotations occasionally assigned CD4 identity to cells with a clear CD8 transcriptional profile. Instead, we applied an extended version of the gating strategy used consistently throughout our manuscript. We first identified T cells based on $CD3E > 0$ and $(CD3D > 0 \mid CD3G > 0)$ together with $TRAC > 0$. We then gated CD8⁺ T cells using the condition $(CD8A > 0 \mid CD8B > 0 \mid (CD8A > 0 \ \& \ CD8B > 0))$ combined with exclusion of $\gamma\delta$ T-cell markers ($TRDC = 0 \ \& \ TRGC1 = 0 \ \& \ TRGC2 = 0$) and $CD4 = 0$. Applying this approach yielded a CD8⁺ T-cell subsets that included cells annotated as CD4⁺ by the automated method but whose expression pattern matched a CD8⁺ phenotype (Rebuttal Figure 8A). Applying this approach ensured that the full CD8⁺ T cell population was retained and that no CD8⁺ T cells were inadvertently omitted from our multimodal analyses.

The gated CD8⁺ T cells were clustered using a weighted-nearest-neighbor framework that integrates scRNA-seq PCA and ATAC LSI components. This analysis showed that CD8⁺ KLRF1⁺ T cells, marked by strong *KLRF1* expression, clustered almost exclusively in a single population (cluster 7, Rebuttal Figure 8B&C), whereas the most terminally

differentiated KLRF1⁻ cells fell into cluster 3. Both clusters were distinct from naïve and central-memory CD8⁺ T cells.

We next used the multimodal nature of the dataset to characterize the differentiation state of clusters 2, 7, and 3. Antibody-derived tag (ADT) signals (e.g., CD45RA, CD62L) and a heatmap of differentially expressed genes revealed that clusters 7 and 3 represent the most differentiated effector-memory cells (Rebuttal Figure 8D). Cluster 3 showed higher proportions and higher expression of CD45RA, consistent with a T_{EMRA}-like profile, whereas cluster 7 exhibited a more memory-like phenotype. At the transcriptional level, both clusters expressed effector genes such as *NKG7*, *GZMH*, *GZMB*, and *PRF1*, whereas cluster 2 displayed lower effector-gene expression.

To further distinguish potential Tpex-like from Tex-like states, we compiled a curated list of human Tex and Tpex marker genes from the literature (Brummelman *et al.*, 2018; Li *et al.*, 2018; Sade-Feldman *et al.*, 2018; Thommen *et al.*, 2018; Utzschneider *et al.*, 2018; Khan *et al.*, 2019; Miller *et al.*, 2019; Scott *et al.*, 2019; Yost *et al.*, 2019; Beltra *et al.*, 2020; Galletti *et al.*, 2020; Sekine *et al.*, 2020; Gabriel *et al.*, 2021; Zheng *et al.*, 2021; Sun *et al.*, 2023)(Rebuttal Figure 8E) and generated corresponding module scores. Mapping these scores onto the UMAP showed that KLRF1⁻ cluster 3 cells had significantly higher Tex and lower Tpex scores compared to KLRF1⁺ cluster 7 cells (Rebuttal Figure 8F).

Finally, we examined chromatin accessibility using ATAC gene-score comparisons between clusters 7 and 3 (Rebuttal Figure 8G). The KLRF1⁺ cluster showed significantly higher accessibility at loci associated with memory and progenitor states—including *IL7R*, *CXCR5*, and *TCF7*—while the exhaustion-associated *ENTPD1* locus, encoding for CD39, was markedly more accessible in the KLRF1⁻ cluster. These epigenetic patterns reinforce the conclusion that CD8⁺ KLRF1⁺ T cells retain memory-associated, progenitor-like features.

Taken together, the transcriptomic profiles, protein-marker signals, Tex/Tpex gene-module scoring, and ATAC accessibility differences collectively support the interpretation that CD8⁺ KLRF1⁺ T cells exhibit Tpex-like rather than Tex-like characteristics. Because functional validation is not possible in human lung tissue, we continue to use the cautious term “Tpex-like” in the manuscript.

These newly performed experiments and their corresponding findings have been incorporated into the revised manuscript and are presented on pages 22-23 as well as in Figure EV 5.

Rebuttal Figure 8: Multimodal scRNA-seq + ATAC analysis of $CD8^+$ T cells from (Thomson et al., 2023). (A) UMAP embedding of gated $CD8^+$ T cells colored by predicted T-cell types from the original annotation as provided by Thomson et al. in the dataset (predicted.t-celltype.l2). Major populations are color-coded similarly: naïve (green), TCM (blue), MAIT (pink), TEM/CTL (orange), and minor populations (Treg, dnT, $\gamma\delta$ T) shown

separately. **(B)** Leiden clusters obtained using a weighted nearest-neighbor (WNN) graph integrating scRNA-seq PCA and ATAC LSI components. **(C)** Left: UMAP colored by KLRF1 expression. Right: z-scored heatmap of KLR gene expression. **(D)** Antibody-derived tag (ADT) signals (top) and the most variable differentially expressed scRNA-seq genes (bottom), with lineage-informative markers color-coded to match differentiation groups in panel A. **(E)** UMAP embeddings of Tex and Tpex module scores calculated from the gene sets shown in the Appendix Table S8 **(F)** Comparison of T pex and Tex scores of the clusters 3 and 7. Boxplots show the median and interquartile range; whiskers indicate values within $1.5 \times$ IQR, p value calculated from t test. **(G)** Differential ATAC gene scores between clusters 7 and 3. Gene scores enriched in the KLRF1+ cluster 7 are $\log_2FC > 0$. Loci associated with memory/Tpex states and exhaustion and $FDR < 0.05$ are highlighted.

4. Some recent studies demonstrate that even terminally exhausted CD8 T cells can express effector molecules such as IFN γ , TNF α and Granzyme B, even at levels higher than Tpex cells. The authors should at least include this pool of literature to discuss the discrepancies.

Response:

We thank the reviewer for raising this important point. We agree that the functional capacity of exhausted CD8⁺ T cells is more heterogeneous than originally appreciated, and recent studies have shown that certain terminally exhausted (Tex) subsets can indeed retain or regain the ability to produce effector molecules such as IFN- γ , TNF, granzyme B, or perforin. This expanding body of literature provides important nuance, and we have now included these studies in the revised Discussion (page 28).

Several recent findings illustrate that terminal exhaustion does not equate to complete functional silence. For example, (Minnie *et al.*, 2024) demonstrated that TIM-3⁺ terminal Tex cells in hematologic malignancies can still produce granzyme B, perforin, and IFN- γ .

At the same time, several studies indicate that Tpex cells remain the main source of robust IL-2 and polyfunctional cytokine production, whereas terminal Tex cells show a shift toward IFN- γ production with diminished TNF- α output (Miller *et al.*, 2019; Beltra *et al.*, 2020; Wang *et al.*, 2022) similarly observed higher TNF- α expression in Tpex compared with Tex in human tumors, despite higher granzyme B in Tex. These patterns are consistent with our own observations in both scRNA-seq data sets and co-culture experiments, KLRF1⁺ (Tpex-like) cells are enriched for TNF, whereas KLRF1⁻ terminal clusters are enriched for late cytotoxic markers. This correspondence reinforces that our data fit within the established continuum of exhaustion.

Overall, these studies collectively highlight that terminally exhausted CD8⁺ T cells are not uniformly functionally inert, and that differentiation-molecule expression alone does not define exhaustion state. We have added these points to the revised Discussion to ensure full alignment with current understanding of T-cell exhaustion heterogeneity.

Minor point:

In Fig. 1B, it is confusing to see the arrows between the dot plots. Please specify whether the arrows indicate the sequential expression of the markers by CD8 T cells upon stimulation or are arbitrarily placed by the authors for comparison purpose.

Response:

We thank the reviewer for pointing this out. We agree that the arrows in the original version of Figure 1B could be misinterpreted as indicating a sequential or developmental

progression, which was not intended. Our only aim was to guide the reader across the different phenotypically defined CD8⁺ T cell subsets.

To avoid this confusion, we have removed all arrows entirely. This adjustment mirrors the revision made in response to Reviewer 1's similar comment. The updated panel is shown in the response to Reviewer 1, Minor Comment 1, Rebuttal Figure 5.

Referee #3:

This is a carefully executed study showing that CD4+ T cells contribute to the maintenance of pre-exhausted CD8+ T cells through interaction of AICL with KLRF1. Moreover the authors elegantly demonstrate that this interaction fails in tumor tissue.

I only have some minor details, primarily on the references. I noticed them as they refer to my own papers and propose that the authors cross-check other references as well.

1. *Fig1.B-D shows cytokine production. Please, clarify in text and legend whether/how cells have been stimulated.*

Response:

We apologize for this missing information. We added clarifying sentences in the beginning of the legend in Figure 1B and C to highlight that these were stimulated T cells: “**B) Exemplary dot plots of T cells from CD3⁺-enriched PBMCs stimulated for 6h depicting TNF versus IFN- γ of pre-gated subpopulations**”. The legend for Fig 1C. then reads: “**C) TNF and IFN- γ production of the stimulated subpopulations as defined in panel B**”.

2. *We already showed activation-induced, transient expression of AICL in T cells (hence its name) in Hamann et al, Immunogenetics 1997.*

Response:

We thank the reviewer for pointing this out. Your earlier work (Hamann *et al.*, 1997) was already cited in the Discussion, and we have now also included this reference at the point where AICL is first introduced. The sentence in the Introduction has been revised to:

“**Activated CD4⁺ T helper cells upregulate the KLRF1 ligand AICL (Hamann *et al.*, 1997)**” We also added the Hamann paper a second time where we introduce AICL. The modified sentence there reads “**Activation-induced upregulation of AICL has been described in lymphocytes (Hamann *et al.*, 1997) and expression has been reported to be expressed by monocytes, macrophages, granulocytes, and NK cells (Welte *et al.*, 2006)**”.

3. *Define NKC*

Response:

We would like to apologize for not defining Natural Killer gene complex in the introduction and added the abbreviation accordingly. The sentence now reads “**located in the Natural Killer gene complex (NKC) in a tail-to-tail orientation (Bartel, Bauer and Steinle, 2013)**”

4. The reference in the text to SI Fig.6A should be corrected into Fig.6A (without SI).

Response:

We thank the reviewer for drawing our attention to this mistake. The reference in the text is corrected now reference the proper figure panel Figure EV 3A, which shows that none of the genes examined—including *NKG7*—were exclusive to NK cells and that *NKG7* is also expressed by CD8⁺ T cells. This correction has been implemented in the revised manuscript on page 19.

5. The key paper for GPR56 as checkpoint is Chang et al, Cell Rep 2016 (PMID: 27184850). For the full history, see Hsiao et al, BCPT 2023 (PMID 36750420). - The sentence in the Discussion about AICL cites Vitale et al, EJI 2021. I assume this should be Welte et al, Nat Commun 2006.

Response:

We thank the reviewer for these helpful reference suggestions. We have now incorporated Chang et al., Cell Reports 2016 (Chang et al., 2016) into the section describing GPR56, specifically in the paragraph summarizing the association between GPR56 expression and reduced AICL levels in CD4⁺ T cells. The sentence now reads: “[...] GPR56 was inversely associated with this process, as CD4⁺ T cells co-expressing KLRF1 and GPR56 exhibited significantly reduced AICL expression, indicating a potential regulatory interaction, similar to the inhibitory role of GPR56 in NK cells (Chang et al., 2016).”

We also corrected the reference in the Discussion regarding AICL-induced NK cell activation. As suggested, we have replaced the citation Vitale et al., 2021 with the appropriate foundational study by Welte et al., Nature Communications 2006 (Welte et al., 2006).

In addition, we consulted Hsiao et al., BCPT 2023 (Hsiao et al., 2023) as recommended and ensured consistency with the historical context of GPR56 biology provided in the revised manuscript.

References

- Bartel, Y., Bauer, B. and Steinle, A. (2013) "Modulation of NK Cell Function by Genetically Coupled C-Type Lectin-Like Receptor/Ligand Pairs Encoded in the Human Natural Killer Gene Complex," *Frontiers in Immunology*, 4. Available at: <https://doi.org/10.3389/fimmu.2013.00362>.
- Beltra, J.-C. *et al.* (2020) "Developmental Relationships of Four Exhausted CD8+ T Cell Subsets Reveals Underlying Transcriptional and Epigenetic Landscape Control Mechanisms," *Immunity*, 52(5), pp. 825-841.e8. Available at: <https://doi.org/10.1016/j.immuni.2020.04.014>.
- Brummelman, J. *et al.* (2018) "High-dimensional single cell analysis identifies stem-like cytotoxic CD8+ T cells infiltrating human tumors," *The Journal of Experimental Medicine*, 215(10), pp. 2520–2535. Available at: <https://doi.org/10.1084/jem.20180684>.
- Chang, G.-W. *et al.* (2016) "The Adhesion G Protein-Coupled Receptor GPR56/ADGRG1 Is an Inhibitory Receptor on Human NK Cells," *Cell Reports*, 15(8), pp. 1757–1770. Available at: <https://doi.org/10.1016/j.celrep.2016.04.053>.
- Gabriel, S.S. *et al.* (2021) "Transforming growth factor- β -regulated mTOR activity preserves cellular metabolism to maintain long-term T cell responses in chronic infection," *Immunity*, 54(8), pp. 1698-1714.e5. Available at: <https://doi.org/10.1016/j.immuni.2021.06.007>.
- Galletti, G. *et al.* (2020) "Two subsets of stem-like CD8+ memory T cell progenitors with distinct fate commitments in humans," *Nature Immunology*, 21(12), pp. 1552–1562. Available at: <https://doi.org/10.1038/s41590-020-0791-5>.
- Guo, C. *et al.* (2021) "Mining TCGA Data for Key Biomarkers Related to Immune Microenvironment in Endometrial cancer by Immune Score and Weighted Correlation Network Analysis," *Frontiers in Molecular Biosciences*, 8, p. 645388. Available at: <https://doi.org/10.3389/fmolb.2021.645388>.
- Hamann, J. *et al.* (1997) "AICL: a new activation-induced antigen encoded by the human NK gene complex," *Immunogenetics*, 45(5), pp. 295–300. Available at: <https://doi.org/10.1007/s002510050208>.
- Hsiao, C.-C. *et al.* (2023) "The adhesion G protein-coupled receptor GPR56/ADGRG1 in cytotoxic lymphocytes," *Basic & Clinical Pharmacology & Toxicology*, 133(4), pp. 286–294. Available at: <https://doi.org/10.1111/bcpt.13841>.
- Kallies, A., Zehn, D. and Utzschneider, D.T. (2020) "Precursor exhausted T cells: key to successful immunotherapy?," *Nature Reviews. Immunology*, 20(2), pp. 128–136. Available at: <https://doi.org/10.1038/s41577-019-0223-7>.
- Khan, O. *et al.* (2019) "TOX transcriptionally and epigenetically programs CD8+ T cell exhaustion," *Nature*, 571(7764), pp. 211–218. Available at: <https://doi.org/10.1038/s41586-019-1325-x>.

- Li, J. *et al.* (2018) "High Levels of Eomes Promote Exhaustion of Anti-tumor CD8+ T Cells," *Frontiers in Immunology*, 9, p. 2981. Available at: <https://doi.org/10.3389/fimmu.2018.02981>.
- Miller, B.C. *et al.* (2019) "Subsets of exhausted CD8+ T cells differentially mediate tumor control and respond to checkpoint blockade," *Nature Immunology*, 20(3), pp. 326–336. Available at: <https://doi.org/10.1038/s41590-019-0312-6>.
- Minnie, S.A. *et al.* (2024) "TIM-3+ CD8 T cells with a terminally exhausted phenotype retain functional capacity in hematological malignancies," *Science Immunology*, 9(94), p. eadg1094. Available at: <https://doi.org/10.1126/sciimmunol.adg1094>.
- Saad, E.B. *et al.* (2024) "PD-1 endocytosis unleashes the cytolytic potential of checkpoint blockade in tumor immunity," *Cell Reports*, 43(11). Available at: <https://doi.org/10.1016/j.celrep.2024.114907>.
- Sade-Feldman, M. *et al.* (2018) "Defining T Cell States Associated with Response to Checkpoint Immunotherapy in Melanoma," *Cell*, 175(4), pp. 998-1013.e20. Available at: <https://doi.org/10.1016/j.cell.2018.10.038>.
- Scott, A.C. *et al.* (2019) "TOX is a critical regulator of tumour-specific T cell differentiation," *Nature*, 571(7764), pp. 270–274. Available at: <https://doi.org/10.1038/s41586-019-1324-y>.
- Sekine, T. *et al.* (2020) "TOX is expressed by exhausted and polyfunctional human effector memory CD8+ T cells," *Science Immunology*, 5(49), p. eaba7918. Available at: <https://doi.org/10.1126/sciimmunol.aba7918>.
- Sun, Q. *et al.* (2023) "BCL6 promotes a stem-like CD8+ T cell program in cancer via antagonizing BLIMP1," *Science Immunology*, 8(88), p. eadh1306. Available at: <https://doi.org/10.1126/sciimmunol.adh1306>.
- Thommen, D.S. *et al.* (2018) "A transcriptionally and functionally distinct PD-1+ CD8+ T cell pool with predictive potential in non-small-cell lung cancer treated with PD-1 blockade," *Nature Medicine*, 24(7), pp. 994–1004. Available at: <https://doi.org/10.1038/s41591-018-0057-z>.
- Thomson, Z. *et al.* (2023) "Trimodal single-cell profiling reveals a novel pediatric CD8 α + T cell subset and broad age-related molecular reprogramming across the T cell compartment," *Nature Immunology*, 24(11), pp. 1947–1959. Available at: <https://doi.org/10.1038/s41590-023-01641-8>.
- Utzschneider, D.T. *et al.* (2018) "Active Maintenance of T Cell Memory in Acute and Chronic Viral Infection Depends on Continuous Expression of FOXO1," *Cell Reports*, 22(13), pp. 3454–3467. Available at: <https://doi.org/10.1016/j.celrep.2018.03.020>.
- Wang, D. *et al.* (2022) "A comprehensive profile of TCF1+ progenitor and TCF1- terminally exhausted PD-1+CD8+ T cells in head and neck squamous cell carcinoma: implications for prognosis and immunotherapy," *International Journal of Oral Science*, 14(1), p. 8. Available at: <https://doi.org/10.1038/s41368-022-00160-w>.
- Welte, S. *et al.* (2006) "Mutual activation of natural killer cells and monocytes mediated by NKp80-AICL interaction," *Nature Immunology*, 7(12), pp. 1334–1342. Available at: <https://doi.org/10.1038/ni1402>.

Yost, K.E. *et al.* (2019) "Clonal replacement of tumor-specific T cells following PD-1 blockade," *Nature Medicine*, 25(8), pp. 1251–1259. Available at: <https://doi.org/10.1038/s41591-019-0522-3>.

Zheng, L. *et al.* (2021) "Pan-cancer single-cell landscape of tumor-infiltrating T cells," *Science (New York, N.Y.)*, 374(6574), p. abe6474. Available at: <https://doi.org/10.1126/science.abe6474>.

Dear Prof. Sawitzki,

Thank you for the submission of your revised manuscript to our editorial offices. I have already forwarded to you the reports I received from the two referees that I asked to re-evaluate your study. Please find them again below.

After going through your preliminary point-by-points response (further revision plan), and after cross-commenting with referees #2 (who now fully supports publication of the study) and #3 (who was already positive about the previous version), I have decided to proceed with the submission. Please address the remaining concerns of referee #1 in a final revised manuscript and/or in a final detailed point-by-point response, as indicated in your revision plan.

Please also acknowledge more explicitly the limitations of the study, particularly considering the rarity of ACL⁺CD4 T cells and the modest effects observed in CD8 T cell cytokine production. Please also tone down overly strong statements, e.g. the claim on page 12 ("After identifying the AICL-KLRF1 axis as a key driver of cytokine production..."). I would also agree with a title change, within the character limit (see below), and with as few abbreviations as possible.

Moreover, I have several editorial requests. Please also provide a p-b-p-response regarding these with your final submission.

Editorial requests:

- Please add keywords to five and order the manuscript sections like this, using only these names:

Title (100 characters including spaces) page - Abstract (max. 175 words) - Keywords - Introduction - Results - Discussion - Methods - Data availability section - Acknowledgements (please include here all the funding information) - Disclosure and Competing Interests Statement - References - Figure legends - Expanded View Figure legends

- Please format the author list in the manuscript similar to the author list in the submission system (First Name; Last Name). Moreover, please indicate the corresponding author and provide the corr. authors's email on the title page.

- We updated our journal's competing interests policy in January 2022 and request authors to consider both actual and perceived competing interests. Please name this section 'Disclosure and Competing Interests Statement' and put it after the Acknowledgements section.

See: <https://link.springer.com/partners/embo-press/editorial-policies#Competing%20interest%20disclosures>

- Please use our reference format (using et al after 10 author names):

<https://link.springer.com/journal/44319/submission-guidelines#cms-Reference-guidelines>

- Please check again that the number "n" for how many independent experiments were performed, their nature (biological versus technical replicates), the bars and error bars (e.g. SEM, SD) and the test used to calculate p-values is indicated in the respective figure legends (main, EV and Appendix figures). Please also check that all the p-values are explained in the legend, and that these fit to those shown in the figure. Please provide statistical testing where applicable. Please avoid the phrase 'independent experiment' but clearly state if these were biological or technical replicates. Please also indicate (e.g. with n.s.) if testing was performed, but the differences are not significant. In case n=2, please show the data as separate datapoints without error bars and statistics. See also:

<https://link.springer.com/journal/44319/submission-guidelines#cms-Figure-and-data-presentation>

If n<5, please show single datapoints for diagrams. Moreover:

- Please define the annotated p values ****/**/*/* as well as provide the exact p-values for the same in the legend of figure 3E, 5C, D; EV2 B as appropriate.

- Please note that the exact p values are not provided in the legends of figures 1C, D; 2C, D; 3B, C, F; 4F, G; EV5 F

- Please indicate the statistical test used for data analysis in the legends of figures 3E, 5C, D, G; EV2 B

- Please note that the box plots need to be defined in terms of minima, maxima, centre, bounds of box and whiskers, and percentile in the legends of figures 1C, D; 2C, D; 3B, C, E, F; 4F, G; 5C, D, F, G; 6E, F; EV2 B, F, G; EV4 E-H

- Please note that information related to n is missing in the legends of figures 3E, 4F, 5C, D, F, G; EV2 B, F, G; EV4 F, EV5 F."

- Please provide more detailed funding information and make sure that all the funding information is also entered into the online submission system and that it is complete and similar to the one in the acknowledgement section of the manuscript text file.

- All Materials and Methods need to be described in the main text using our 'Structured Methods' format, which is required for all research articles. According to this format, the Methods section should include a Reagents and Tools Table (listing key reagents, experimental models, software, and relevant equipment and including their sources and relevant identifiers), uploaded as separate file, and a Methods section in which we encourage the authors to describe their methods using a step-by-step protocol format with bullet points, to facilitate the adoption of the methodologies across labs. More information on how to adhere

to this format as well as downloadable templates (.doc) for the Reagents and Tools Table can be found in our author guidelines (section 'Structured Methods'):

<https://link.springer.com/journal/44319/submission-guidelines#cms-Manuscript-organisation-and-formatting>

- Please add the word "Appendix" to the table names and legends throughout the Appendix file. Please add the legends for the Appendix figures on the same page as the figure and remove the Appendix legends from the manuscript main text file. Please also remove the two separately uploaded Appendix figures. All Appendix items need to be uploaded as part to a single Appendix PDF file.

- Thanks for providing the source data. Please upload this as one folder per main figure, grouping together all the files for this figure (and ZIPped together).

In addition, I would need from you uploaded separately:

I look forward to seeing the further revised version of your manuscript when it is ready. Please let me know if you have questions regarding the revision.

Best,

Referee #1:

The authors have made a valiant effort to revise their manuscript and most of the points I raised have been addressed well. In general, the approaches taken in this study are sophisticated (with the exception of the functional experiment discussed below) and the question addressed is of broad interest to the field of tumor immunology as well as to the fundamental T cell immunology field.

However, I remain unconvinced about the central message of the article about the role of the AICL-KLRF1 interaction in mediating help from CD4 to CD8 T cells.

1. The reported effect sizes in experiments that aim to block the interactions between these molecules are small. In fact, the magnitude of the difference between the isotope control and the anti KLRF1 antibody is similar to the difference between the isotope control and the no antibody group.
2. Add to this the fact that the method to block the interaction is open to interpretation: it is possible that, as the authors argue, a plate bound antibody blocks a molecule by promoting its internalisation. However, plate immobilised antibodies also elicit cross-linking, which often causes activation of surface molecules. The fact that soluble antibodies also have some effect is not sufficiently convincing, as even those could be agonistic to some degree. It is difficult to know whether we are genuinely seeing the results of loss of function here, therefore. Granted, the fact that addition of AICL+ CD4 T cells elicits the opposite effect could be taken as evidence that the interaction is stimulatory, but this assay does not directly test this and, again, the effect size is very small.
3. Even if the specific inhibition data are statistically significant, their biological meaning is questionable. Given that the effect on the amount of cytokine produced per cell (gMFI) is marginal, the interpretation would have to be that the interaction between AICL-KLRF1 would allow a small percentage of CD8 T cells to respond that otherwise would not. It would help tremendously if the authors could identify such a population and show that on those cells, the effect size is much larger. Without that, I am afraid the results are insufficiently convincing to me.

Referee #2:

The authors have adequately addressed my previous review concerns with new data and revised the manuscript accordingly.

Point by Point Reply

We thank the editor and reviewers for their continued evaluation of our manuscript. In this final revision, we have provided a revised manuscript and a detailed point-by-point response addressing all remaining comments raised by referee #1, as well as all editorial requests outlined in the decision letter. In line with the editor's guidance, we have focused on clarifying the scope and limitations of the study, toning down overly strong statements, refining the manuscript framing (including the title), and explicitly acknowledging the modest effect sizes and the rarity of AICL⁺ CD4⁺ T cells, while avoiding additional experiments beyond the revised scope.

Reviewer's Comments:

Referee #1:

- The reported effect sizes in experiments that aim to block the interactions between these molecules are small. In fact, the magnitude of the difference between the isotope control and the anti-KLRF1 antibody is similar to the difference between the isotope control and the no antibody group.

- We agree with the referee that the observed effect sizes are modest. In the revised manuscript, we now explicitly acknowledge this limitation and have adjusted the wording throughout to avoid overstatement. Importantly, both AICL⁺ CD4⁺ T cells and KLRF1⁺ CD8⁺ T cells represent relatively rare subsets, and we do not propose the AICL–KLRF1 axis as a dominant or exclusive helper mechanism. Rather, our data support a contributory role of this interaction in maintaining cytokine competence within a subset of pre-exhausted CD8⁺ T cells. This more conservative interpretation is now clearly reflected in the Results, Discussion, and title.

- Add to this the fact that the method to block the interaction is open to interpretation: it is possible that, as the authors argue, a plate bound antibody blocks a molecule by promoting its internalisation. However, plate immobilised antibodies also elicit cross-linking, which often causes activation of surface molecules. The fact that soluble antibodies also have some effect is not sufficiently convincing, as even those could be agonistic to some degree. It is difficult to know whether we are genuinely seeing the results of loss of function here, therefore. Granted, the fact that addition of AICL⁺ CD4⁺ T cells elicits the opposite effect could be taken as evidence that the interaction is stimulatory, but this assay does not directly test this and, again, the effect size is very small.

- We acknowledge the referee's concern regarding the interpretation of antibody-based perturbation. To address this, our conclusions do not rely on a single blocking strategy. Instead, they are supported by three independent and mechanistically distinct approaches that converge on the same functional outcome:

1) Plate-bound anti-KLRF1, which induces rapid surface down-modulation of KLRF1 and reduces cytokine-producing CD8⁺ T cells.

2) Soluble anti-KLRF1 blockade, performed with matched donor numbers and time points, which also results in significant KLRF1 down-modulation and reduced cytokine production, albeit with weaker kinetics.

3) An antibody-independent, ligand-focused approach, in which AICL⁺ versus AICL⁻ CD4⁺ T helper cells were sorted and tested in co-culture. Only AICL⁺ CD4⁺ T cells enhanced cytokine production in CD8⁺ T cells, despite being otherwise matched for differentiation state.

- The convergence of these orthogonal strategies mitigates concerns regarding method-specific artefacts and supports the conclusion that AICL–KLRF1 interactions contribute to cytokine competence in CD8⁺ T cells.

- Even if the specific inhibition data are statistically significant, their biological meaning is questionable. Given that the effect on the amount of cytokine produced per cell (gMFI) is marginal, the interpretation would have to be that the interaction between AICL-KLRF1 would allow a small percentage of CD8 T cells to respond that otherwise would not. It would help tremendously if the authors could identify such a population and show that on those cells, the effect size is much larger. Without that, I am afraid the results are insufficiently convincing to me.

- We agree that identifying the CD8⁺ T-cell subset directly affected by AICL–KLRF1 interactions would have been informative. However, this was not technically feasible, as antibody-mediated interference led to rapid down-modulation of KLRF1 from the cell surface, preventing reliable gating on KLRF1⁺ cells within the same samples. This limitation applied to both plate-bound and soluble antibody conditions. We therefore based our interpretation on convergent evidence obtained from complementary approaches—receptor interference using two modalities and ligand-defined CD4⁺ T-cell co-culture—rather than on post-blockade identification of the responding subset.

Editorial requests:

1) General comment

- Please also acknowledge more explicitly the limitations of the study, particularly considering the rarity of AICL⁺CD4 T cells and the modest effects observed in CD8 T cell cytokine production. Please also tone down overly strong statements, e.g. the claim on page 12 ("After identifying the AICL-KLRF1 axis as a key driver of cytokine production..."). I would also agree with a title change, within the character limit (see below), and with as few abbreviations as possible.

- We thank the editor for this important guidance. In the revised manuscript, we now explicitly acknowledge the key limitations of the study, both in the Results and Discussion sections.
- First, the rarity of AICL⁺ CD4⁺ T cells is now stated directly in the Results section describing AICL induction (pages 10 and 11), where we clarify that AICL upregulation is restricted to a small subset of activated CD4⁺ T cells rather than representing a broad helper T-cell feature.
- Second, we explicitly acknowledge the modest magnitude of the observed effects on CD8⁺ T cell cytokine production in the Discussion (page 23), placing these findings in the context of limited responding cell fractions.
- In addition, we have revised the concluding Discussion to emphasize that the AICL–KLRF1 axis represents a contributory, rather than dominant or exclusive, mechanism supporting cytokine competence in pre-exhausted CD8⁺ T cells (pages 12, 19, 27). Overly strong phrasing has been removed throughout the manuscript, including the statement previously referring to the axis as a “key driver” (page 12).
- Finally, in line with the editor’s suggestion, we have revised the manuscript title to reduce abbreviations and better reflect the scope of the findings: *“The AICL–KLRF1 axis supports CD4–CD8 T cell communication and cytokine competence in pre-exhausted CD8⁺ T cells.”*

1) 2) Specific requests

- Please add keywords to five and order the manuscript sections like this, using only these names:

Title (100 characters including spaces) page - Abstract (max. 175 words) - Keywords - Introduction - Results - Discussion - Methods - Data availability section - Acknowledgements (please include here all the funding information) - Disclosure and Competing Interests Statement - References - Figure legends - Expanded View Figure legends.

- We checked again the section order and added Keywords alphabetically

- Please format the author list in the manuscript similar to the author list in the submission system (First Name; Last Name). Moreover, please indicate the corresponding author and provide the corr. authors's email on the title page.

- Author list format has been changed and corresponding author and her e-mail address added

- We updated our journal's competing interests policy in January 2022 and request authors to consider both actual and perceived competing interests. Please name this section 'Disclosure and Competing Interests Statement' and put it after the Acknowledgements section.

- We have added the section into the manuscript and checked the field in the Detailed Information tab during manuscript submission.

- Please use our reference format (using et al after 10 author names).

- Imported EMBO Reports style to Zotero and updated the document accordingly.

- Please check again that the number "n" for how many independent experiments were performed, their nature (biological versus technical replicates), the bars and error bars (e.g. SEM, SD) and the test used to calculate p-values is indicated in the respective figure legends (main, EV and Appendix figures). Please also check that all the p-values are explained in the legend, and that these fit to those shown in the figure. Please provide statistical testing where applicable. Please avoid the phrase 'independent experiment' but clearly state if these were biological or technical replicates. Please also indicate (e.g. with n.s.) if testing was performed, but the differences are not significant. In case n=2, please show the data as separate datapoints without error bars and statistics. See also:

<https://link.springer.com/journal/44319/submission-guidelines#cms-Figure-and-data-presentation>

- All Figure legends: changed "N" to "n" to match the formatting in EMBO and nature of replicate added to all legends

Moreover:

- Please define the annotated p values ****/***/**/* as well as provide the exact p-values for the same in the legend of figure 3E, 5C, D; EV2 B as appropriate.

- Generally: we have added the exact p values to all graphs to make it consistent throughout the Figures and skipped the annotation of p values wherever possible, meaning: only "ns" is annotated and mentioned in the legend. Figure 4G remains annotated and exact p values are

provided in the legend due to space restrictions in the panel itself. Figure 5F and G remain annotated since p values are all below 0.0001 and annotation is provided in legend text.

- Figure 3B, E and 3F exact p values added to figure
- Figure 5C, 5D annotation and exact p values added to figure legend
- Extended Figure 2B, annotated and exact p values added

- Please note that the exact p values are not provided in the legends of figures 1C, D; 2C, D; 3B, C, F; 4F, G; EV5 F.

- Figure 1C, 1D exact p values were added into the plot
- Figure 2B,C; exact p values added into figure. “ns” annotation added to legend
- Figure 3B,C,E,F: exact p-values added in plot
- Figure 4F exact p value added in plot, 4G: exact p values provided in the legend (see comment above)
- Figure 5C, 5D, exact p values were added into figure. 5F, 5G annotation provided in legend
- Extended Figure 2B, exact p values were added in plot
- Extended Figure 5F mapping **** $p < 0.0001$ checked in the legend, annotated p values added in the panel F

- Please indicate the statistical test used for data analysis in the legends of figures 3E, 5C, D, G; EV2 B

- Figure 3E, statistical test was added. It was mentioned in 3F already and is now indicated in 3E and 3F.
- Figure 5C and 5D, statistical tests and p values have been added to Fig5C and 5D legend.
- Statistical test in figure 5G was already mentioned in the legend. It now clearly reads: “For panels E and F, Wilcoxon rank sum test with Benjamini & Hochberg adjusted p values was used to compare differences”
- Extended Figure 2B, statistical test added in figure legend.

- Please note that the box plots need to be defined in terms of minima, maxima, centre, bounds of box and whiskers, and percentile in the legends of figures 1C, D; 2C, D; 3B, C, E, F; 4F, G; 5C, D, F, G; 6E, F; EV2 B, F, G; EV4 E-H

- Figure 1C and 1D, correct box plot definitions added in figure legend
- Figure 2B and 2C, correct box plot definitions added in figure legend
- Figure 3B, 3C, 3E, 3F, correct box plot definitions added in figure legend
- Figure 4F whiskers had same color as datapoints and were not visible: Box plot color darkened and added correct box plot definition in figure legend
- Figure 4G, correct box plot definition added in the figure legend
- 5C, 5D, 5F, 5G, correct box plot definitions added in the figure legend
- Figure 6E and 6F, correct box plot definitions added in figure legend.
- Extended Figure 2B, 2F and 2G, all box plot definition added in figure legend
- Extended Figure 4E, 4F, 4G&H, all box plot definitions added in figure legend

- Extended Figure 5F. Even though missing in the request, box plot definitions were added there as well.

- Please note that information related to n is missing in the legends of figures 3E, 4F, 5C, D, F, G; EV2 B, F, G; EV4 F, EV5 F."

- Figure 3E, n and replicate nature added
- Figure 4F n added
- Figure 5C, 5D, n and replicate nature added
- Figure 5F, 5G, n and replicate nature added for each panel
- Extended Figure 2B, F, n and replicate nature added
- Extended Figure 4F (and E)
- Extended Figure 5F, n and replicate nature added

- Please provide more detailed funding information and make sure that all the funding information is also entered into the online submission system and that it is complete and similar to the one in the acknowledgement section of the manuscript text file.

- Funding information has been added into the Acknowledge section and added into the online submission field upon uploading

- All Materials and Methods need to be described in the main text using our 'Structured Methods' format, which is required for all research articles. According to this format, the Methods section should include a Reagents and Tools Table (listing key reagents, experimental models, software, and relevant equipment and including their sources and relevant identifiers), uploaded as separate file, and a Methods section in which we encourage the authors to describe their methods using a step-by-step protocol format with bullet points, to facilitate the adoption of the methodologies across labs. More information on how to adhere to this format as well as downloadable templates (.doc) for the Reagents and Tools Table can be found in our author guidelines (section 'Structured Methods'):

<https://link.springer.com/journal/44319/submission-guidelines#cms-Manuscript-organisation-and-formatting>

- Reagent Table.docx has been made according to the guidelines

- Please add the word "Appendix" to the table names and legends throughout the Appendix file. Please add the legends for the Appendix figures on the same page as the figure and remove the Appendix legends from the manuscript main text file. Please also remove the two separately uploaded Appendix figures. All Appendix items need to be uploaded as part to a single Appendix PDF file.

- For the revision, we included the two Appendix Figures in the upload and the corresponding legends in the main text for convenient reading. The figures have been in the Appendix PDF, along with the legends in said PDF file. We have now removed the legends from the text.
- The title of the manuscript has been changed in the appendix PDF file as well
- “Appendix” has been added in all table names

- Thanks for providing the source data. Please upload this as one folder per main figure, grouping together all the files for this figure (and ZIPed together).

- The Source Data has been re-zipped into separate .zip files for each figure and uploaded

In addition, I would need from you uploaded separately:

Both, the short summary and the bullet points are uploaded as separate word files but for convenience we write the content of the files here:

- a short, two-sentence summary of the manuscript (not more than 35 words).

- Late-differentiated CD8⁺ KLRF1⁺ T cells receive cytokine-supporting signals from CD4⁺ AICL⁺ T cells. Spatial and single-cell analyses show both populations are reduced in lung tumors, alongside diminished TNF and IFN-γ expression. (31 words)

- two to four short (!) bullet points highlighting the key findings of your study (two lines each),

- AICL–KLRF1 interactions contribute to cytokine competence in pre-exhausted CD8⁺ T cells
- T_{pex}-like CD8⁺ KLRF1⁺ T cells are enriched in non-tumor lung tissue and reduced in tumors

- a schematic summary figure as separate file that provides a sketch of the major findings (not a data image) in jpeg or tiff format (with the exact width of 550 pixels and a height of not more than 400 pixels) that can be used as a visual synopsis on our website.

- Synopsis is attached (550x400px at 600dpi in RGB)

Prof. Birgit Sawitzki
Berlin Institute of Health (BIH)
Center of Immunomics
Chariteplatz 1
Berlin, Berlin 10117
Germany

Dear Prof. Sawitzki,

Thank you for the submission of your final revised manuscript to our editorial offices. It now went through this and your final p-b-p-response and consider the remaining points of referee #1 and the editorial requests as adequately addressed.

I am thus very pleased to accept your manuscript for publication in the next available issue of EMBO reports. Thank you for your contribution to our journal.

You may qualify for financial assistance for your publication charges - either via a Springer Nature fully open access agreement or an EMBO initiative. Check your eligibility: <https://link.springer.com/journal/44319/how-to-publish-with-us>

Yours sincerely,

>>> Please note that it is EMBO Reports policy for the transcript of the editorial process (containing referee reports and your response letter) to be published as an online supplement to each paper. If you do NOT want this, you will need to inform the Editorial Office via email immediately. More information is available here: <https://link.springer.com/partners/embo-press/editorial-policies#Peer%20review>